# Research and Developments of Heterogeneous Catalytic Technologies [note 1]

**DOI:** 10.3390/molecules30153279

**Published:** 2025-08-05

**Authors:** Milan Králik, Peter Koóš, Martin Markovič, Pavol Lopatka

**Affiliations:** Institute of Organic Chemistry, Catalysis and Petrochemistry, Slovak University of Technology, Radlinského 9, 812 37 Bratislava, Slovakia; peter.koos@stuba.sk (P.K.); martin.markovic@stuba.sk (M.M.); pavol.lopatka@stuba.sk (P.L.)

**Keywords:** heterogeneous catalyst, kinetics, mass and heat transport, process design

## Abstract

This review outlines a comprehensive methodology for the research and development of heterogeneous catalytic technologies (R&D_HeCaTe). Emphasis is placed on the fundamental interactions between reactants, solvents, and heterogeneous catalysts—specifically the roles of catalytic centers and support materials (e.g., functional groups) in modulating activation energies and stabilizing catalytic functionality. Particular attention is given to catalyst deactivation mechanisms and potential regeneration strategies. The application of molecular modeling and chemical engineering analyses, including reaction kinetics, thermal effects, and mass and heat transport phenomena, is identified as essential for R&D_HeCaTe. Reactor configuration is discussed in relation to key physicochemical parameters such as molecular diffusivity, reaction exothermicity, operating temperature and pressure, and the phase and “aggressiveness” of the reaction system. Suitable reactor types—such as suspension reactors, fixed-bed reactors, and flow microreactors—are evaluated accordingly. Economic and environmental considerations are also addressed, with a focus on the complexity of reactions, selectivity versus conversion trade-offs, catalyst disposal, and separation challenges. To illustrate the breadth and applicability of the proposed framework, representative industrial processes are discussed, including ammonia synthesis, fluid catalytic cracking, methanol production, alkyl tert-butyl ethers, and aniline.

## 1. Introduction

Catalytic processes occupy a central role in modern chemical and biochemical technologies [1,2,3,4,5]. The fundamental characteristics of catalysis can be succinctly defined by three principal features: (i) acceleration of the chemical reaction rate; (ii) invariance of the thermodynamic equilibrium composition at a given temperature and pressure; and (iii) the catalyst is not consumed during the reaction. However, the latter does not preclude structural or textural modifications of the catalyst during operation; for example, the transformation of iron catalysts during ammonia synthesis or of platinum catalysts during ammonia oxidation in nitric acid production [6]. In both cases, initially robust agglomerates of oxides and/or alloys evolve into porous skeletal structures with significantly increased specific surface area, thereby enhancing catalytic activity. The widely accepted mechanistic basis for catalytic action is the lowering of the activation energy barrier, which is achieved through specific interactions between reactants and catalytic centers [3,7]. These interactions influence the energies of the frontier molecular orbitals—HOMO (Highest Occupied Molecular Orbital) and LUMO (Lowest Unoccupied Molecular Orbital)—facilitating reactants’ easier participation in the transformation process. The strength of these interactions is often quantified by adsorption heat, which has long been associated with catalytic activity through the concept of the *volcano plot*. This empirical relationship correlates the reaction rate with adsorption heat, suggesting an optimal intermediate value as per the Sabatier principle [8], which remains relevant even in contemporary catalytic fields, including single-atom catalysis (SAC) and electrocatalysis [9]. Earlier theories also explored the role of d-orbital electron configurations in transition metals, where vacant d-orbitals can interact with electron-donating groups (EDGs), such as nucleophilic species, while electron-withdrawing groups (EWGs) interact with lone pair-containing catalyst sites (e.g., N or P-based functionalities) [8,10]. Geometric factors—such as the spatial arrangement of surface atoms enabling favorable chemisorption—were also recognized, with benzene adsorption on metal nickel surfaces being a classic example. Terms like coherent, semicoherent, and incoherent interfaces [11] were recently introduced in connection with photocatalysis [12]. However, these concepts are also highly relevant for classical (nano)composite catalysts. Coherent interfaces arise from perfect lattice matching across the interface, while an “ideal” incoherent interface exhibits no lattice matching. Surface energy is lowest for coherent interfaces (0 to 200 mJ·m^−2^), increases for semicoherent ones (200–500 mJ·m^−2^), and is relatively high for incoherent interfaces (500–1000 mJ·m^−2^). Nanometric “jumps” in surface energy strongly affect the chemisorption or interactions of reactants, intermediates, and products, as well as photon adsorption in photocatalysis. These interactions enable catalysis to proceed efficiently at lower temperatures compared to non-catalyzed reactions [3], often leading to reduced energy requirements, simpler reactor design, suppression of side reactions, and improved selectivity and yield [1]. Additionally, specific catalysts exhibit stereoselectivity, a hallmark of many enzymatic and biochemical processes [3,4,5,13].

Catalytic reactions proceed on catalytic centers, represented either by specific chemical moieties (e.g., –SO_3_H, –OH, organometallic complexes) or by structural features of solid materials, such as edges, corners, steps, and vacancies, which locally alter surface energy. Catalytic systems and catalysts are generally classified into three major categories [3,14].

Homogeneous: The catalyst and reactants are in the same phase, typically liquid. Gas-phase homogeneous catalysis is rare but exemplified by the oxidation of SO_2_ to SO_3_ using nitrogen oxides.Heterogeneous: The catalyst and reactants are in different phases, usually involving solid catalysts interacting with gaseous, vapor, and/or liquid reactants.Biocatalysis: Catalysis mediated by whole microorganisms or isolated enzymes, typically in the liquid phase. While classical biocatalytic processes include alcohols [15,16] and citric acid production [17,18,19,20], more recent developments involve engineered enzymes for specialty chemical [21,22,23,24,25,26] and pharmaceutical synthesis, including enantioselective transformations and active pharmaceutical ingredient (API) production [27,28,29,30].

A related field, organocatalysis, involves catalysis by non-metal heteroatoms [31]. While fundamentally a type of homogeneous catalysis, it often exhibits hybrid characteristics [3,32].

A rapidly growing area is SAC [33], where isolated metal atoms anchored to solid supports act as well-defined active catalytic centers. Their interaction with supports modulates reaction activity through strong metal–support effects [34].

Catalysis under external stimuli has led to the classification of additional catalytic systems, including (i) electrocatalysis [35]; (ii) microwave catalysis [36]; sonocatalytic processes [37]; (iii) mechanocatalysis [38]; (iv) magnetodriven catalysis [39]; (v) photocatalysis, including light-driven processes [40]; (vi) plasmonic catalysis [41]; (vii) piezocatalysis and piezoelectrocatalysis [42,43,44]; and (viii) combined procedures, e.g., plasmonic and electromagnetic [45], enhancement of mass transport by magnetic field in electrocatalysis [46], piezo-assisted photocatalysis [44], photothermal [47], microwave and ultrasound [48], and others.

A noteworthy class within heterogeneous catalysis is hybrid catalysts, where homogeneous active moieties are supported on solids. These can be further divided into (i) Heterogenized catalysts, where active groups (e.g., –SO_3_H, –OWO_2_, organometallic complexes) are chemically bonded to organic polymers [49] or inorganic supports [50] with anchored functional (catalytic) groups, including SAC [51]; (ii) immobilized or supported catalysts, where catalytic entities (e.g., metal nanoparticles [52], organocatalytic groups [53], or enzymes [54]) are physically adsorbed or held by electrostatic forces. In such systems, the nature of the support significantly influences overall catalytic performance compared to homogeneous origins.

Choosing between homogeneous and heterogeneous catalysis involves complex trade-offs. For gas-phase reactions (e.g., ammonia synthesis, SO_2_ oxidation, oxidation of naphthalene to phthalic anhydride), heterogeneous catalysis is typically preferred [55]. In liquid-phase systems, heterogeneous catalysts offer easier separation [7,56,57], but advances in separation technologies have enabled efficient recycling of homogeneous catalysts [58,59]. However, heterogeneous systems often suffer from limitations in mass and heat transport, which can lead to local hot spots, fast deactivation, and reduced selectivity [14,60,61].

Catalyst development has traditionally focused on intrinsic activity under ideal (kinetic) conditions, minimizing transport limitations [14,56]. Industrial applications, however, demand scalable solutions involving larger catalyst particles, optimized distribution of active sites, and compatible reactor designs [56]. “Real-world” examples include low-loading metal catalysts on alumina for selective hydrogenation, honeycomb-supported transition metals for exhaust treatment, and hydrodeoxygenation catalysts for green diesel production [1,62].

A frequent concern among process engineers is the limited practical applicability of lab-scale catalyst data. Effective process design requires comprehensive experimental data, including catalyst lifetime, resistance to impurities in the feed, sensitivity to operating conditions, and regeneration strategies. For illustration, a Web of Science (WOS) search conducted on 7 July 2025, yielded the following publication counts: “heterogeneous catalyst” (193,393); AND “design” (103,053); AND “reactor” (26,455); AND “deactivation” (5529); AND “mass transport” (244); AND “heat transport” (10). While it is often argued that application-oriented data are proprietary or protected by patents, it is equally important to recognize that complex experimental research facilitates the transfer of academic discoveries into industrial practice.

This review seeks to encourage more integrative studies among catalyst researchers, promoting the translation of fundamental insights into applied technologies. Short descriptions of catalyst design strategies and properties, coupled with relevant industrial or emerging applications, are presented to serve as a foundation for future interdisciplinary efforts, particularly in the context of green chemistry and sustainable development [63,64,65,66]. In comparison with other typical catalytic papers, the presented one emphasizes the connections between catalytic phenomena at the atomic/molecular level, kinetics, mass and heat transport, and the design of appropriate equipment. Naturally, these areas cannot be discussed in detail within a single paper. Therefore, the review includes relevant references where the reader can find more comprehensive information. The text is organized in a highly structured manner, with chapters, subchapters, numbered sections, and bullet points. This structure is intended to help the reader navigate the topics more easily and understand the relationships among them.

## 2. Properties of Catalysts

Catalyst properties can be systematically characterized by six main groups of physicochemical parameters [67].

(i)Chemical composition and crystallographic structure.(ii)Texture and physical–chemical properties.(iii)Temperature and chemical stability.(iv)Mechanical stability.(v)Mass, heat, and electrical transport properties.(vi)Catalytic performance.

Evaluation of these quantities is carried out by diverse chemical and physicochemical characterization methods; details see, e.g., [68].

### 2.1. Chemical Composition and Crystallographic Structure

The characterization of catalyst composition and structure varies significantly between homogeneous and heterogeneous systems.

For homogeneous catalysts, the focus is primarily on the following:Chemical analysis and determination of the molecular structure, typically achieved through classical spectroscopic and analytical techniques.

For heterogeneous catalysts, a wider array of techniques is employed due to their more complex and multiphase nature. These include the following:Chemical analysis of both the support material and the active catalyst with deposited metal, often involving decomposition in acidic media, followed by atomic absorption (AAS) or emission spectrophotometry—Inductively Coupled Plasma Optical Emission Spectroscopy and Mass Spectroscopy (ICP OES, ICP MS).Nuclear Magnetic Resonance (NMR) performed in both solution and solid state to elucidate the structural features of the support, including surface functional groups.X-ray reflection spectroscopy (XRF) to characterize surface composition.X-ray Powder Diffraction (XRPD), used for phase identification and crystallographic analysis; also applicable for estimating average crystallite size via the Scherrer equation.Electron Diffraction X-ray Analysis (EDX), often coupled with electron microscopy, to determine surface composition and the spatial distribution of metal species.Wavelength-dispersive X-ray spectroscopy (WDS, WDX), offering enhanced sensitivity over EDX for surface elemental analysis.Fourier Transform Infrared Spectroscopy (FTIR), utilized for the identification of surface functional groups and chemical bonding environments.Raman spectroscopy for characterizing molecular and surface species.X-ray photoelectron spectroscopy (XPS), critical for determining the valence state of metal particles on the surface.

### 2.2. Texture and Physical–Chemical Properties

Determination of porosity, external and internal specific surfaces, and pore size distribution.

Particle size distribution measurements in the range of 10 nm to 5 mm.Optical microscopy for determining particle size, shape, and surface texture (resolution: 250 nm; high-quality laser scanning confocal microscopy (LSCM): resolution down to 0.5 nm).Raman spectroscopy for identifying surface phases and irregularities, e.g., quantification of graphite and disordered carbon content on activated carbon [69].Nitrogen or krypton adsorption–desorption isotherms for determining porosity, specific surface areas (external and internal), and pore size distribution. Various adsorption–desorption models are used [70,71], covering pore sizes from 0.1 to 100 nm.Mercury porosimetry (applicable only to mechanically stable materials) for pore size distribution ranging from 7.5 nm at 200 MPa to 15 μm at atmospheric pressure.Transmission electron microscopy (TEM), including High-Resolution TEM (HR-TEM) and Scanning TEM (STEM), for determining the particle size distribution of metal crystallites, atomic arrangement, and crystallographic phases (resolution: 0.1 nm).Chemisorption of H_2_ or CO for evaluating the specific surface area of metallic crystallites or agglomerates (applicable when internal volume is accessible—see [49]).

A different approach must be taken for OMOP-based and biocatalysts, which typically exhibit significant textural changes due to strong interactions with the reaction environment (reactants, products, solvents). These materials can exhibit swellability (often referred to as “breathing”). Therefore, they must be characterized in a state equivalent to their condition in the actual reaction system [15,49].

Acido basic properties:Titration with basic components, e.g., NaOH solution, to determine acidity.Adsorption measurements of basic components (e.g., NH_3_, organic amines) to evaluate acidity; FTIR spectroscopy is used to identify Lewis and Brønsted acid sites.Titration with acidic components, e.g., HCl, to determine alkalinity.Adsorption measurements of acidic components, e.g., CO_2_, to determine alkalinity.

Hydrophilic and hydrophobic characteristics play a crucial role in catalytic systems where water is produced (e.g., hydrogenation of nitro compounds to amines), in condensation reactions (e.g., formation of ketimines from ketones and amines), or when water is present in the feed as an impurity (a common occurrence). Water molecules can adsorb onto the catalyst surface, occupying active sites and hindering catalytic reactions. An exception to this behavior arises when a hydrophilic agent is required to open up the internal structure of a hydrophilic material, such as a gel-type polymer porous particle [49,72,73].

The main methods used for evaluation of hydrophilic/hydrophobic properties are as follows:Contact angle (CA)—Water is used as the reference hydrophilic liquid. A CA value less than 90° indicates a hydrophilic surface, while a value greater than 90° indicates a hydrophobic surface. This is a static measurement method. Powder samples must be compacted into larger particles with a “flat surface” (e.g., tablets) to enable accurate measurements.Washburn dynamic method (WDM)—A liquid with defined polarity rises through a porous medium in a cylindrical setup; both the rate of wetting and the maximum height achieved are recorded [74]. The referenced study also provides useful correlations between CA and WDM data.Adsorption of adsorptives with a different polarity—This method evaluates surface polarity based on the adsorption behavior of molecules with varying polarities. Surface roughness significantly affects the results [75].Zeta Potential (*ζ-*Potential)—Values close to zero suggest low surface polarity and are typically associated with hydrophobic surfaces. However, establishing a quantitative correlation between zeta potential and hydrophilicity is complex and not straightforward [76].

Special attention must be paid to hydrophilic samples that adsorb the probing molecules into their internal volume, leading to changes in internal geometry—such as swelling or “breathing”—as observed in OMOP catalysts [49]. For these types of catalysts/materials, additional parameters are used to characterize their hydrophilic properties, including swellability and Water Retention Value (WRV), which is the capacity of a sample—originally defined for wood fibers or pulp—to retain water. These parameters are also useful for evaluating changes in material properties after thermal-humidity treatment (artificial aging) [77].

### 2.3. Temperature and Chemical Stability

Thermogravimetric analysis (TGA)—Measures the extent of decomposition of the carrier or catalyst as temperature increases. TGA can be performed in an oxidative atmosphere (Temperature Programmed Oxidation—TPO) or a reductive atmosphere (Temperature Programmed Reduction—TPR).Differential scanning calorimetry (DSC)—Measures decomposition and thermal effects, such as the release of water from the crystalline lattice or pyrolysis effects in organic polymer carriers. DSC is often followed by analysis of degradation products (e.g., using GC or GC-MS).(Micro)Pyrolysis combined with pyrolysis product analysis—Similar to DSC with follow-up analysis, but carried out under different conditions to study pyrolysis products in more detail.Hydrolytic, Acidic, Basic, Aminolytic, Alcoholytic, and Other Stability Tests—These tests are particularly important for evaluating the stability of polymer-based catalysts.

### 2.4. Mechanical Stability

A detailed description of the mechanical characteristics of catalyst particles (e.g., extrudates, spheres, cylinders, cylinders with holes, star-shaped forms, etc.) and the methods used to measure them can be found in [78]. The main categories of mechanical properties include the following:Crush Strength;Young’s Modulus of Elasticity;Breakage by Collision;Breakage by Stress in a Fixed Bed;Breakage in Contiguous Equipment.

Similar to textural properties, OMOP and bio-catalysts must be characterized more specifically, considering their interactions with components of the reaction system. These interactions can lead to reduced mechanical stability.

### 2.5. Mass, Heat, and Electrical Transport Properties

The following properties are particularly important for larger catalyst particles (>1 mm), commonly used in fixed-bed or tubular reactors [6,56]:Diffusivity in pores—Includes bulk diffusion, Knudsen diffusion, wall effects, and the impact of tortuosity.Thermal conductivity—Critical for heat management in exothermic or endothermic reactions.Hydrodynamic resistance—Resistance of the catalyst bed or layer to the flow of gas/vapor or gas–vapor–liquid mixtures.Electrical conductivity—Particularly important for catalyst supports, as it strongly influences metal–support interactions.Character of a semiconductor’s surface—Especially relevant for metal oxides (e.g., ZnO, NiO, Fe_2_O_3_, V_2_O_5_, Cr_2_O_3_). These materials can exhibit n-type behavior (donating electrons to chemisorbed molecules) or p-type behavior (withdrawing electrons) [8,79]. This distinction is important for quantifying Van der Waals interactions and their influence on HOMO and LUMO energy levels, see, e.g., [80]. Measurements of isoelectric points provide an initial estimation of surface electronic properties [81].

### 2.6. Catalytic Performance

To evaluate the catalytic performance of a material, the following terms are commonly used [82]:**Katal, kat**—SI coherent derived unit for catalytic activity, defined as the amount of reactant (in moles) converted per second: (mol_reactant_·s^−1^).**Molar Catalytic Activity (kat/mol)**—Catalytic activity per mole of catalyst: (mol_reactant_·s^−1^·mol_catalyst_^−1^).**Mass Catalytic Activity (kat/m_catalyst_)**—Catalytic activity per unit mass of catalyst: (mol_reactant_·s^−1^·kg_catalyst_^−1^).**Turnover Frequency (TOF)** [83]—Also expressed as (mol_reactant_·s^−1^·mol_catalyst_^−1^), TOF is functionally identical to kat/mol but predates it and remains widely used, especially in heterogeneous catalysis. TOF refers to the number of reactant molecules converted per active site per second. It is crucial to specify how the number of active sites is determined, e.g., by acidity, accessible metal atoms via chemisorption, etc. Additionally, temperature, pressure, and reactant concentrations must be reported to fully specify **TOF**.**Turnover Number (TON)**—Defined as the number of moles of reactant converted per mole of catalyst during its lifetime: (mol_reactant_·mol_catalyst_^−1^). TON is useful for describing catalyst durability in non-regenerated systems (e.g., in continuous reactors over time on stream). However, it is generally discouraged, especially when estimated from batch reactors, as it can be misleading [83].**Catalyst productivity (CatProd)**—Expressed in the same units as TON (mol_reactant_·mol_catalyst_^−1^), this parameter describes the total productivity of a catalyst under practical conditions. While not an official IUPAC term, **CatProd** is useful in process design and techno-economic evaluations. It requires well-defined experiments, typically continued to the point at which the catalyst is no longer economically viable or is fully consumed, as in some polymerization catalysts [84]. Like **TOF**, full specification includes temperature, pressure, and reactant concentrations.**Selectivity to a desired product** (**S_des_pr_**)—The number of moles of the desired product relative to the theoretical maximum according to stoichiometry, at a given conversion of a key reactant.**Catalyst lifetime (CLT)**—The total time the catalyst remains active in the reactor (typically in days or years), from its introduction until deactivation to an unacceptable level.**Catalyst life cycle (CLC)**—The full period covering catalyst use, deactivation, regeneration, and eventual replacement. It is also expressed in days or years.

To determine these parameters, a well-defined set of experiments must be conducted. In gas–solid (G–S) or liquid–solid (L–S) reaction systems involving porous catalysts, the following nine transport and reaction steps should be considered:0.Transport of reactants from the bulk phase to the vicinity of the catalyst (often neglected due to intensive mixing);1.Transport of reactants through the external surface layer surrounding the catalyst;2.Diffusion of reactants into the catalyst pores toward active sites;3.Chemisorption of at least one reactant onto the active sites;4.Surface catalytic reaction;5.Desorption of products from the catalyst surface;6.Diffusion of products out of the catalyst pores;7.Transport of products through the external surface layer;8.Transport of products into the bulk phase (often neglected due to intensive mixing).

In gas–liquid–solid (G–L–S) systems (e.g., hydrogenation, oxidation, or alkylation of liquid substrates), interfacial mass transfer between gas and liquid phases (for reactants and possibly for products) must also be considered, as they significantly influence overall catalytic performance.

In kinetic experiments, it is assumed that mass transport limitations are negligible; that is, the rate of mass transfer greatly exceeds the rate of the chemical reaction, which is the rate-determining step. Satterfield [56] provides detailed criteria and conditions to identify this regime. Two useful guidelines are as follows:**Apparent Activation Energy (E_a_)** as an indicator:
(a)If transport limitations dominate, E_a_ ≈ 4–12 kJ·mol^−1^.(b)For catalytic reactions, E_a_ typically ranges from 30 to 120 kJ·mol^−1^.(c)Non-catalyzed reactions tend to have E_a_ > 120 kJ·mol^−1^.
This implies that performing experiments at different temperatures and calculating E_a_ can help identify whether the regime is kinetic or transport-limited. However, caution is needed; at high temperatures, the reaction rate may exceed the transport rate, resulting in a drop in apparent activation energy, e.g., hexene combustion over platinum gauze at >400 °C [56].
**Effect of Particle Size**:Transport limitations become more significant with larger catalyst particles (>1 mm). If active sites are uniformly distributed, decreasing particle size reduces diffusion limitations until a threshold is reached, beyond which mass-specific activity remains constant.

Minimizing external mass transport is typically achieved by intensive mixing. In gas-phase systems, gradientless reactors (where reactant concentration at the inlet equals that at the catalyst bed outlet) are used (Figure 1). In liquid-phase systems, similar reactors can be used, along with mechanically stirred suspension reactors, externally circulated reactors, or vertical tubular reactors, depending on the process requirements.

#### 2.6.1. Kinetic Regime

The advantage of gradientless reactor is in a simple estimation of reaction rate (${\dot{\xi }}_{w}$, mol s^−1^ kg_cat_^−1^) with respect to concentration of reactants [56,85]:(1)$${\dot{\xi }}_{w}=\frac{{\dot{n}}_{i,0}-{\dot{n}}_{i,E}}{w_{cat}}$$

${\dot{n}}_{i,0},\;{\dot{n}}_{i,E}$—molar flow of the key component “i”, in the input and output and output of the reactor (mol s^−1^)

$w_{cat}$—amount of the catalyst (kg)

For comparison, expression for reaction rate (${\dot{\xi }}_{w,tj}$, mol s^−1^ kg_cat_^−1^) in the batch reactor is:(2)$${\dot{\xi }}_{w,tj}=\frac{dc_{tj}}{dt}\frac{V_{R}}{w_{cat}}$$

*c*_i,tj_—concentration of the component “i” in the time point *t*_j_ (mol m^−3^);

*t*—time (s);

*V_R_*—volume of the reactor (m^3^).

It is evident that under constant operating conditions (temperature, pressure, flow rate, and input concentration) in a continuous reactor, the reaction rate in a gradientless reactor remains constant—provided the catalyst does not undergo deactivation. In contrast, in a batch reactor, the reaction rate typically changes over time, usually decreasing (except in the case of autocatalytic reactions). As a result, data analysis from gradientless reactors is generally simpler than from batch reactors. However, modern computational tools enable effective processing of batch reactor data (combination of true reaction kinetics, mass, and heat transport phenomena), and the accuracy of the derived kinetic parameters largely depends on the quality of the experimental measurements.

**Figure 1 molecules-30-03279-f001:**
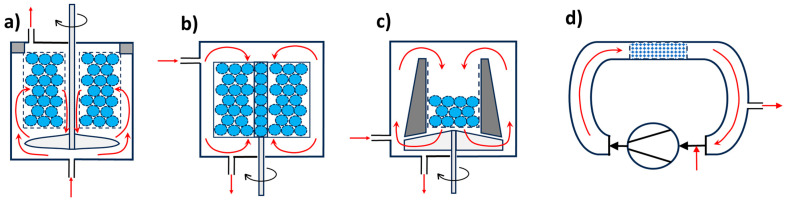
Charts of gradientless flow reactors. (**a**) Notre Dame—catalyst is located in a fixed-bed cylinder basket, circulation of the reaction mixture is via the axial–radial impeller; (**b**) catalyst is located in the frame basket fixed to a shaft and spinning (Carbery type reactor); (**c**) catalyst is in fixed bed basket, and circulation is by axial–radial impeller (Berty reactor); (**d**) catalyst is in a tube reactor, and mixing is maintained by recirculation using an external pump/fun. Red arrows indicate movement of reaction components. (Charts are derived from pictures published in [56,86]).

As reported in [85], a modern and comprehensive approach to catalytic research includes consideration of mass and heat transport phenomena. Efficient catalyst evaluation is increasingly conducted using high-throughput experimentation (HTE) and microchannel reactors—systems comprising multiple parallel reactors that can operate with different catalysts, catalyst amounts (affecting residence time), and under varying conditions (particularly temperature). Analytical techniques such as GC, GC-MS, or HPLC are typically handled by one or a few instruments, which are periodically connected to the reactor outputs for sampling and analysis [87].

Two types of reaction rate models are used for catalytic reaction, as follows:Statistical, experimental, usually in the form of power-law equations.Mechanistic, considering steps of chemical transformation. Commonly, three basic types of chemical reactions are considered:
a.Reactants are chemisorbed on catalytic centers (Langmuir–Houghen–Hinshelwood–Watson models—LHHW);b.One reactant is chemisorbed, and another one comes as not-chemisorbed (Elley– Rideal models—ER);c.One of the reactants (e.g., oxygen) is exchanging positions in the crystal framework of the catalysts (Mars van Krevelen models—MarKre).

Power law models have the form:(3)$${\dot{\xi }}_{w,T}=k_{RT}w\prod_{i=1}^{N_{C}}a_{i}^{b_{i}}$$

*k*_RT_—chemical reaction rate constant at temperature “T” (units depend on the form of kinetic equation);

*N*_C_—number of components;

*a*_i_—activity of the component “i” (mol m^−3^, or Pa);

*b*_i_—exponent in the power-law kinetic model.

If a component “i” does not enter into the reaction, the exponent *b*_j_ is equal to zero.

A rather universal reaction model for LHHW and ER mechanisms considering a reaction at the same type of catalytic centers:(4)$${\dot{\xi }}_{w,T}=k_{R,T}w\frac{\prod_{i=1}^{N_{C}}{\left(K_{i,ads}a_{i}\right)}^{b_{i}}}{{\left(1+\sum_{i=1}^{N_{C}}{\left(K_{i,ads}a_{i}\right)}^{d_{i}}\right)}^{e}}$$

$K_{i,ads}$—Adsorption constant for the component “i”;

*b*_i_, *d*_i_, *e*—coefficients.

In purely mechanistic models, the exponents *b*_i_ and *d*_i_ are equal. For example, a square root dependence typically indicates the dissociation of a component [56,61,86]. However, in real systems where additional effects are present—such as mass transport limitations or interactions between occupied and unoccupied catalytic sites—the coefficients *b*_i_ and *c*_i_ may differ. The exponent *d* generally represents the number of catalytic sites involved in the surface reaction. For instance, if two chemisorbed species participate in the rate-determining step, *d* equals to 2. For more complex models involving, for example, two types of catalytic centers, refer to [71].

#### 2.6.2. Mass and Heat Transport

Figure 2a illustrates the effect of mass transport hindrance on the performance of microporous catalysts. It is evident that purely microporous catalysts (D < 2 nm), such as zeolites, are generally not well-suited for achieving high catalyst productivity. To allow reactants to access the entire internal structure of such catalysts, the particle size must be extremely small (typically < 1 μm). Otherwise, a significant portion of the catalyst volume remains unused. While these small catalyst particles are suitable for liquid-phase reaction systems, they impose greater demands on separation processes—particularly filtration [88,89]. Large-scale industrial applications of liquid-phase systems using pure (sub)micron-sized zeolite particles are virtually unknown. An alternative approach is to incorporate small catalytic particles into a larger composite catalyst structure—the most well-known example being fluid catalytic cracking (FCC) catalysts. Much better catalytic performance is typically achieved using structured-texture catalysts, which combine micro-, meso- (2 nm < D < 50 nm), and macropores (D > 50 nm) (see Figure 2b) [8,10,79]. This hierarchical pore structure enhances diffusion and accessibility of active sites across a wide range of reactant sizes. Figure 2c shows the dependence of the diffusion coefficient of a low-molecular-weight gas on the pore size (either the diameter *D* of a cylindrical pore or the width of a slit). One-dimensional mass transport in a plate-like material can be described by Fick’s law.(5)$${\dot{J}}_{i,x}=-D_{i,x}\frac{{\delta a}_{i,x}}{\delta x}$$

${\dot{J}}_{i,x}$—molar flow of the component “i” in the direction “x” (mol m^−2^ s^−1^);

$D_{i,x}$—diffusion coefficient for the component “i” in the direction “*x*” (m^2^ s^−1^);

*a*_i,x_—activity of the component “i” in the place “x” (mol m^−3^, or Pa).

For a system with chemical reactions at a steady state regime:(6)$$\frac{\partial {\dot{J}}_{i,x}}{\partial x}=\sum_{j=1}^{N_{R}}{\dot{\xi }}_{R,j}\nu_{j,i}$$

*N*_R_—number of reactions;

${\dot{\xi }}_{R,j}$—rate of the reaction “j” (mol m^−3^ s^−1^);

*ν*_j,i_—stoichiometric coefficient for the component “i” in the reaction “j”.

Heat transport in a plate material is expressed by a similar relationship (Fourier’s law):(7)$${\dot{Q}}_{x}={-k}_{Q,x}\frac{{\delta T}_{,x}}{\delta x}$$

${\dot{Q}}_{x}$—heat flow in the direction “x” (J m^−2^ s^−1^);

*k*_Q,x_—thermal conductivity in the place “x” (J K^−1^ m^−1^).

Heat flow for a system with chemical reactions at a steady state regime:(8)$$\frac{{\partial \dot{Q}}_{x}}{\partial x}=\frac{\sum_{j=1}^{N_{R}}{\dot{\xi }}_{R,j}(-\Delta_{R}H_{j})}{c_{P}\rho }$$

Δ_R_*H*_j_—molar reaction enthalpy for the reaction “j” (J mol^−1^);

*c*_P_—specific heat (J K^−1^ kg^−1^);

*ρ*—density of the catalyst including void spaces (kg m^−3^).

It is worthy to add that the combination of Equations (5) with (6) and (7) with (8) results in second-order partial differential equations that must be solved numerically. The initial and boundary conditions for Equations (5)–(8) and their combinations can be found in the catalytic reaction engineering literature, see, e.g., [7,60,61,62]. These references also provide formulations for various catalyst geometries (e.g., spheres, cylinders) as well as for non-steady-state regimes.

According to Equation (5), a higher flow rate is achieved with a higher activity gradient. However, due to the typically low value of *D*_i,x_, mass transport in small pores remains very limited (Figure 2c). Figure 2a also illustrates how to save expensive catalytic materials—it is ineffective to load catalytic moieties in regions inaccessible to reactants. This highlights the importance of supported structured catalysts.

When a combination of mass transport limitations and intrinsic reaction kinetics is present, the resulting behavior is referred to as apparent reaction kinetics. In such cases, activation energies fall between the values typical of purely diffusion-limited and purely kinetically controlled regimes. Under these conditions, mechanistic models (LHHW, ER, MarKre) are generally not meaningful; instead, power-law models are preferred. However, it is essential to define the experimental conditions (e.g., concentrations, pressure, temperature, and residence time) under which the model is valid.

An interesting study [90], which investigated the varying activity of different surface sites (edges and terraces) on a gold catalyst, emphasized the role of surface energy and Gibbs free energy in determining catalytic activity. For nanocatalysts, specialized kinetic equations have been developed [91,92,93], and more detailed mechanistic models have been proposed [94].

So far, catalyst deactivation—the most important factor for estimating the catalyst life cycle (CLC)—has not been addressed. This topic will be discussed in the next section of this contribution.

## 3. Heterogeneous Catalysts

Figure 3 depicts a few industrial supports and catalysts, as well as some typical processes they are used for.

According to their method of preparation [95] and the placement of catalytic centers, heterogeneous catalysts are classified as follows:Unsupported catalysts—Catalytic centers are relatively uniformly distributed throughout the entire catalyst volume. All components are typically mixed before the final formulation of the catalyst.Supported catalysts—Catalytic centers are located on the surface of a support, including within micropores. This includes surface functionalization of a pre-formed support and/or the deposition or generation of metal nanoparticles on the support.

### 3.1. Unsupported Catalysts

Metals, metal oxides, sulfides, zeolites, hydrotalcites, polymers synthesized from functional monomers, and other catalytically active species represent the main types of unsupported catalysts. These are also commonly referred to as bulk catalysts.

A special subgroup of unsupported catalysts is known as skeletal catalysts, which are characterized by a relatively regular pore structure. The most well-known examples are Raney-type catalysts—typically based on nickel, cobalt, copper, or their alloys with aluminum [96]. The skeletal structure forms either during the pretreatment of the alloy or as part of the catalyst activation process (Figure 4). For instance, relatively smooth Al-Ni alloy particles (Figure 4a) develop a highly porous, rugged morphology after treatment with aqueous NaOH and NaBH_4_ (Figure 4b), which significantly enhances their catalytic activity in hydrogen-dehalogenation reactions [97]. The morphological evolution of Pt-Pd-Ru-Rh gauze wires used in the oxidation of ammonia to nitrogen oxides (in nitric acid production) is shown in Figure 4c,d [98]. More recently, gold-based skeletal catalysts [99] have enabled various chemoselective reactions, such as the oxidation of aldehydes to acids and [4 + 2] benzannulation of *ortho*-alkynylbenzaldehydes. Their high activity and product selectivity are attributed to defects in the crystal structure of gold (e.g., nanoporous structures, nanotubes, powders) and their relatively high specific surface area (5–30 m^2^·g^−1^).

Although less common, zeolites, covalent organic frameworks (COFs), and metal–organic frameworks (MOFs) can also be classified as skeletal catalysts [49]. These materials possess a regular pore structure formed during their synthesis. Surface modification, such as additional functionalization, is often applied to further tune or enhance their catalytic properties. Selected examples of unsupported catalysts are listed in Table 1.

The methodology for the preparation of unsupported catalysts is illustrated in Figure 5. A typical example of a metal catalyst is Raney Nickel (see Figure 4), which is prepared by melting nickel and aluminum, followed by the “extraction” process—dissolution of aluminum using sodium hydroxide. A well-developed pore structure is achieved (see Figure 4b).

Nitrates are commonly used compounds for the preparation of precipitated catalysts, while gaseous ammonia or its aqueous solution is employed to form solid oxyhydroxides. These are subsequently dried, calcined, crushed, and mixed with binders and modifiers. After mixing, the material is shaped into the desired form (e.g., granules, tablets, or extrudates), then dried, optionally calcined, and screened to obtain catalyst particles of suitable quality. Hydrogen is mostly used as an activating (reducing) agent.

An example of an unsupported precipitated catalyst is Fe_2_O_3_-Cr_2_O_3_-K_2_CO_3_ (Figure 3G). The precipitation method requires large amounts of washing water, and the resulting wastewater must be carefully treated to remove inorganic components (mostly heavy metals) to minimize environmental impact.

### 3.2. Supported Catalysts

The following reasons favor the use of supported catalysts:Preparation of relatively large catalyst particles (up to 20 mm), which helps minimize resistance to fluid flow (gas and/or liquid).Increase in mechanical strength compared to unsupported catalysts.Reduced amounts of catalytically active components, particularly expensive metals and their compounds.Favorable interactions with the support, including reactant–support, product–support, solvent–support, catalytic moiety–support, e.g., metal (nano)particles, which influence chemisorption of both reactants and products.Stabilization of catalytic moieties on the surface of support (e.g., minimization of sintration by embedding particles in pores of a support, or by interactions with functional groups of the support).Enhanced control over reaction temperature and contact time by adjusting the amount and distribution of catalytic moieties (e.g., Pd on alumina for selective hydrogenation of acetylene in ethylene production).

Catalyst supports are generally classified into the following three main groups [79]:Inorganic (e.g., elemental carbon or metals, oxides, carbonates);Organic (various types of polymers);Hybrid (inorganic–organic composites).

Table 2 presents examples of commonly used catalyst carriers. In some cases (e.g., hydrotalcites, clays, zeolites), these materials can act directly as catalysts. The listed supports vary significantly in specific surface area and porosity, which influences their ability to anchor catalytic centers (functional groups or metal/oxide nanoparticles) and thus affect overall catalytic activity. Catalytic performance strongly depends on the accessibility of catalytic sites, which is a function of the size of reactant and product molecules relative to the pore dimensions. Highly porous materials typically exhibit lower mechanical strength, making them unsuitable for direct use in large catalyst bodies (e.g., in fixed-bed reactors). Therefore, composite catalysts are often required. In contrast, mechanically robust but non-porous materials (with low specific surface area) are modified to improve porosity. This can be achieved, for example, by acid etching (e.g., of cordierite) [101], or by deposition of a porous layer [102] (used in the manufacture of flue gas abatement catalysts [79]). Electrical conductivity of the support indicates its potential for electronic interactions with the catalytic centers, while thermal conductivity is crucial for minimizing temperature gradients between the catalyst surface and interior. A key property is the maximum operating temperature (T_op,max_), which indicates the temperature beyond which structural collapse may occur. It should be noted that when catalytic moieties are deposited on a support surface, the actual safe operating temperature is usually significantly lower than (T_op,max_).

Organic and metal–organic polymers (OMOPs) as catalyst supports and catalysts have been extensively discussed in our recent paper [49]. These materials include the following:Natural polymers (e.g., celluloses, lignocellulose, chitin, starch);Synthetic polymers, primarily those containing heteroatoms (O, N, P, S, etc.);Covalent Organic Frameworks (COFs) and Metal–Organic Frameworks (MOFs)

Generally, OMOP-based catalysts have low thermal conductivity. Their electrical conductivity varies depending on the polymer structure, particularly those allowing electron and/or hole transport. One example is polyaniline, known for its *π*-conjugated electron system. Conductivity is also influenced by dopants. The porosity and accessibility of the polymer’s interior depend on interactions with the reaction medium. In highly compatible liquid environments, the “void volume” (volume not occupied by the support/catalyst material) can reach up to 10 cm^3^ g^−1^. A major limitation of OMOP-based catalysts is their relatively low thermal stability, typically ranging from 80 to 200 °C [49,114,115]. Nonetheless, more thermally stable polymers—such as grafted polyacrylonitrile or polyether ether ketone (PEEK) in fiber form—are useful in organic synthesis, including photocatalytic applications [116].

Interactions between the support material and both the catalytic species and the reaction environment play a critical role. These interactions can be classified into the following two main types:Interactions with Atoms and Surface Defects of the Support Framework: This type is especially relevant for inorganic semiconductor-type supports, and has been recognized since the early systematic studies of catalysis (pre-1960) [8,10]. The electronic nature of atoms in the support—whether electron-donating or electron-withdrawing—significantly influences catalytic performance. One way to evaluate these effects is by comparing electronegativities (χ). For example, effect of gold (χ = 2.54) as electron attracting atom with respect to Fe (χ = 1.83) exhibited positive effect on lowering the activation energy (*E*_A_) in oxidation of CO over the Fe_2_O_3_/Al_2_O_3_ catalyst, and manganese (χ = 1.55) as electron repelling with respect to Fe increased *E*_A_ [117]. A peculiar position belongs to cerium (χ = 1.12) mainly as CeO_2_. Due to low electronegativity, a strong electron repelling effect helps in reactions where activation starts with the addition of an electron to a reactant, e.g., in oxidations, the best-known examples are convertors for oxidation abatement of flue gases [79]. Low electronegativities are also exhibited by lantanoids (χ = 1.1–1.3); however, industrial applications of these metals and their compounds as promoters are limited due to their relatively high prices. Furthermore, metal particles supported on oxides or other substrates can exhibit Strong Metal–Support Interactions (SMSI), a phenomenon studied extensively since the 1970s [118]. Notable support materials involved in SMSI include ZnO, TiO_2_, CeO_2_, MgO, and nitrides, especially in combination with transition non-noble or noble metals. These interactions are now well characterized using advanced analytical techniques (see Figure 6).Interactions with Functional Groups: Functional groups on the support—especially those containing lone-pair donating atoms like nitrogen (e.g., –NH_2_, –NH–CO– in polyamides, polyurethanes, polyureas, and porphyrins) or phosphorus—can significantly influence the behavior of supported catalysts [49]. In the case of metal particles, these can form Covalent Metal–Support Interactions (CMSI), enhancing stability and tuning reactivity.

When comparing organic and metal–organic polymer (OMOP) supports with inorganic counterparts, it is important to emphasize the strong interactions between the reaction environment and the texture of OMOPs. These effects are particularly pronounced in liquid-phase systems, but they also occur in vapor–gas reactions due to capillary condensation. Such interactions cause swelling and “breathing” of the catalyst structure, which enhances the accessibility of catalytic centers within the material. In a swollen OMOP matrix, the transport of reactants resembles diffusion through a viscous medium rather than the “pushing” mechanism through rigid pores of inorganic catalysts. As a result, the mass transport of reactants in OMOPs can be faster compared to an inorganic catalyst with a similar void volume. However, the advantage of enhanced accessibility is partially offset by lower mechanical and chemical stability, which reduces the lifetime of OMOP-based catalysts.

The production of supported catalysts (Figure 7 and Figure 8) is generally more complex than that of unsupported catalysts. Figure 7 focuses primarily on inorganic catalysts [6,95], while Figure 8 illustrates OMOP-based catalysts [49]. Routes A–E represent various treatments of a support with deposited metal or metal compounds.

In the case of metal particle deposition, the support may be directly converted into a catalyst (Route A). In some cases, additional activation may be necessary, for example, applying a sulfur compound to decrease activity and increase selectivity (Route B). A support with a deposited metal compound can be dried and calcined to obtain active catalytic sites (Route C). Route D is similar to Route C. Route E involves removing undesired compounds (washing), followed by drying/calcination and activation of the catalytic centers.

The scheme in Figure 7 is also applicable to structured catalysts.

The preparation of OMOP-based catalysts also includes the synthesis of support from non-functional or functional monomers, as well as from COFs and MOFs.

Support materials are typically in the form of spheres, cylinders, star-shaped particles, or skeletons [78,95]. After this, catalytic moieties or their precursors are introduced. The main steps for preparing supports and unsupported catalysts include precipitation, solid–liquid separation, washing, drying, calcination, milling, and formulation into catalyst particles. There are two main categories of procedures used for introducing catalytic species onto supports.

Chemical modification of a support:
Oxidation—to introduce functional groups such as carboxyl, keto, aldehyde, and hydroxyl;Amination—addition of amine, substituted amine, or tetraalkylamine groups;Sulfonation—formation of –SO_3_H groups;Covalent binding of organocatalysts—e.g., nitrogen- or phosphorus-based moieties, metal–organic catalytic complexes;Immobilization of enzymes or enzyme-like catalytic moieties.


These methods are used to prepare heterogenized catalysts—homogeneous catalysts made heterogeneous by anchoring onto supports.

Introduction of metal (nano)particles:
Fixation of pre-synthesized nanoparticles;Adsorption of metal precursors followed by decomposition or reduction, through techniques such as:
Wetting processes;Adsorption, ion-exchange procedure;Chemical vapor deposition;Magnetron sputtering [119,120], where metal from a bulk source is vaporized (via heat or electron beam) and directed using a magnetic field toward the support; addition of other compounds enables formation of oxides, nitrides, carbides, etc.;Electrochemical deposition, especially effective for single-atom catalysts (SACs).

For most supported catalysts, activation is necessary. This can involve calcination, regeneration of oxidized catalytic surfaces, or reduction of metal oxides to metallic particles. Industrial-scale reductions are typically performed using hydrogen or CO. Laboratory-scale reductions may use NaBH_4_, amines, alcohols, sugars, or aldehydes. When metal precursors are introduced in the form of inorganic salts (e.g., amino-palladium nitrate) or metal–organic compounds (e.g., palladium acetate), the reduction pathway can lead to redistribution of metal within the catalyst particle compared to the initial distribution [121,122]. Notably, activation/reduction of industrial catalysts is often carried out directly in the reactor.

### 3.3. Composite Catalysts

Broadly defined, composite catalysts are those consisting of two or more components. More specifically, they include systems where catalyst particles incorporate distinct (nano)particles and often possess additional functionalities, such as membranes that protect catalytic nanoparticles. Sometimes referred to as assembled catalysts, these systems are built by organizing known catalysts or materials in a controlled manner [123]. Microporous materials such as zeolites, COFs, and MOFs are commonly employed in their design [79].

Composite catalysts with metal nanoparticles combine the benefits of supported catalysts (see interactions illustrated in Figure 6). Examples include catalysts used in the Fluid Catalytic Cracking (FCC) process, containing zeolite, alumina, and clay, or three-way catalysts for automotive flue gas treatment.

### 3.4. Promoters

Promoters, also referred to as modifiers, are additives introduced into a catalyst or directly into the reaction system to enhance catalyst performance in terms of activity (**TOF**, **CatProd**), selectivity (**S_des_pr_**), and stability (**CLT**) [124,125]. By definition, promoters exhibit little to no intrinsic catalytic activity on their own. If a component significantly contributes to catalytic activity, it is more appropriately classified as a co-catalyst. Promoters have played a crucial role since the early development of industrial catalysis [10,126]. A classic example includes the addition of alkali metals to iron-based catalysts in the synthesis of ammonia, which modifies electronic properties and improves performance [1,6,127]. Another example is the introduction of chlorine-containing compounds into the feed stream during the epoxidation of ethylene, where they enhance selectivity and stability of the silver catalyst [1,128,129]. Promoters can be broadly categorized into two main types [124].

Promoters Permanently Incorporated into the Catalyst Structure.

These modifiers are embedded into the catalyst and influence performance through various mechanisms.
Structural Promoters: Prevent undesirable structural changes, such as the sintering of metal nanoparticles, thereby preserving active surface area.Electronic Promoters: Modify electron density at active sites. For instance, alkali metals increase the electron density on iron catalyst surfaces, enhancing nitrogen adsorption and activation in ammonia synthesis [6].Surface modifiers: Influence surface chemistry and physical properties. Examples include sulfur-induced poisoning of certain sites on Fischer–Tropsch catalysts to improve olefin selectivity [130], conversion of hydrophilic to hydrophobic surfaces via tethering of polytetrafluoroethylene oligomers [131], ionic liquids [132], or anchoring of stereospecific ligands (heterogenized) to improve enantioselectivity. For example, over 90% selectivity to (R)-1-phenyl-1-propanol was achieved using PS-supported hydroxyamides in the enantioselective ethylation of benzaldehyde [133].

2.Promoters Introduced into the Reaction Medium: These act dynamically, affecting surface interactions and reaction pathways. They may compete for adsorption sites, modify the reaction mechanism, or create transient intermediates. For instance, chlorine-containing species formed in situ during ethylene epoxidation facilitate oxygen activation over silver catalysts [129], and water vapor used to suppress carbon deposition in the dehydrogenation of ethylbenzene to styrene over iron catalysts [134]. Numerous additional cases highlight the use of in situ promoters to alter surface behavior and enhance catalyst performance [79].

### 3.5. Structured Catalysts

Structured catalysts (**StrCat**) are defined as highly ordered arrangements of heterogeneous catalyst components designed to ensure uniform and reproducible flow directions [123]. These catalysts aim to achieve uniform residence time distributions, homogenized concentration and temperature profiles, and the elimination of various types of maldistributions. The primary objective of designing and using StrCats is to maximize catalyst activity, selectivity, and lifetime. In a broad sense, all shaped or formulated catalysts—such as extrudates with channels, asterisk-shaped bodies, sponges, solid foams, gauzes, or various types of monolithic packings—are considered structured catalysts. In a narrower context, the term refers specifically to membranes, monoliths, and hierarchically structured materials, particularly zeolites [88,135,136].

Structured catalysts can be categorized into three major groups.

Structures formed as a result of standard preparation processes:
**Inorganic**, e.g., *γ*-alumina [95];**Organic**, e.g., Hypercrosslinked polymers [49,137,138].Tailored structured catalysts [139,140]:
Fibers and meshes:
**Inorganic**: Examples include platinum alloy meshes used in ammonia oxidation [6], other metal meshes [141,142], carbon-based meshes [143,144], and glass-based fibers [113,145];**Organic**: Polymer-based fibers used for specific catalytic tasks [116,146,147].Skeletal catalysts:
Corrugated supports prepared from inorganic or organic sheets:
**Inorganic**: Materials such as stainless steel or metal alloys [148,149] (examples in Figure 9) offer high thermal conductivity and stability (up to 1200 °C) but typically have low surface areas. Microporosity can be introduced via anchoring of catalytic nanoparticles [79,150].**Organic** [151,152,153]: These offer lower cost and inherent microporosity but suffer from limited thermal (max ~200 °C) and chemical stability (e.g., cellulose solubility in alkaline solutions).Extruded and 3D printed:
**Inorganic:** Skeletons are typically made of silica, spinels (e.g., MgAl_2_O_4_), or cordierite ((Mg,Fe)_2_[Si_5_Al_4_O_18_]·nH_2_O), using extrusion. Additives like aluminum hydroxide or zinc salts improve plasticity and structural stability [96]. After calcination, monoliths can withstand up to 1200 °C. Porosity is improved via a washcoat layer (e.g., Al_2_O_3_) with anchored catalytic nanoparticles [148]. For more complex architectures, 3D printing is employed, enabling vertical and horizontal channel formation [154]. Additive Manufacturing (AM 3D) methods can include zeolites, MOFs, COFs, or burnable organic additives to increase porosity during calcination.**Organic:** Skeletons are typically fabricated from thermoplastics like polylactic acid, polypropylene, polyamides, or ABS [155]. Advanced techniques involve photopolymerization initiated by light [156]. Printing resolutions can reach 25 µm [157].**Hierarchical catalysts** (HiCat): Figure 2b refers to the multi-level porous architecture (micro-, meso-, and macropores) that facilitates enhanced diffusion and accessibility to active sites. These catalysts are particularly relevant for zeolites [123,140,158,159,160]. Conventional zeolites are microporous (0.4–0.8 nm). Post-treatment (e.g., desilication with alkaline hydroxides) introduces mesoporosity, improving performance by reducing transport limitations. Hierarchical structuring can also be achieved using bulky templates (alcohols, amines), polymer nanoparticles, or carbon black, followed by calcination. In this context, the term hierarchy factor (*HF*) is important; *HF* = (*V*_microp_/*V*_p_) × (*S*_mesop_/*S*_BET_) with *V* = volume, *S* = surface. Typical HiCats of zeolite type exhibit *HF*s from 0.15 to 0.2. HiCats’ zeolite type extended potential for their application [160,161,162]. In addition to zeolites, some other HiCats were prepared and tested [139,163,164,165]. OMOPs, including MOFs and COFs StrCats with regular texture, are also considered as HiCats [166].**Encapsulated catalysts** (EnCat): Encapsulation of catalytic active moieties (enzymes, metal–organic complexes, metal nanoparticles) is an effective method for protecting catalysts from deactivation. The active species are enclosed within a capsule with permeable walls that allow the passage of reactants and products. Encapsulation prevents the formation of agglomerates (e.g., sintering in the case of inorganic materials), while diffusion through the surrounding layer can regulate selectivity. Various techniques for preparing EnCats, including the encapsulation of magnetic particles for easy catalyst separation, are summarized in [167]. EnCats are also used industrially, for example, palladium encapsulated in polyureas [168,169]. EnCats based on polymeric organic films are typically suited for mild reaction conditions, with operational temperatures not exceeding 150 °C [167,170,171,172]. When enzymes are encapsulated, the working temperature is even lower (around 75 °C), although this is still higher than the operational limit for free enzymes [173]. Catalytic species can also be encapsulated within porous inorganic materials, such as zeolites, which allow much higher working temperatures—up to 400 °C [174,175]. In this context, it is worth comparing encapsulated catalysts with metal catalysts anchored to supports containing strong chelating groups (e.g., nitrogen-based moieties). Recent studies have demonstrated that palladium supported on crosslinked polyureas exhibits higher stability than encapsulated variants [122]. This finding highlights the importance of considering both approaches during catalyst design, with attention to both reaction rate and catalyst lifetime.

### 3.6. Electrocatalysts

The oldest and most thoroughly studied electrocatalytic process is the electrolysis of water [176]. Despite its long history dating back to 1800, water electrolysis remains a subject of growing interest—particularly for hydrogen production as an energy carrier using electricity from renewable sources such as wind, hydro, and photovoltaics [177,178,179]. Moreover, the theoretical foundations and many experimental insights from water electrolysis are applicable to other electrocatalytic processes. Figure 10 illustrates the influence of the metal type on the exchange current density (j_0_). The well-known “volcano plot” behavior of the hydrogen oxidation reaction shows that the highest exchange current density occurs at an intermediate metal–hydrogen (M–H) bond strength—a classic principle in catalysis. Noble metals such as Pt, Ir, and Rh clearly demonstrate superior performance, while nickel stands out as the best among non-noble metals. It is worth noting that pure palladium is not suitable due to its high hydrogen solubility, which leads to degradation of performance. The values shown in Figure 10 were obtained for pure metals in acidic solutions [180], but similar qualitative trends can also be expected in other environments.

Electrocatalytic processes are assessed by two basic criteria:Faradaic efficiency:(9)$$\eta_{FE}(\%)=100\frac{n_{Prod,t}n_{el,mol}F}{\int_{0}^{t}I_{t}dt}$$

*n_Prod,t_*—number of moles produced in the time interval 0–t;

*n_el,mol_*—number of electrons needed for production of 1 mol product by electrochemical process;

*F*—Farraday constant equal to 96,485.3321 s A mol^−1^, or C mol^−1^, where C denotes the Coulomb unit (A s);

*I_t_*—a value of electric current (A) in the time t;

*A*—Amper.

A correct estimation of *η*_FE_ is rather cumbersome. Even with careful preparation of the electrochemical cell, due to the shortness of knowledge of the chemical reactions occurring at the interface and competing reactions (corrosion processes, product crossover, etc.), estimation is complicated. Consequently, erroneous values are often obtained [181]. In the case of water electrolysis, the value of *η*_FE_ is often close to 100%.

Energetic efficiency


(10)
$$\eta_{EE}(\%)=100\frac{\sum_{i=1}^{N_{P}}n_{i,t}H_{i,T}-\sum_{j=1}^{N_{R}}n_{j,t}H_{j,T}}{\int_{0}^{t}I_{t}U_{t}dt}$$


*n_i,t_* and *n_j,t_*—number of moles of products and reactants, generated/consumed in the time interval 0–t, respectively;

*H_i,T_* and *H_j,T_*—enthalpy of products and reactants at temperature T (K), respectively (J mol^−1^);

*N_P_* and *N_R_*—number of components and reactants leaving/entering the process, respectively;

*U_t_*—voltage of electric current (V); the product *I_t_U_t_dt* (A V s) gives energy in Ws, which is equal to 1 J.

In the case of water electrolysis, the theoretical energy requirement is 237 kJ mol^−1^, which is derived from free energy changes (voltage 1.23 V—minimum theoretical value) [182,183].

Due to overpotential and ohmic resistance of electrolyzer compartments [183], a practical voltage is higher than theoretical. Typical values for water electrolyzers are in the range from 1.5 to 2 V, which implies at *η_EF_* close to 100%, *η_EE_* is equal to 82% (1.23/1.5*100) and 62% (1.23/2*100), respectively. Energy not used for electrochemical transformation of reactants to products is dissipated as heat, which needs to be removed from the electrolyzer by cooling. The effect of current density on the overpotential is characterized by a Tafel slope, i.e., the slope of the curve overpotential vs. logarithm of current density [176]. A higher value typically indicates a less suitable electrode, reflecting higher ohmic and activation potentials, as well as a strong influence of the type of electrolyte and mass transport on the electrolyzer’s performance.

A more comprehensive evaluation of energy efficiency must also account for the energy consumed by auxiliary equipment such as pumps, compressors, and mixers [184,185]. In terms of reducing overpotential, various catalysts play a significant role; for example, composites and alloys such as Pt-Bi, Pt-Cu, and Pt-Co; composites with MOFs and COFs [186]; and single-atom catalysts such as copper embedded in graphene [187]. Electrode coatings using materials like Al_2_O_3_, SiO_2_, TiO_2_, SnO_2_, and HfO_2_ can also be considered a form of catalysis, as they help reduce overpotential and, in some cases, inhibit corrosion [188]. Overall, the development of advanced electrodes, membranes, and the design of electrochemical compartments remains an active area of research. Techniques commonly used in classical catalyses, such as Density Functional Theory (DFT) and work function calculations for modeling adsorption/desorption of reactants and products, are now also applied to optimize mass and charge transport in the design of novel electrocatalytic systems [189,190,191].

### 3.7. Photocatalysts

The classification of catalysts as homogeneous or heterogeneous also applies to photocatalytic systems. In photocatalysis, the absorption of photon energy increases the reactivity of molecules, enabling them to participate more readily in chemical reactions. In homogeneous systems, photon absorption can directly activate the chemical bonds of reactants (e.g., through polarization or, in extreme cases, radical formation), or it may occur via a homogeneous photocatalyst that serves as an energy carrier [192]. An important aspect of photocatalysis is the relationship between bond dissociation enthalpies and the corresponding electromagnetic radiation energy, which is wavelength-dependent (λ, in nm). For example: (bond, dissociation energy—kJ/mol, λ—nm): H−H: 436, 274; C−C: 347, 345; O=O: 498, 240; N≡N: 941, 127; C−H: 413, 290; C−F: 552, 216) [192]. Because most chemical bonds are relatively strong, photocatalysts are often required to facilitate these reactions. Effective homogeneous photocatalysts include metal complexes such as tris(bipyridine)ruthenium(II) chloride (λ_abs,max_ = 455 nm, in the visible region) and [Ir(dF(CF_3_)ppy)_2_(dtbpy)]PF_6_ ((4,4′-Di-t-butyl-2,2′-bipyridine)bis[3,5-difluoro-2-[5-trifluoromethyl-2-pyridinyl-kN)phenyl-kC]iridium(III) hexafluorophosphate, λ_abs,max_ = 380 nm, near-UV region).

In heterogeneous photocatalysis, semiconductors play a central role. Chemisorbed reactants are activated through the transfer of electrons and holes generated by light absorption (Figure 11, left). The ability of a semiconductor to absorb photon energy depends on its band gap: larger band gaps require higher-energy photons (e.g., UV light), while smaller band gaps allow for excitation with visible or near-infrared light. As shown in Figure 11 (right), common photocatalysts span a range of band gaps—from Ta_2_O_5_ (4.0 eV, highest) to CdSe (1.7 eV, lowest). Titanium dioxide (TiO_2_, rutile form), one of the most widely used photocatalysts, has a band gap of approximately 3.0 eV. A recent paper [193] describes graphitic carbon nitride (g-C_3_N_4_) as a promising photocatalyst due to its suitable bandgap (2.6–2.8 eV), high chemical stability, and responsiveness to visible light. A new method for preparation of highly crystalline g-C_3_N_4_ has been developed, involving the simple pressing of sodium chloride and carbon nitride into a pellet, followed by heat treatment. The resulting catalysts were successfully tested for water splitting and photoelectrochemical water splitting.

To enhance the absorption of energy at specific wavelengths and improve interactions with reactants, (nano)composite heterojunction photocatalysts (HPCs) are used [195]. Figure 12 illustrates the basic types of these catalysts. Typical configurations involve combinations of n-type and p-type semiconductors. In addition to the well-known Z-scheme (Figure 12), the S-scheme heterojunction is also commonly employed [196]. In the Z-scheme, two distinct semiconductors—Photocatalyst I (PS I) and Photocatalyst II (PS II)—work in conjunction with an electron donor/acceptor (A/D) pair. PS I and PS II can be in direct physical contact (as in the direct Z-scheme, Figure 12) or spatially separated. In the latter configuration, during photocatalysis, photogenerated electrons migrate from the conduction band (CB) of PS II to the valence band (VB) of PS I via an electron-conductive donor/acceptor mediator, such as metallic Ag, Au, or Pt. The S-scheme heterojunction consists of a reduction photocatalyst (RP) and an oxidation photocatalyst (OP) with staggered band structures. Although the structure resembles a type-II heterojunction, the charge transfer pathway is fundamentally different, providing more efficient separation of photogenerated charge carriers [196].

Typical combinations of HPCs are based on SnO–TiO_2_, NiO–TiO_2_, CdS–WO_3_, TiO_2_/g-C_3_N_4_ [195,197], alfa-Fe_2_O_3_/g-C_3_N_4_ [198], Cu_2_O/In_2_O_3_ and Cu_2_O/In_2_O_3_/TiO_2_ [199], SiH/CeO_2_, Fe_2_TiO_5_/Fe_2_O_3_/TiO_2_, Ag_2_O/Fe_2_O_3_/TiO_2_, Bi_2_O_3_/C_3_N_4_/TiO_2_ [200]; combination with MOFs, e.g., with the UiO-66: NH_2_-UiO-66/CoFe_2_O_4_/CdIn_2_S_4_ enabling double p-n junction [201], zeolitic MOF—ZIF67: ZIF67/NiMoO_4_ [202], covalent triazine-based framework (CTF): PtOx/CTF-1 and RuOx/CTF-1 [203]. The synthesis of electro- and photocatalysts—including hydrothermal/solvothermal methods, ball milling, template-assisted approaches, and others—is discussed in [204]. Materials based on doped nanodiamonds represent a newly emerging class of photo- and electrocatalytic materials [205]. Beyond their electronic properties, the manner of contact between semiconductor components also plays a crucial role in the catalytic performance of HPCs [206].

The incorporation of metal nanoparticles into a photocatalyst (PC) or HPC enables the introduction of plasmonic catalysis [207]. Surface plasmons refer to the collective, in-phase oscillations of delocalized electrons within metal nanoparticles, excited by the electromagnetic field of incident light at the interface between a metal and a dielectric material. This phenomenon is commonly observed in structured arrays of metal nanoparticles deposited on dielectric supports such as TiO_2_. Plasmonic resonances typically occur in the visible spectrum and are primarily associated with noble metal nanoparticles like gold (Au), silver (Ag), and copper (Cu).

Unlike light absorption above the semiconductor bandgap, which leads to direct charge separation, plasmonic absorption does not directly generate charge carriers. Instead, metal nanostructures function as optical antennas, concentrating light into intense localized electric fields. These enhanced fields are often increased by several orders of magnitude and depend strongly on the spatial arrangement, shape, and size of the nanoparticles. Localized surface plasmon resonance (LSPR) occurs at the interface between the metal nanoparticle and the photocatalyst, generating amplified surface and interparticle electric fields.

These fields enhance light absorption by surrounding molecules or materials coupled to the NP and/or produce “hot carriers” (high-energy electrons or holes) within the NPs themselves [208,209]. In simple terms, a plasmon can be thought of as a nanoscale “picometric surface ball lightning”, and interacting with it is akin to its annihilation. Both the localized electric field and elevated temperatures induced by the plasmon can influence the course of chemical reactions.

Figure 13 illustrates the plasmon-enhanced reduction of CO_2_ to CO over a plasmonic HPC [210]. Iron-based MOFs (Fe-MOFs), especially when combined with other metals (e.g., M/Fe-MOF, where M = Co, Cu, Mg), have shown particularly high efficiency [211]. Additional examples of plasmonic HPCs can be found in [41,212,213,214,215,216,217]. Furthermore, the catalytic effect of the electric field can be significantly amplified by the application of an external electric field [45].

In comparison to classical heterogeneous catalytic processes, including electrocatalytic ones, the evaluation of efficiency in photocatalytic (PC), heterojunction photocatalytic (HPC), and plasmonic HPC systems is more complex.

Quantum yield (*QY*), quantum efficiency (*QE*), photonic yield (*PY*), and apparent photonic yield (*APY*) are the main quantities recommended by IUPAC for evaluating photocatalytic processes [218].

*QY* for a photocatalytic process is defined as the ratio between the number of useful reaction events (i.e., the number of reactant species transformed via a chemical reaction) and the number of photons absorbed at a specific wavelength. However, in heterogeneous photocatalytic suspensions, the direct measurement of absorbed photons is not straightforward. This is because radiation losses due to scattering outside the reactor also contribute to radiation extinction. As a result, solving the photon balance using radiation transport models becomes necessary, which is a rather complex task [219,220].

The need to express catalytic efficiency in a more practical and application-oriented way has led to the development of alternative definitions, particularly in relation to photocatalytic water splitting [221]. One of the most representative is the Solar-to-Hydrogen (*STH*) Conversion Efficiency, which is defined as:(11)$$STH\left(\%\right)=100\frac{Output\;energy\;of\;hydrogen\;evolved}{Energy\;of\;incident\;solar\;light}=100\frac{\left(mmol\;H_{2}s^{-1}\right)\;*\;237(kJ\;{mol}^{-1})}{p_{in}\left(MW\;{cm}^{-2}\right)\;*\;Area({cm}^{-2})}$$

*p*_in_—Energy of input irradiation.

Another approach is to evaluate the photocatalytic thermodynamic efficiency factor (*PTEF*)**,** which is defined as the ratio of the rate of formation of a reaction component (e.g., •OH) to the input energy [222,223,224]. Depending on the evaluation criteria used, different values may be obtained [224]. Therefore, economic and ecological considerations—such as capital expenses (CAPEX) and operational expenses (OPEX), as well as catalyst lifetime and deactivation—must also be considered.

### 3.8. Deactivation

As illustrated in Table 1, the lifetime of catalysts is limited due to loss of activity and selectivity; that is, the catalyst undergoes deactivation. This can occur through one or more of the following mechanisms [79,225,226]:Poisoning by impurities, e.g., reaction of sulfur compounds with metals in hydrogenation processes.Formation of side products that adhere to the surface and block reactants from accessing catalytic sites (e.g., tar formation on cracking catalysts).Detachment of anchored functional groups from the surface of heterogenized catalysts (acidic groups, basic groups, metal–organic groups, etc.).Transformation of catalytic centers into soluble or vaporizable compounds and subsequent leaching into the reaction environment (common in reductions involving oxidizing reactants such as nitro- and nitroso-compounds).Growth of metal (nano)particles into larger ones (sintering), resulting in loss of activity due to decreased specific surface area.Abrasion or breakage of catalyst particles, mainly occurring in fluid and suspension reactor systems, although catalysts in fixed-bed reactors also experience this slowly over time.

Deactivation tends to be significantly higher in liquid-phase reaction systems. The following measures can help prevent or mitigate deactivation:Use catalysts that are chemically stable and operate within appropriate temperature ranges; for example, observe the limited working temperature of OMOP catalysts.Operate at the lowest possible temperature that still ensures good catalyst performance.Utilize supported catalysts, preferably structured catalysts (StrCats) when possible.Choose suitable chelating groups for supported catalysts.Minimize the presence of chemical compounds that can attack or poison the catalyst.Ensure proper flow conditions around the catalyst surface—adequate for efficient transport of reactants and products but not so intense as to cause catalyst abrasion; this also applies to mixing intensity.

These measures must be implemented while considering the overall economic feasibility of the process.

## 4. Catalytic Technologies

Figure 14 shows a general arrangement of catalytic technology. Fresh reactants and solvents/inerts are mixed with recycled streams and, after physical treatment (e.g., pre-heating), they enter the reactor system. A catalyst is either embedded in the reactor before the process starts or added gradually during operation. External energy, most frequently electromagnetic irradiation (microwave, light, plasmonic, etc.), can also accelerate the chemical process. The reaction mixture then proceeds to separation units (filtration, sedimentation, extraction, distillation, crystallization, membrane processes, chromatographic separation, etc.), where products, waste, unreacted reactants, and solvents are isolated. Unreacted reactants and regenerated solvents are recycled back to the reactor. In some cases, the catalyst is separated and recycled, as well. An efficient catalytic technology design integrates the reaction heat with separation units [227,228]. In special technology arrangements, such as reactive distillation [229,230] or membrane reactors [135,231,232], reaction and separation steps occur within a single apparatus.

Technological units can operate under batch mode (all reactants, solvents, and catalysts are added at the start and the reaction mixture is processed after a set reaction time), continuous mode (all streams flow continuously; the catalyst is usually deposited before the process begins), or semi-batch mode (one reactant, e.g., hydrogen or oxygen, is added gradually until the desired conversion is reached). Isothermal (constant temperature) and adiabatic (reaction heat is carried away with the output mixture) regimes are commonly applied. A special category is autothermal reactors [86,233], in which the reaction heat is balanced by the heat content of incoming reactant streams—such as in ammonia production [6].

### 4.1. Gas-Phase Reactors (G-S)

Compared to laboratory reactors, mass transport limitations at the gas–solid (G–S) interface significantly affect the performance of industrial G–S reactors. The molar flow of component *i* at the G–S interface can be expressed as:(12)$${\dot{J}}_{i,GS}={(k}_{GS}a_{LS}(p_{i,G\_bulk}-p_{i,S})$$

${\dot{J}}_{i,GS}$—molar flow of the component “i” for the G-S interfacial (mol·m^−2^·s^−1^);

*k*_GS_—transport coefficient for the G-S transport (mol·Pa^−1^, s^−1^, or m·s^−1^);

*a_GL_*—interfacial area for the G-L and L-S, respectively (m^2^·m^−3^);

*p*_i,G,bulk_, *p*_i,S_—Partial pressure of the component “i” in the bulk and on the catalyst surface, respectively (Pa).

Regarding the value of *k*_GS_, the intensity and type of mixing must be considered [234]. Mixing can be categorized into macromixing, mesomixing, and micromixing regimes, with approximate characteristic length scales of 0.1 m, 0.001 m, and 10^−5^ m, respectively. These length scales correspond to domains that can move relatively freely, randomly colliding and restructuring. Each domain contains thousands of molecules or reactive species and may differ in composition and flow rate within the reactor, a situation referred to as segregated flow. The extent of segregation can be assessed from the residence time distribution function [234]. Chemical reactions occur at the picometer scale (~10^−12^ m, involving orbital interactions at 10–50 pm). Thus, even in the micromixing regime, established by strong turbulent motion, diffusion phenomena still influence the reaction rate. Several authors [235,236,237] have described the effect of mixing intensity on flow segregation and its impact on reaction conversion. These studies, along with developed models, aid in reactor design [238,239]. Modern simulation software, such as COMSOL, enables efficient modeling of mixing phenomena [240].

The highest values of *k*_GS_ occur under micromixing conditions. Regarding external heat transfer, in addition to heat transport within catalyst particles, heat must be transferred through the catalyst bed (in fixed-bed reactors) and through the reactor walls. Heat transport within the reaction zone is directly proportional to convective mass transport, with conductive heat transfer (e.g., radiation) as a secondary contribution. Highly exothermic reactions with significant conversion generate substantial local temperature increases. When temperature strongly affects selectivity or when structural risks to the reactor exist, reactor diameters must be kept narrow (e.g., tubes with ID 1–5 cm). Consequently, cooling methods involving cooled reactants, tubes, fluidized beds, or moving catalyst beds are employed.

Basic types of gas-phase catalytic reactors are shown in Figure 15, with their main features summarized in Table 3. Fixed-bed reactors are suitable for reaction systems with mild heat generation or where temperature variations do not significantly affect selectivity (e.g., oxidation of SO_2_ to SO_3_ over vanadium catalysts). A special type of fixed-bed reactor is the skeletal reactor, characterized by low mass transport resistance [7]. These reactors are widely used for treating flue gases, where pollutant concentrations are low, but flow rates are high [7,241,242,243,244]. Multitube reactors (Figure 15b) are employed for strongly exothermic reactions, such as the oxidation of naphthalene to phthalimide. They can contain thousands of tubes, resulting in more complex construction compared to fixed-bed reactors. Fluidized-bed reactors (Figure 15c) are used when intensive mass transfer from the bulk to the catalyst surface is necessary (e.g., polymerization) or when catalysts deactivate rapidly, such as in FCC [245]. Moving-bed reactors (Figure 15d) are less common but are used when catalysts deactivate quickly and require regeneration, such as in pyrolysis [246,247].

### 4.2. Liquid-Phase Reactors (L-S)

Typical liquid-phase heterogeneous catalytic reactors operate with suspensions of catalyst particles ranging from 0.05 to 3 mm in size. The main types of catalytic processes carried out in the liquid phase include esterification, etherification, alkylation, condensation, and polymerization. Catalysts are usually heterogenized materials, such as acid and basic catalysts (e.g., sulfonated and aminated polymers [49]). Other catalyst groups include zeolites and zeolite-type materials (mainly acidic catalysts) [88,162,248,249], activated clays (acidic) [250,251], and hydrotalcites (basic) [252,253,254]. Modified metal–organic frameworks (MOFs) are also used in suspension form for liquid-phase reactions [255,256,257]. Biotechnological reactors, including wastewater treatment systems using microorganism flocs ranging from 0.05 to 2 mm, can also be considered as liquid-phase heterogeneous catalytic reactors [19,258,259,260].

Besides internal diffusion (Equation (5)) and intrinsic reaction kinetics (Equations (3) and (4)), the rate of the process is also controlled by liquid–solid (L–S) mass transport:(13)$${\dot{J}}_{i,LS}=k_{LS}a_{LS}(c_{i,L\_bulk}-c_{i,S})$$

${\dot{\;J}}_{i,LS}$—molar flow of the component “i” for the LS interfacial (mol·m^−2^·s^−1^);

*k*_LS_—transport coefficients for the L-S transport (mol·Pa^−1^, s^−1^, m·s^−1^);

*a_LS_*—interfacial area for the L-S (m^2^·m^−3^);

*c*_i,L,bulk_—concentration of the component “i” in the bulk liquid (mol·m^−3^).

Because diffusivity in the liquid phase is significantly lower than in the gas phase (by about a factor of 10^−4^), mixing plays a crucial role in the effectiveness of the catalytic process [234]. Typically, mesomixing and micromixing regimes are applied. Proper, intensive mixing ensures sufficient L–S mass transport and heat transport via convection. The intensity of mixing is proportional to the energy consumption per unit mass of the reaction mixture (kW/t), which can vary from 0.1 to 4 kW/t [261,262,263]. The energy utilization for generating a high L–S interfacial area and high mass transfer coefficient depends on the type of impeller used (Rushton turbine, marine propeller, pitched blade, shaft impeller, etc.) and the number of impellers (dual or multiple impellers on one shaft) [86,263]. In the case of organic modified porous (OMOP) catalysts and biocatalysts, a gentle mixing mode that minimizes mechanical damage to solids is necessary [259].

### 4.3. Gas Liquid Phase Reactors (G-L-S)

G-L-S reactors are typical with a more complex mass transport, i.e.,

G-L:


(14)
$${\dot{J}}_{i,GL}=k_{GL}a_{GL}(p_{i,G\_bulk}-{K_{i,eq,GL}c}_{i,L})$$


${\dot{\;J}}_{i,GL}$—molar flow of the component “i” for the G-L interfacial (mol·m^−2^·s^−1^);

*k*_GL_—transport coefficients for the G-L transport (mol·Pa^−1^, s^−1^, m·s^−1^);

*a_GL_*—interfacial area for the G-L interfacial (m^2^·m^−3^);

$K_{i,eq,GL}$—equilibrium constant for the component “i”, G-L transport (Pa mol^−1^ m^3^).

L-S (Equation (10));Internal diffusion (Equation (5)).

Figure 16 illustrates the basic types of gas–liquid–solid (G–L–S) reactors, and Table 4 summarizes their typical features. The simplest and cheapest reactor is the so-called trickle bed reactor (TBR), in which the catalyst is deposited in a fixed bed, and liquid and gas flow through the void spaces between catalyst particles. Usually, the gas phase is introduced at the bottom of the reactor; however, co-current flow from the top is also possible [86,264]. These reactors are quite robust and are frequently used for treating crude oil fractions, such as desulfurization and denitrogenation over Co–Mo sulfide catalysts [265]. In hydrotreatment processes, deactivation is suppressed by a large excess of hydrogen, which decomposes higher molecular weight side-products that could potentially stick to the catalyst surface.

The most commonly used G–L–S reactors for laboratory research and industrial applications are mechanically stirred reactors (MSR). Figure 16b depicts a stirred reactor with a simple impeller, cooled via a jacket, and equipped with an internal filter. Proper, intensive mixing enables sufficient gas–liquid and liquid–solid mass transfer, as well as heat transfer via convection. The mixing intensity in MSRs is usually higher than in L–S reactors, reaching up to 10 kW/t. If the reactor’s mixing system is not properly designed, mechanical stirring energy is dissipated as heat. As shown in Table 3, the volume of stirred reactors ranges from a few cubic centimeters (laboratory scale) to several hundred cubic meters (e.g., aeration systems in wastewater biotreatment). Bubbled column reactors (BCRs) are much simpler than stirred reactors, so even large-volume reactors (>10 m^3^) can be operated at relatively high pressures (up to 5 MPa). A major advantage of BCRs over MSRs is the gentler treatment of catalyst particles, which leads to longer catalyst life due to less abrasion. However, BCRs have lower mass transfer coefficients (*k*_GL_, *k*_LS_) and smaller interfacial areas (a_GL_). Transport parameters, including better “fluidization” of solids, can be improved by recycling the gas phase, as shown in Figure 16c. Often, a portion of the liquid phase is also circulated along with the gas phase, which imposes special requirements on pumps. These reactors are called “ebullated bed reactors” (EBR) [266,267,268] and are widely used in petrochemical applications [265].

**Figure 16 molecules-30-03279-f016:**
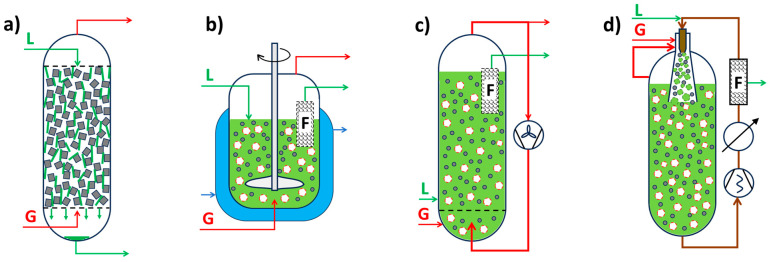
G-L-S catalytic reactors: (**a**) trickle bed, (**b**) mechanically stirred with an inbuilt filter—F, (**c**) bubble column or ebullated with an inbuilt filter and a circulation pump, (**d**) loop reactor with a circulation pump, heat exchanger and a filter. The images were prepared using knowledge from [56,86,268].

Figure 16d depicts a Buss-type loop reactor [269,270]. Thanks to a special combination of ejector and diffuser with gas-phase recycling, this reactor is among the most efficient in terms of gas–liquid and liquid–solid mass transfer. Catalyst particles are small (up to about 1 mm), so internal diffusion does not limit the catalytic process (mainly hydrogenation). A special pump allows slurry containing up to 50% gas and vapor to be circulated while causing minimal damage to the catalyst particles. However, this type of reactor is not suitable for biocatalysis.

### 4.4. Special Chemical Catalytic Reactors

In addition to the reactors described above, there are more complex arrangements such as parallel flow, radial flow, composite structured packing, catalytic distillation reactors, programmed-release catalysts, and others [62,268,269]. All these designs aim to maximize catalyst productivity and minimize deactivation. However, greater engineering complexity leads to higher costs, which must be considered in the overall design of the catalytic reactor–separation system.

### 4.5. Electrocatalytic Processes and Reactors

Figure 17 introduces five main types of water electrolyzers (ELZ). They operate at the following temperatures (°C) and pressures (bar): PEMWE: 50–80, 1–70; AEMWE: 40–60, 1–35; AWE: 70–90, 1–30; SOEC: 700–950, 1; PCCEL: 300–600, 1. The operating conditions indicate that PEMWE, AEMWE, and AWE are fed with liquid water, while SOEC and PCCEL operate with water vapor. Platinum- and nickel-based electrodes are used for liquid-phase ELZ. In some cases, iridium and iridium oxide are also employed. Metal and metal oxide particles are micro- or sub-micrometer-sized to achieve a high specific surface area (activity) along with sufficient mechanical and chemical stability.

In modern AWE systems, KOH (5–7 mol·dm^−3^) is used as the electrolyte and polyphenylene sulfide (PPS) mesh-based polysulfone (PSF)-zirconia (ZrO_2_) composite diaphragm (140 μm thickness) as a separator [271]. Sulphonated polyfluorohydrocarbon polymer (Nafion) is used for transport protons. For anion transport membranes, many types of polymers are employed, such as polysulfone, poly(ether ketone), poly(ether imide), poly(phenylene oxide), polybenzimidazole, and others. Various cationic groups are attached to the polymer backbone, including nitrogen-based, phosphonium, phosphatrane, sulfonium, and metal-containing anion-conducting groups, such as ruthenium(II) complexes. Yttria-stabilized zirconia (YSZ) is used as the electrolyte in SOEC. (Y,Yb)-doped Ba(Ce,Zr)O_3_ serves as the electrolyte in PCCEL. Perovskite-type materials (e.g., LSCF, LSM, Ni/YSZ, Ni-BZY/LSC, BCFYZ) are used as electrodes/catalysts in SOEC and PCCEL electrolyzers [176].

SOEC technologies are driven by their ability to operate at high current densities (e.g., 3.6 A cm^−2^ at 1.48 V, 950 °C) and efficiencies up to 90%. PCCEL electrolyzers also exhibit high efficiency. Unfortunately, both SOEC and PCCEL suffer from low mechanical stability, as the thickness of individual layers is only in the tens of microns.

In the PEMWE and PCCEL the O_2_ is generated (Oxygen Evolution Reaction: OER) at anode:(15)$$H_{2}O\;\to \;0.5O_{2}\;+\;2e^{-}\;+\;2H^{+}\;(E_{A}^{0}\;=\;+1.229\;V\;\mathrm{vs.}\;\mathrm{SHE},\;25\;{}^{\circ}C)$$

Proton is transported through the proton-conducting membrane and H_2_ is formed (Hydrogen Evolution Reaction: HER) on the cathode:(16)$$2H^{+}\;+\;2e^{-}\;\to \;H_{2}\;(E_{C}^{0}\;=\;0.000\;V\;\mathrm{vs.}\;\mathrm{SHE},\;25\;{}^{\circ}C)$$

In AWE, reduction of water at pH = 14 occurs at cathode:(17)$$2H_{2}O\;+\;2e^{-}\;\to \;H_{2}\;+\;2{\mathrm{HO}}^{-}\;(E_{C}^{0}\;=\;-0.828\;V\;\mathrm{vs.}\;\mathrm{SHE},\;25\;{}^{\circ}C)$$

Hydroxyl ions are conducted through the diaphragm, and they are oxidized at the anode(18)$$2{\mathrm{HO}}^{-}\;\to \;0.5O_{2}+H_{2}O+2e^{-}\;(E_{A}^{0}\;=\;-0.4101\;V\;\mathrm{vs.}\;\mathrm{SHE},\;25\;{}^{\circ}C)$$

Mechanism in AEMWE is similar to AWE

In the SOEC, thanks to high temperature, water vapor is reduced at the cathode and hydrogen is formed (HER):

H_2_O(g) + 2e^−^ → H_2_(g) + O^2−^(19)


Oxygen ions migrate to the anode where O_2_ is formed (OER) by oxidation

O^2−^ → 0.5O_2_(g) + 2e^−^(20)


Electrolytic cells for hydrogen production (HER) are assembled into stacks of up to several hundred cells (Figure 18). As mentioned above, the voltage efficiency is currently (2025) about 65%. It is expected that with improved catalysts (electrode compositions) and more advanced membranes and separators, efficiencies exceeding 95% can be achieved by 2050 [176,272,273,274,275,276,277,278].

Details about the development of advanced 1 GW AWE and PEM electrolyzers can be found in [277].

Electrocatalytic reactors are mainly used for liquid-phase redox reactions, which involve three main steps [279]:Electron transfer;Reorganization of intramolecular bonds;Reorganization of the solvation shell.

One of the reactants (e.g., H_2_, O_2_) is generated from a component in the reaction mixture, usually from water [280,281,282]. To lower the reaction barrier, some mediators (promoters) can be added, such as *p*-toluenesulfonic acid (PTSA) or pyridinium *p*-toluenesulfonate (PPTS) in hydrogenation systems [283], iodide compounds in carbene reduction [284], salts of transition metals and sulfur compounds in the oxidation systems [282].

In addition to water electrolysis (HER), electrocatalytic reduction of carbon dioxide (CO_2_ERR), coupled with the formation of valuable chemicals (CO, CH_4_, alkanes and alkenes, formic acid, etc.), is of great interest [285,286,287,288,289,290,291,292]. Among the many reactions, the generation of CO and O_2_ is listed as follows:

Cathode:(21)$${\mathrm{CO}}_{2}(g)\;+\;H_{2}O(l)\;+\;2e^{-}\;\to \;\mathrm{CO}(g)\;+\;2{\mathrm{OH}}^{-}(\mathrm{aq}.)\;(E_{C}^{0}\;=\;-0.52\;V\;\mathrm{vs.}\;\mathrm{SHE},\;25\;{}^{\circ}C)$$

Anode:(22)$$2{\mathrm{OH}}^{-}(\mathrm{aq})\;\to \;½O_{2}(g)\;+\;H_{2}O(l)\;+\;2e^{-}\;(E_{A}^{0}\;=\;+0.40\;V\;\mathrm{vs.}\;\mathrm{SHE},\;25\;{}^{\circ}C)$$

It is worth comparing the minimum voltage (0.92 V) for the electrocatalytic conversion of CO_2_ to CO in water (alkaline environment), which is significantly lower than that for water electrolysis (1.23 V). This highlights the importance of direct CO_2_ERR directly by water. The formation of CO and other C_1_ products (COOH^−^, CH_4_, CH_3_OH) is mainly carried out using gold, silver, zinc, and tin catalysts [293,294]. Copper-based electrocatalysts are typically used for C–C coupling, enabling the generation of multicarbon (C_2_^+^) chemicals such as ethylene (C_2_H_4_) and ethanol (CH_3_CH_2_OH) [294].

Details about the construction (electrodes, membranes, electrolyte composition) and operation of electrolyzers are well described in [289]. Figure 19 depicts a typical laboratory electrolyzer for CO_2_ conversion in an alkaline environment. High yields of C_2_^+^ products (C_2_H_4_, CH_3_CH_2_OH) have been achieved [295].

Procedures for the electrocatalytic synthesis of C_3_^+^ alkanes and aromatics using CO_2_ as a starting material are described in [296]. The majority of electrocatalytic reactors operate with reactants in the liquid phase in arrangements similar to that depicted in Figure 17. A gas-phase transformation of CO_2_ to formate is described in [297]. The key element of the experimental setup was a filter press cell. A humidified CO_2_ stream was fed to the cathode (Bi/C-GDE) containing Bi catalytic particles. A 1 M KOH aqueous solution was supplied to the anode (DSA/O_2_, Ir-MMO [mixed metal oxide] on platinum). As the current density increased (from 90 to 600 mA cm^−2^), the faradaic efficiency for formate varied from 96% to 74%, with CO and H_2_ as side products.

Reference [298] reports a cobalt phosphide electrocatalyst designed for efficient hydrogen evolution from both water and seawater. The CoP-Co_2_P/PC catalyst achieved current densities of 10 mA·cm^−2^ with Tafel slopes of 55.4 mV·dec^−1^ in alkaline solutions and 93.6 mV·dec^−1^ in neutral solutions, requiring overpotentials of only 96.7 mV and 162.1 mV, respectively. Moreover, this catalyst showed promising activity in alkaline seawater electrolysis, reaching a low overpotential of 111.2 mV at 10 mA·cm^−2^, along with a Tafel slope of 64.2 mV·dec^−1^.

While these Tafel slopes are higher than those of the benchmark Pt/C electrocatalyst (33.1 mV·dec^−1^), the CoP-Co_2_P/PC catalyst demonstrated significantly better stability, with only a 9% decline in performance after 24 h compared to a 36% decline for Pt/C after 8 h. The experimental findings align well with DFT calculations, which identified the H–OH* to H* conversion as the rate-limiting step on the CoP side of the Co_2_P-CoP heterostructure, associated with an energy barrier of 0.48 eV.

Electrocatalysis can also be successfully applied to various other redox reactions, such as C–H bond cleavage and nucleophilic substitution at the *α*-C carbon in monoalcohols, transformation of the C–C bond in diols, formation of alcohols, aldehydes, and acids from saturated carbonyls, hydrogenolytic splitting of C–O and C–C bonds in lignin, and others [299,300]. Copper- and nickel-containing electrodes are mainly used for these processes. Special attention is given to the treatment of 5-hydroxymethylfurfural (HMF), which can be converted to precursors for ethers, ketones, polyurethanes, polyesters, and polyethers, such as 2,5-dihydroxymethylfuran (DHMF), 2,5-dimethyltetrahydrofuran (DHMTHF), 2,5-dimethylfuran (DMF)—high-energy-density premium biofuels—2,5-hexanedione (HD), and 5,5′-bis(hydroxymethyl)hydrofuroin (BHH). Oxidation of HMF produces valuable chemical products, including 2,5-diformylfuran (DFF), 5-hydroxymethyl-2-furancarboxylic acid (HMFCA), 2,5-furandicarboxylic acid (FDCA), and maleic acid (MA), which are precursors/intermediates for the polymer industry and chemical/pharmaceutical technologies [301].

Electrocatalytic processes are also important for the treatment of liquid organic hydrogen carriers (LOHCs). The hydrogenation–dehydrogenation of LOHCs (typically organics containing multiple aromatic groups) is commonly carried out by heterogeneous catalysis, with H_2_ produced by electrolysis using renewable electrical energy [280]. The typical anodic reaction is the oxygen evolution reaction (OER), which suffers from sluggish kinetics requiring high overpotentials, thereby reducing energy efficiency and increasing costs. Additionally, the production of O_2_ often lacks onsite utilization. Therefore, combining OER with the production of valuable oxidation products can significantly offset these disadvantages. In recent years, research on the electrocatalytic oxidation of biomass-derived platform chemicals such as HMF, furfural (FF), glycerol, lignin, and others has attracted widespread attention and made significant progress. Acetonitrile, as an LOHC, is also of interest; it can be hydrogenated to ethylamine [302], and via dehydrogenation [303], H_2_ is produced.

Recent work has focused on the “in situ” generation of H_2_O_2_ formed from water and O_2_ reduction (ORR) [304,305,306,307]. Mesoporous carbon with oxygenated functional groups, prepared via a nano-casting approach, was used as the cathode catalyst for O_2_ reduction to H_2_O_2_. A honeycomb-shaped TS-1 catalyst was used for oxidation, achieving 86% H_2_O_2_ conversion with 99% selectivity for the conversion of ethylene to ethylene glycol [307].

Similar to classical (chemical) catalysts, deactivation plays an important role in electrocatalytic processes, particularly with organic components (e.g., intermediates in CO_2_ERR). Intensive mixing of the fluid (liquid, gas-vapor) surrounding the electrode minimizes deactivation. Mixing may be achieved by mechanical stirring, circulation, or a rotating disk electrode [308]. Deactivation increases with temperature, which is directly related to the electric current density. This means that higher productivity of the electrocatalytic reactor is accompanied by faster deactivation. One regeneration method is high-temperature pulse annealing [309]. Another approach involves periodic input of metal complexes, e.g., copper lactate, for systems using copper electrodes [310].

Despite the vast number of researchers and published papers, electrocatalytic technologies, except for water electrolysis, are not yet widely exploited for large-scale production. One reason is that electricity in general, and green energy (solar, wind, hydropower) in particular, is still more expensive than energy from fossil fuels—for example, the price of green hydrogen is about double that of hydrogen produced by steam–methane reforming (SMR). However, the high purity (>99.99%) of green hydrogen offers a significant advantage over SMR hydrogen (~99%), which is important for synthesizing valuable organic chemicals. Another disadvantage is the higher purity requirement for reactants to minimize deposition (sticking) of undesired products on electrodes and membranes. Nevertheless, with the increasing scarcity of fossil resources, the use of electrocatalytic methods is expected to grow.

### 4.6. Heterogeneous Photocatalytic Processes and Reactors

In the design of a photocatalytic process, four main groups of topics need to be considered [311]:Light source:
Natural or Artificial;Light concentrators (mirrors, lens):
Without Light Concentrators;With Light Concentrators;Wavelength:
Visible (400–780 nm);UV (190–400 nm);Type:
Conventional Lamps;Light Emitting Diodes (LED);Catalyst:
TiO_2_ or not TiO_2_ based;Size:
Micron;Nanometer;Catalyst movement:
Immobilized (On the wall of the reactor, Bed, Fibers, Membrane, or Disk);Suspended;Reactor:
Batch or Continuous;Hydrodynamic behavior:
Continuously Stirred (CSTR);Plug Flow (PFR).

Progress in development of photocatalytic reactors is illustrated by the following milestones [311]:

1990—Parabolic Trough Reactor (PTR);

1992—Classical Annular Reactor (CAR);

1996—Fluidized bed Reactor (FBR);

1998—Tube Light Reactor (TLR) and Multi-tube Reactor (MTR);

2001—Spinning Disk Reactor (SDR);

2004—Optical Fiber Reactor (OFR);

2005—Light-emitting Diode Photocatalytic Reactor (LED PCR);

2006—Micro-channel Reactor (MCR);

2010—Integrated Membrane Reactor (IMR);

2012—Capillary Array Photocatalytic Reactor (CAPR);

2014—Photocatalytic Micro-reactor (PCMR);

2019—Packed Bed Reactor (PBR).

The light source (type and intensity) strongly affects both CAPEX and OPEX. Exploiting sunlight is the cheapest option for photocatalytic (PC) processes. Figure 20 shows a sunlight-driven set of PC reactors [312]. The proposed concentrator cavity, statically aligned with an east–west axis and the reactor aperture normal statically oriented toward the sun path at the equinox, allowed for high UV–vis photocatalytic efficiency throughout both the day and the year. The reduction of potassium iron(III)-oxalate to potassium iron(II)-oxalate under photon absorption was used for chemical actinometry. An economic analysis based on CAPEX and OPEX, considering the changing intensity of UV sunlight, was performed. UV light concentrators are also applied with artificial sources [313,314], as well as in commercially available photoreactors [315]. A great advantage of artificial light sources is the ability to tune both the wavelength (LEDs from 240 to 900 nm; commercially available wavelengths range from 365 nm to 625 nm with efficiencies up to 60%) and intensity [315]. These factors have specific effects on the activity and selectivity of photocatalytic reactions [315,316,317,318,319].

Photocatalysts may be used for both gas and liquid phase reactions; photoactive catalytic species can be deposited on the reactor walls (films/layers) or on the surface of a support [311,320,321,322,323,324,325]. Liquid phase photoactive catalysis is also possible with suspended micro- or nano-photocatalyst particles. In this latter case, separation of the suspended catalyst is an inevitable step in technology. Separation is performed either directly in the reactor (via an in-built filter) or in a unit following the reactor (sedimentation, filtration, centrifugation, etc.). A great advantage is the use of magnetic catalytic particles, e.g., doped with Fe_3_O_4_, enabling magnetic separation [326].

MOFs and COFs occupy a special position in photocatalysis. Their tunable properties—through various central atoms and linkers—allow a wide range of chemical reactions to be performed [49,327,328,329,330,331,332]. However, for long-term applications, stability—mainly determined by the type of linkers—must be considered with respect to temperature and the properties of the reaction mixture [333].

Suspended catalysts exhibit higher photocatalytic activity than fixed ones [219,321], but require additional reactor arrangements (in-built filters) or separate separation handling. Non-moving catalysts (layers, supported) do not offer such high activity; however, separation is not an issue [219]. Regarding deactivation, non-moving catalysts tend to be more sensitive than suspended catalysts [219]. Similar to classical heterogeneous catalysts, a higher flow rate of reactants and solvents/inerts around or above the catalyst surface slows deactivation [334,335]. This emphasizes the advantages of continuous micro-photoreactors (microchannel reactors) (Figure 21) [219,336,337].

A special category of catalysis is photothermal catalysis [338], which is carried out at temperatures up to 1000 K and enables more efficient performance of processes such as the reverse water–gas shift (RWGS), artificial photosynthesis, methanol production, Fischer–Tropsch synthesis, water splitting, ammonia synthesis, and others. Composites consisting of inorganic mixed oxides and pure transition metals (e.g., Ru/Mo_2_TiC_2_, Ni/LaInO_3_, Co/SrTiO_3_) that can withstand elevated temperatures are used as efficient photothermal catalysts [338].

Applications of PC, HPC, and plasmonic HPC may be divided into four groups:Splitting of water (hydrogen evolution) [200,221,224,316,336,339,340].C1 chemistry:
○Reduction of CO_2_ and artificial photosynthesis [210,341,342,343,344,345];○Transformation (Oxidation) of CH_4_ [203,346,347,348,349].Waste treatment:
○Gas phase (Air) [211,322];○Liquid (Water) phase [314,323,350,351,352].Special chemicals:
○Reactions with the “in situ” generated hydrogen peroxide [306,353,354,355];○Other (non-chromoselective) reactions [194,332,356,357];○Chromoselective reactions, i.e., initiation of a target functional group with the irradiation of a suitable wavelength [192,317,357,358,359].

### 4.7. Microreactors (MRs)

For all the above-mentioned reactors, issues related to mass and heat transport phenomena have been discussed. Chemical reactions occur at the picometer scale (10^−12^ m), with orbital interactions occurring within a space of about 50 pm. Flow and mixing of gases and/or liquids typically involve domains ranging in size from about 10 cm (macromixing) down to 10 µm (micromixing) [234]. Domains of approximately 10 to 100 μm contain many thousands of reacting species. These individual domains can differ in composition from one another. Collisions between these domains, accompanied by mixing at the nanoscale level, contribute to overall mixing, homogeneity, and mass and heat transfer.

Higher homogeneity and more intensive contact of reactants can be achieved by flowing through equipment with a small cross-sectional area, such as a capillary with an inner diameter smaller than 1 mm (microchannels). This is the typical dimension of a flow microreactor (MR) [360,361]. Of course, in the case of fixed-bed or micromonolith reactors, the reactor dimensions are larger, ranging from a few millimeters to several centimeters. In monolith reactors, microchannels are formed by void spaces among particles and between the walls. Flow segregation in a MR is significantly suppressed, resulting in more homogeneous reaction zones.

It is worth adding that evaluating flow in microchannels using only the Reynolds number (Re) is not sufficient for classifying the flow type (laminar: Re < 2000; turbulent: Re > 4000; transition: 2000 < Re < 4000) [362].(23)$$Re=\frac{D_{ch}{\bar{u}}_{F}\rho_{F}}{{µ}_{F}}$$

$D_{ch}$—diameter of a channel (m);

${\bar{u}}_{F}$—average flowrate (m·s^−1^);

$\rho_{F}$—density of the fluid (kg·m^−3^);

${µ}_{F}$—dynamic viscosity of the fluid (kg·m^−1^·s^−1^).

According to the above definition of Re, microreactors (MRs) operate with Reynolds numbers ranging from 0.1 up to 1500, i.e., they should exhibit “pure laminar flow” without any convective radial transport of mass and heat (only conductive transport). However, more detailed inspection reveals interactions of fluid domains flowing in laminar layers, which contribute to mixing [363,364,365]. When two or three phases (L1-L2, L-S, G-L, G-L-S) flow through a microreactor, the flow pattern becomes even more complex [262]. Immiscible liquid phases or liquid and gas phases form relatively large isolated (macro)domains, often with sizes controlled by the dimensions of the microchannel; this is known as slug flow or Taylor flow [366,367,368]. Solid nanoparticles contribute to mixing by increasing mass and heat transfer coefficients [369]. Mass transport intensification can also be achieved using ultrasound waves [370], magnetic fields in the case of magnetically sensitive moieties in the reaction mixture [371], and other external force fields [372]. Better contact between reactants, especially when present in different phases, can be attained by using more structured microreactors (e.g., mesh channels, inbuilt butterfly-shaped static mixers) [360,373,374,375,376] and/or various pre-mixers [377].

Compared to classical reactors, MRs exhibit higher mass transfer coefficients (which are crucial for efficient catalysis) and heat transfer coefficients (allowing easier control of the desired temperature) [239,375,378,379]. Due to the small channel dimensions in microreactors, they can be constructed with relatively thin walls even for high-pressure operation (requirements for wall thickness increase with diameter), which positively contributes to heat exchange and temperature control [380]. Thin walls are also advantageous for photocatalytic processes. Another benefit of MRs composed of semiconductors (e.g., ceramics) or catalysts/supports is the slight generation of piezoelectricity when a non-electrically conductive liquid flows turbulently with small changes in local pressure around such semiconductor parts [42,43]. All these factors can increase the overall rate of the chemical process as well as the selectivity toward desired products, valid for both non-catalyzed and catalyzed reaction systems [360,379,381,382,383,384,385,386,387]. An interesting group of MRs is membrane reactors [388,389,390], particularly “tube-in-tube reactors” (TiTR) [391]. The main reaction mixture flows either in the inner tube or in the inter-tube space, while low-molar reactants and/or products (H_2_, O_2_, NH_3_, H_2_O, etc.) flow in the other space [360,391,392]. The walls of the inner tube are permeable, allowing low-molar components to diffuse through. This arrangement enables proper dosing of reactants into the reaction zone and/or removal of products from the reaction mixture. Consequently, the equilibrium composition shifts toward a higher concentration of the desired product, or the extent of elimination of undesired components (e.g., wastewater treatment) is increased [136,389,393,394,395]. TiTRs have all the advantages of MRs, including intensive micromixing, high mass and heat transfer, and easy temperature control.

The positive features of MRs have led some to call a properly designed and operated MR a “miraculous apparatus”. However, increasing effort is being devoted to elucidating the peculiar behavior of MRs using chemical–physical and chemical engineering tools [361,396,397].

MRs can be constructed from the following [374]:Silicon/ceramic materials (well-characterized, high precision but expensive fabrication, high-temperature resistance up to 1300 K);Glass (allows visualization of reaction and flow, electroosmotic flow (EOF) possible, withstands high operating pressures up to 50 bar);Polymers (low cost, various fabrication techniques, tunable properties, disposable microreactors possible, low chemical and temperature resistance—max 500 K);Metals (durable, well-established fabrication techniques, sensitivity to reaction environment, suitable for higher temperature—up to 1100 K—and pressure—up to 50 bar—regimes).

Details about microreactor fabrication are in [361,373,374,375,398,399], while 3D printing [399] and lithography [375] represent modern methods. Heterogeneous catalysts can be used as fixed beds—isolated microparticles or micromonoliths—or as nanoparticles fixed on the walls or suspended in a flowing fluid (liquid) [360,400]. In situ growth methods, chemical vapor deposition (CVD), sol–gel, electrochemical deposition, liquid phase deposition, particle packing, and others are discussed in detail in [401]. Additional contributions to pressure drops must be considered for fixed-bed reactors [239,360,385].

Planar (linear or spiral) and coil configurations are the main types of MRs. The arrangement of microchannels in a planar microreactor and the relationship with pressure drop are illustrated in Figure 22 [402].

The best results (virtually full conversion of methane) were achieved with serpentine (e), coiled (f), coiled with serpentine (g), and coiled with double serpentine (h) reactors. However, these reactor configurations also exhibited the highest pressure drop (approximately 70 kPa), while reactors (a–d) showed only about a 4 kPa pressure drop but achieved an average methane conversion of less than 50%. From a fabrication standpoint, serpentine reactors are recommended as the most favorable option.

The advantages of microreactors (MRs) are somewhat offset by notable challenges, such as significant pressure drops along the capillary, fouling and clogging by higher-molecular-weight side products, and increased costs when scaling up to higher production volumes [398,403,404].

Nevertheless, performing reactions typically carried out in large-scale equipment—such as methane oxidation [402], nitrobenzene hydrogenation [405,406], or ammonia synthesis [407]—within MRs serves as a strong demonstration of their benefits. MRs are powerful tools for studying catalytic reaction mechanisms and diffusion phenomena at the (sub)nanoscopic level. Estimation of rate constants, adsorption constants, and transport parameters—each dependent on temperature and pressure—is easier, more precise, better reproducible, and more economical (due to reduced reactant consumption) in MRs compared to conventional reactors of larger volumes [379,385]. Properly designed MRs are also well-suited for High-Throughput Screening (HTS) of individual catalyst particles [408,409].

High-value chemicals, including biologically active compounds and pharmaceuticals, are among the main targets for MRs applications [391,392]. Figure 23 shows a scheme for the preparation of arylaldehydes from aryl bromides via reaction with CO and H_2_ (hydroformylation) [410]. The synthesis begins with a homogeneous catalyst composed of Pd(OAc)_2_ and the chelating agent cataCXium A (di(1-adamantyl)-n-butylphosphine). During the reaction, Pd(II) is reduced to Pd(0), with some reverse oxidation back to Pd(II). However, not all Pd(0) species are reoxidized; some form nanoparticles that are carried out with the product stream, as evidenced by a change in the output color. A palladium mass balance (output vs. input) revealed a significant portion of Pd metal was deposited on the reactor walls. To address this, each experiment was followed by washing with 20% HNO_3_ at 60 °C.

Yields of various arylaldehydes ranged from 53% to 98%, except heterocyclic and nitro-substituted compounds. The nitro group exhibits a strong electron-withdrawing effect, which makes it difficult for the bromide anion to dissociate from the aryl bromide.

The catalytic effect of palladium nanoparticles anchored on the reactor walls was evaluated in a control experiment without pre-washing the reactor with HNO_3_ and without adding Pd(OAc)_2_ to the feed. In this case, hydroformylation was catalyzed solely by the palladium deposited on the walls, but the reaction rate was approximately 100 times lower than that achieved with the homogeneous catalyst. This result supports the general principle in heterogeneous catalysis that, in terms of atomic efficiency, heterogeneous catalysts are significantly less active than homogeneous ones. However, their lower atomic activity can be compensated for by longer residence times and the ability to reuse the catalyst.

It is also important to note the dual effect of intensive micromixing: it enhances reaction rates (positive) but may reduce the stability of heterogeneous or heterogenized catalysts due to metal leaching (negative). Leaching of metal compounds or nanoparticles is particularly severe when the metal undergoes oxidation state changes—such as Pd(0) ↔ Pd(II)—as occurs in coupling reactions, carbonylations, and aminocarbonylations. Intensive mixing hinders the redeposition of reduced metal species onto supports or reactor walls, resulting in higher overall leaching rates [411,412,413]. This phenomenon limits the efficiency of MRs for such reactions. However, anchoring metal species on supports with a strong affinity for the metal can significantly reduce leaching [122].

The statements in the previous paragraph help explain why the majority of catalytic organic syntheses in microreactors are either conducted without a catalyst [391,414,415] or with a homogeneous catalyst [360,391,392,394,408,416,417,418].

A notable and successful example of heterogeneous catalysis is the synthesis of the drug rufinamide (brand name *Banzel*, an anticonvulsant), produced from 2,6-difluorobenzyl azide and propiolamide via a “click reaction” using copper capillaries, where the copper walls themselves acted as the catalyst [419]. Scaling up the metal microreactor by numbering up 25 copper capillaries (1/16” OD, 0.04” ID, each 2 m in length, arranged as coils) enabled a productivity of over 30 g·h^−1^ of rufinamide—competitive with classical reactor systems.

Another successful case is a bifunctional acid–base hierarchically structured monolithic microreactor (SBA particles doped with Zr), which showed high activity and stability in a continuous-flow tandem catalytic process for the synthesis of cyanocinnamate (CCNM) from benzaldehyde dimethyl acetal and ethyl cyanoacetate [420]. Yields of CCNM above 77% were sustained for over 300 min of time on stream.

An efficient enantioselective heterogeneous process performed in a microreactor is the preparation of quaternary isotetronic (QC) acids via direct self-aldol reactions of pyruvates over a silica-supported pyrrolidine-tetrazole catalyst (SiO_2_@Ley catalyst) [421]. Using ethyl pyruvate, a yield of 66% of hydroxy-free QC-isotetronic acid with 85% enantiomeric excess (ee) was achieved. Notably, the SiO_2_@Ley catalyst outperformed its homogeneous counterpart in terms of enantioselectivity (85% vs. 62% ee). This improvement is possibly due to favorable hydrogen bonding interactions with the SiO_2_ matrix, which influence the geometry of the transition state.

An indisputable advantage of microreactors lies in their suitability for photocatalytic reactions. The thin reactor walls and small dimensions allow for highly efficient use of light energy, contributing to the overall energy efficiency and cost-effectiveness of such systems [336,337,343,356,369,422,423,424,425,426,427,428].

PROS and CONS of MRs:

PROS (high-performance attributes):High intensity of mixing; enhanced mass and heat transport;Increased chemical reaction rates;Efficient application of external energy sources (e.g., light, microwaves, ultrasound);Potential integration with membrane technologies (especially “tube-in-tube” reactors);Precise temperature control;High operational pressure tolerance;High selectivity;High volumetric productivity.

CONS:Fouling and clogging;High pressure drop;Requirement for high-purity reactants;Catalyst deactivation (due to abrasion or leaching);Not suitable for large-scale production (despite high volume-specific productivity).

The list of PROS and CONS suggests that microreactors are best suited for producing specialty chemicals (such as active pharmaceutical ingredients, APIs) and for targeted applications, such as the removal of specific pollutants.

## 5. Design of Catalytic Technologies

Similar to other technical disciplines, chemical research can be divided into three main categories.

Basic Research (BaR, also known as fundamental or pure research)The focus is on expanding knowledge and understanding the essence of phenomena, without aiming at practical applications. In chemistry, this includes hypothesizing, experimentation, analysis, and synthesis of data related to phenomena at the picometric scale (10^−12^ m) and down to the attosecond scale (10^−18^ s), i.e., interactions of atoms, electrons, subatomic particles, and electromagnetic radiation.Applied Research (ApR)The goal is to solve practical, real-world problems. ApR uses insights from BaR and solutions derived by analogy to previously solved issues. In chemistry, ApR involves the characterization of reaction systems (e.g., reaction kinetics, mass and heat transfer, catalysts—see section on catalyst properties), and the study of physical–chemical properties such as enthalpy, rheological behavior, melting/boiling points, electrical conductivity, magnetic susceptibility, and polarizability. It also quantifies interactions among reactants (e.g., deviations from ideal behavior, miscibility). Both experimental and theoretical approaches are used to achieve desired values of relevant parameters.Technological Research (TeR)The aim is to develop new products that meet emerging societal needs or to improve existing production routes. Objectives include enhancing product quality (e.g., enantiomeric purity in APIs), reducing raw material and energy consumption, improving process safety, and minimizing environmental impact. These improvements depend on factors such as catalyst performance (activity, selectivity, productivity, lifetime, regenerability), reactor type, solvent choice, process conditions (pressure, temperature, mixing mode, batch vs. continuous operation), separation units, waste treatment, and synergistic effects (Figure 14). TeR makes full use of BaR and ApR findings but can also stimulate further BaR or require complementary ApR. It commonly involves experimental and mathematical modeling and includes application testing, which can span several years. For example, testing a new fertilizer typically requires at least four years of field trials (excluding the “worst year”) to evaluate efficacy, consistency, and economic viability. For biologically active substances or APIs, the evaluation period can be even longer. TeR must fully adhere to green and sustainable design principles [63,64,66].Chemical technologies can be divided into three main groups [55,228,429].

Large-Scale Technologies (Bulk Chemical Production)
Inorganic (NH_3_, HNO_3_, H_2_SO_4_, etc.);Organic (components for fuels, monomers, polymers, etc.);Biotechnologies (fermentation, enzymatic hydrolysis, etc.).Chemical specialties:
Inorganic (components for production of catalysts, pure chemicals for electronics, special technical products—glasses, etc.);Organic and organic–inorganic mixed (additives to polymers, antidegradants, explosives, dyes, biologically active substances);Biotechnologies (production of API and their precursors, e.g., penicillin, etc.).Waste Treatment (liquid, gas, soil).


**How to choose the Right Catalyst in New Catalytic Technology Design?**


When designing a new catalytic process, one must decide between the following:A relatively inexpensive catalyst with average activity and selectivity, combined with a simple separation system.A more expensive, highly active, and selective catalyst, which may be sensitive to impurities, temperature, and pressure fluctuations. This choice would necessitate high-purity inputs and sophisticated process control systems, increasing costs.

Typically, commercial catalysts are cheaper, more reproducible, and come with manufacturer guarantees. Major producers [430] offer a wide variety of catalysts for most standard reactions. However, novel syntheses often require the development of new catalysts, particularly for temperature-sensitive or chemoselective transformations. Fortunately, catalyst manufacturers can assist with scale-up and may produce the catalyst at larger scales beyond the lab.

The decision between using a commercial or custom-developed catalyst should be informed by Technological Research (TeR), supported by mass and energy balance, subsequent economic analysis, and environmental impact assessments (e.g., separation systems and waste treatment). Considerations include CAPEX and OPEX, energy pricing, catalyst regeneration potential, and disposal of fully deactivated catalysts.

Figure 24 illustrates different views on a chemical process from a picometric level (pm, molecular material) up to metric level (m, chemical plant) [431]. This figure also illustrates areas of BaR (“pm” level), ApR (“µm” and “cm” level), and TeR (“cm” and “m” level).

“A Dream of the Best Catalyst and the Ideal Catalytic Technology” has existed since the very definition of a catalyst itself [6,8,10]. A typical sequence in the development of a new catalytic technology is presented in Figure 25. The chart, adapted from the handbook [6], highlights the numerous interactions and dependencies involved in such development. Inputs to this process include the required amount of target product, technical constraints, safety considerations, and environmental limitations (as described in detail in [6]). Outputs of the process involve the type of converter (reactor), the type, size, and shape of the catalyst, and reactor vessel parameters (e.g., volume). A particularly important input is “catalyst die-off information”, which refers to the catalyst’s lifetime—an essential factor in evaluating the overall economic viability of the process.

Based on sources [6,60,96,431,432], we have prepared a typical chemical engineering diagram resulting in a reactor model, suitable for technological design (right-hand side of Figure 25). Traditionally, development begins with thermodynamic considerations, including reaction equilibria and heat effects. The catalyst selection is usually made by analogy with known reaction types (e.g., hydrogenation, oxidation, alkylation, cyclization), and/or by evaluating the potential of newly prepared, original catalysts. Kinetic analysis is then performed, often in combination with quantum chemical modeling, considering all relevant steps (“true reaction kinetics”, diffusion, chemisorption, conductive heat transport, external mass and heat transport), as described in part 2 of this paper. After theoretical and computational studies, a laboratory reactor is constructed, and catalytic tests—including deactivation monitoring—are conducted. Kinetic model parameters are then estimated. The model is further enhanced with a description of flow/mixing pattern and transport properties of fluid. Ultimately, a comprehensive reactor model is developed. This, combined with separation units, pumps, heat exchangers, and other auxiliary equipment, becomes the basis for modeling the entire catalytic process.

A summary of modern modeling techniques—such as machine learning (ML), artificial neural networks (ANN), ab initio and DFT calculations, solvent interaction simulations, computer-aided molecular design (CAMD), support vector machines (SVM), and others—applied for material description and predicting catalytic performance is provided in [431]. Although the referenced study focuses on separation processes with a special emphasis on zeolite and MOF catalysts, the methodologies are fully adaptable to other catalytic systems.

The choice of catalyst was determined by the type of chemical technology. In the 20th century, solid inorganic materials were the most common catalysts for large-scale petrochemical and chemical processes [6,10,56,100,265,433]. Starting around the 1950s, polymer-based catalysts, particularly for acid-catalyzed reactions, were introduced for processes operating below 120 °C [49,73,264,434,435,436]. From approximately 1980 onward, there was a boom in biocatalysis, with increasing use of pure and modified enzymes [5,21,30,436,437,438,439,440]. Challenges related to enzyme separation have been addressed through immobilization, encapsulation, or integration of enzymes with inorganic and/or organic supports [173,441,442]. As of today (2025), it can be stated that biocatalysis should be considered the first-line option to produce both known and new chemical products. However, in line with the scope of this paper, biocatalytic and enzymatic processes will only be briefly discussed in the following chapter.

In accordance with green chemistry principles and sustainability goals, the next preferred routes should be photocatalytic and electrocatalytic processes—assuming sufficient availability of electrical energy. Several examples were previously provided (Section 4 of this review).

If these options are not economically viable, then a classical heterogeneous catalytic process must be developed. It is worth noting that modern technologies utilizing renewable resources widely apply heterogeneous catalysis. This further highlights the importance of heterogeneous catalysts in contemporary chemical technology, especially when coupled with the design of modern, efficient reactors.

The following are the main stages involved in developing or retrofitting a new technology [228,429]:TeR.Feasibility Study—Includes process principles, process flow diagrams (PFDs), raw material and energy requirements, waste treatment strategies, a preliminary layout, economic evaluation (CAPEX and OPEX), safety and environmental assessments, and comparison with Best Available Technology (BAT).Basic Design (BD)—Covers process principles, PFDs, material and energy balances, equipment selection and integration, control systems, Piping and Instrumentation Diagrams (P&IDs), utility requirements (cooling, water supply, energy), analytics, packaging and transportation of products, waste treatment, preliminary economic and ecological assessments, HAZOP (Hazard and Operability Study), and identification of unresolved issues.Legislation—Involves Environmental Impact Assessment (EIA), REACH (Registration, Evaluation, Authorization, and Restriction of Chemicals), IPPC (Integrated Pollution Prevention and Control), and other applicable regulatory requirements, including those related to hazardous substances and construction permits.Detailed Design (DD)—Provides a more detailed version of all BD elements.Realization—Physical construction and implementation of the technology.Testing—Includes commissioning and validation of the process.Update of DD Documentation—Reflects changes based on real-world implementation.Routine operation.Optimization.Retrofit or Decommissioning of Technology.

Developing a new large-scale technology typically takes 3 to 8 years from the start of TeR to routine operation. This period can be significantly shortened by adopting or purchasing a technology license, including relevant legislative documentation (REACH, HAZOP, data for IPPC). Licensing is common in large-scale processes such as ammonia synthesis, nitric acid production, fertilizer manufacturing, crude oil treatment, pyrolysis, and polymer production.

In heterogeneous catalytic technology development, two approaches are viable.

New technology development (novel catalyst, reactor, separation system).Retrofit (existing or new product, with a new catalyst and adaptation of existing equipment). Retrofits are more cost-effective and quicker, but come with limitations—material, volume, mechanical strength (e.g., maximum pressure and temperature), and constraints related to heating/cooling, mixing, and separation units [228,443,444].

A new large-scale technology typically has a planned lifetime of 8, 12, or 15 years, which is used for depreciation calculations (linear, degressive, or progressive methods). In the manufacture of chemical specialties (produced in tons per year), the operational lifetime can be significantly shorter, often just months, corresponding to the time needed to produce a required amount of product.

The total investment (CAPEX) required for a new design is represented by four main parts.

Battery limits investment (BL, cost_basis = 1)

This refers to the geographic boundary defining the manufacturing area of the process. It includes process equipment and the buildings or structures to house it, but excludes facilities such as boiler houses, site storage, pollution control systems, and general site infrastructure. The cost of specific equipment increases with factors such as size, materials of construction, operating pressure and temperature, internal structure (packing, cooling elements), and flow-control requirements. Typical items include heat exchangers and tanks (including reactors and separation units), piping, valves and fittings, process instrumentation, pumps and compressors, electrical equipment, structural supports, and miscellaneous components.

2.Utility investment (not all items are necessary for every technology) (ca. 0.5*cost_basis)

Electricity generation and distribution, steam generation and distribution, process water, cooling water, firewater, effluent treatment, refrigeration, compressed air, and inert gas (most commonly nitrogen).

3.Off-site investment (not all items are necessary for every technology) (ca. 0.2*cost_basis)

Auxiliary buildings (offices, medical facilities, personnel rooms, guardhouses, warehouses, and maintenance shops), roads, rail connections, fire protection systems, communication systems, waste disposal facilities, product storage, water and fuel supplies not directly connected to the process, plant service vehicles, loading and weighing stations.

4.Working capital (ca. 0.7*cost_basis)

This includes raw materials for plant start-up (including treatment of waste during start-up), inventories of raw materials, intermediates, and products, transportation costs for start-up materials, and the funds needed to carry accounts receivable (credit to customers) minus accounts payable (credit from suppliers), as well as payroll during start-up.

The most critical item for estimation is the Battery Limits (BL) cost, which consists of the price of process equipment plus costs for installation, piping and valves, control systems, foundations, structures, insulation, fireproofing, electrical work, painting, engineering fees, and contingency. These additional items typically add around 150% to the equipment cost. The starting point for BL estimation is the equipment price. Some software, such as ASPEN, contains databases with price data. If such databases are unavailable, published sources such as Smith [228] can be used. To update historical price data to current values, economic indexes (EC) are applied—these are factors greater than 1 used to adjust older prices. Examples include indexes for Construction and Labor, Buildings, and Engineering and Supervision. Values can be found in the book by Smith, or in the Chemical Engineering literature, most notably the Chemical Engineering Plant Cost Index (CEPCI).

A simplified approach sometimes estimates CAPEX as five times the equipment cost. However, in retrofit cases, the CAPEX is significantly lower due to the use of existing equipment and infrastructure and can even approach zero. This has been demonstrated in the R&D of cyclohexylamine (CHA) production from aniline over a Ru/C catalyst. Over approximately eight months of laboratory research, plant testing, and verification of product quality and stability, a production scale of several thousand tons of CHA per year was achieved with nearly 100% selectivity and very low ruthenium catalyst consumption (Kralik et al., unpublished results). Similarly, low CAPEX is typical in the R&D of fine chemicals and chemical specialties when existing reactors and infrastructure are utilized.

Operating costs or Operating Expenses (OPEX) are directly related to production volume and divided into five categories:Raw materials;Catalysts and chemicals consumed in manufacturing (other than raw materials);Utility costs: fuel, electricity, steam, cooling water, refrigeration, compressed air, inert gas;Labor (about 5% of total expenses for batch processes, about 1% for continuous processes);Maintenance (usually ~6% of CAPEX).

Similar to CAPEX, it should be emphasized that estimating OPEX is much easier when existing reactors, separators, infrastructure, including operation staff can be utilized.

For OPEX estimation, researchers should provide data on catalyst price, activity, selectivity, lifetime (productivity), regeneration cost, and disposal cost for spent catalysts. Energy requirements are determined based on material and enthalpy balances.

Fixed costs (FC) are independent of production rate and include the following:Capital cost repayments (depreciation; typical lifetimes for new technology are 8, 12, or 15 years, used for linear, degressive, or progressive depreciation calculations);Routine maintenance;Overheads (e.g., safety, laboratories, personnel facilities, administration);Quality control;Local taxes;Labor unrelated to production (e.g., security);Insurance.

The simplest estimate of economic performance is Economic Potential (EP):

EP = <Value of products> − <Fixed costs> − <Variable costs> − <Taxes>(24)


Researchers should note that material prices for laboratory supplies are generally higher than those for industrial bulk purchases. Bulk chemical prices for large-scale quantities can often be found online or obtained upon request.

If EP is positive, two common evaluation criteria are most frequently applied.

Payback Time (PaybTi): The time needed to recover the capital investment from average annual cash flow. For retrofits, it is the time needed to recover the retrofit cost through improved operating savings. PaybTi should be significantly shorter than the technology’s planned lifetime (usually not exceeding 80% of it).Return on Investment (ROI): The ratio of average annual income over the productive life of the project to the total initial investment, expressed as a percentage.

When comparing alternative technologies, the best choice is generally the one with the shortest PaybTi and the highest ROI.

While this economic evaluation may appear complex and data-intensive, our teaching experience in *Process Design* shows that even spreadsheet-based models (e.g., Excel) can provide reliable results. It is worth noting that accurate data for calculating CAPEX and OPEX depend heavily on geographic, demographic, export, taxation, and other local and global factors. The impact of parameters on the economics of technology is evaluated using parametric studies, i.e., how the value of a given parameter, such as the price of a product, while fixing the value of the remaining parameters, affects a selected economic parameter, most often PaybTi.

Dissemination of results typically decreases from Basic Research (BaR) and Applied Research (ApR) to TeR, with TeR results having high commercial value. If possible and without revealing sensitive details, new technologies are protected via patents. However, the core value lies in the know-how, which increases substantially once technology is successfully realized (commercialized). This value is quantified in license pricing, including royalties on relevant patents. Most major catalyst producers [430] offer technology licenses bundled with their catalysts (see, e.g., [445]). Selected results from successful implementations are published to inform the scientific community and promote the involved company [446,447].

Accurate estimation of reaction kinetics, mass/heat transfer phenomena, reactor modeling, and process optimization requires robust computational tools. For simple tasks (e.g., analysis of differential reactor data), tools such as Excel and Origin are sufficient. For complex systems involving detailed kinetics, mass and heat transfer, and catalyst deactivation (especially under dynamic conditions), custom programs are required. Languages such as FORTRAN, ALGOL, Pascal, C, C++, and modern alternatives like Python [448] are used. Examples of FORTRAN-based modeling efforts by the Slovak University of Technology are documented in [449,450,451,452], including optimization of an industrial bubble column slurry reactor for reductive alkylation of 4-diphenylamine (used in antidegradant production) [453]. A successful example of a “home-made program system” developed for data analysis and reactor modeling is ModEst [60,454,455,456]. It is written in FORTRAN and C++ and regularly updated. ModEst is available (with permission) to researchers and students at Åbo Akademi University (Finland) [457]. For modeling of flow systems and microreactors, COMSOL Multiphysics [240] is among the best available tools [371,378].

The WEBPAGE [458] contains names and URL addresses to 98 chemical process simulators. Among the most commonly used are: ASPEN [459,460], MATLAB [461], and gPROMS [462,463,464]. gPROMS is notable for offering a free basic version for researchers and designers. Based on our experience, ASPEN is excellent for process design but offers only basic reactor models (typically equilibrium or conversion models). A drawback is its limited database of physical–chemical properties for complex molecules, although users can manually add missing data easily [459]. The full ASPEN version includes a database of equipment prices, enabling direct linkage of CAPEX calculations with process simulations. To aid model convergence (especially with multiple recycles), a preliminary material balance in Excel is recommended.

Basic and Detailed Design (BD and DD) are typically developed using AutoCAD, which supports 3D visualization with multiple levels (foundations, floors, apparatuses, piping, insulation, control systems, cabling, walkways, walls, etc.) [465]. 3D structured AUTOCAD models are also used for training operators and searching for defects appearing during the exploitation of realized technology. Updating project documentation post-implementation is essential. Integration of ASPEN and other simulation tools enables more reliable Failure Mode and Effects Analysis (FMEA). An automated workflow system is vital for successful chemical R&D [466].

Although BD and DD are often produced by professional engineering firms (preferably through Engineering, Procurement, and Construction—EPC contracts), collaboration with catalytic researchers ensures better data, minimized risk, and potentially reduced costs. Essential information includes the following (as detailed in [228,429]):Brief description of technology;Block flow diagram (BFD), i.e., definition of main technological units linked with streams, including recycles;If possible, comparison with best available technology (BAT);Raw materials and energy demands;Catalyst and type of reactor (see above);Kinetics;Lifetime, regeneration, and disposal of deactivated catalyst;Preliminary calculation of reactor volume;Range of temperature and pressure for operation;Recommendations for separation units;Physical–chemical, explosive, and hazardous/toxic properties of individual components and their mixture;Characterization of wastes;Preliminary economic, safety, ecological, and technological risk analysis.

When designing a new process for a known or similar product (e.g., fatty acid methyl esters), all technological alternatives should be evaluated, including homogeneous, heterogeneous, and electrochemical routes [467]. Although the “RoadMap on Catalysis for Europe” [468] is over a decade old, it remains a valuable resource for insights into catalytic chemistry and emerging catalytic materials.

## 6. Examples of Technologies

Most papers dealing with catalysts include notes on potential technological applications. However, as shown by the analysis of the SCOPUS database presented in the Introduction of this paper, advancement to the level of production technology is rather rare. Therefore, the focus of this chapter is on classical technologies—such as ammonium, fluid catalytic cracking, methanol synthesis, alkyl-tert-butyl ethers, and aniline production—that are supported by comprehensive catalytic, chemical engineering, and technological research. The characteristics of these established technologies can serve as inspiration for the development of new, modern, and green technologies. The last two subchapters are dedicated to emerging technologies: water splitting and catalytic treatment of plastic waste.

### 6.1. Ammonia

Synthesis of ammonia from nitrogen and hydrogen is probably the most intensively studied catalytic process [6,447,460,469,470,471,472,473,474]. It is remarkable that in the early stages of ammonia synthesis (1918–1925), more than 3000 laboratory experiments were conducted to develop an optimal iron oxide catalyst doped with alkali metals. This catalyst, essentially Fe/Al_2_O_3_/CaO/K, remains in use in modern ammonia synthesis. The current global production of ammonia is approximately 240 million tons (Mt), with projections of 400 Mt by 2040 and 650 Mt by 2050 [475]). Despite the fact that over 95% of ammonia is currently produced from fossil-based feedstocks (primarily natural gas), it is sometimes referred to as a “green product” [473], as over 80% is used in fertilizer production, and plants absorb CO_2_ during growth. The term “green ammonia” is fully justified when green hydrogen is used—i.e., hydrogen produced without CO_2_ emissions [475,476]. Moreover, ammonia is gaining attention as an energy carrier, particularly for solid oxide fuel cells [476,477].

Regardless of the hydrogen and nitrogen sources used for synthesis gas (molar ratio N_2_:H_2_ = 1:3), the overall reaction is:N_2_ + 3H_2_ ⇆ 2NH_3_;      Δ_r_H_298K_ = −46 kJ mol_NH3_^−1^(25)

As shown in Figure 26, the equilibrium concentration of ammonia increases with higher pressure, lower temperature, and a reduced content of inert gases (Le Chatelier’s principle). However, the reaction rate is rather low below 400 °C, while high pressure necessitates more expensive reactor construction (e.g., thicker walls). Therefore, industrial ammonia synthesis is typically conducted at 400–550 °C and 15–50 MPa. The negative impact of higher temperature on equilibrium is mitigated by special reactor designs and autothermal operation modes [6,476]. Ammonia reactors are large and complex. Catalysts are arranged in sections (beds or tubes ~0.1 m in diameter), with heat exchange occurring before gas enters each section. Modern commercial reactors, such as those using Haldor Topsoe’s SynCOR Ammonia™ technology, can produce 1300–3500 tons of NH_3_ per day.

Industrial iron-based catalysts (e.g., Fe/Al_2_O_3_/CaO doped with K, Li, Mg, Mo, W, and rare earth elements) typically consist of irregular particles (1.0–1.2 mm) [6] or hollow extrudates (Figure 3). After reduction (activation), these particles contain nanocrystalline iron domains (20–60 nm), believed to be the active sites for ammonia synthesis. Catalyst deactivation is primarily caused by sintering of these crystallites, a process that accelerates with increasing temperature [478]. This is a key reason why the synthesis is carried out at the lowest practical temperature.

Per-pass conversion of reactants is typically around 20%. Thus, after separation of liquid ammonia, unreacted synthesis gas is recycled, as shown in the block flow diagram (Figure 27). The need for high-pressure operation and gas recycling/compression, combined with hydrogen production via methane steam reforming (26), makes ammonia synthesis a highly energy-intensive process.CH_4_ + 2H_2_O → CO_2_ + 4H_2_;     Δ_r_H_1000K_ = 47 kJ mol_H2_^−1^(26)

*Note:* The reforming of methane is carried out over a nickel catalyst at around 1000 K. Therefore, the reaction enthalpy was calculated for this temperature using data from [479].

The energy demand for ammonia production in 1925 was approximately 50 GJ per ton of NH_3_. Current Best Available Technologies (BATs) have reduced this to about 28 GJ per ton of NH_3_. Nevertheless, given the global ammonia production of 240 million tons, this still represents around 1% of total global energy consumption. Moreover, for every 1 ton of ammonia produced, approximately 1.7 tons of CO_2_ are emitted, contributing significantly to the greenhouse effect. A breakthrough comes with the use of green hydrogen produced via Solid Oxide Electrolysis Cells (SOEC). In this case, the energy required for ammonia synthesis drops to around 1.6 GJ per ton of NH_3_ (excluding the energy required for electrolysis), which is only about 6% of the energy demand in the conventional Haber–Bosch process based on CH_4_ reforming [476]. Additionally, no CO_2_ is produced. Excluding the hydrogen production units (the first two blocks in Figure 27), both CAPEX and OPEX are significantly reduced. A detailed economic analysis is provided in [480].

It may seem that after more than 100 years of R&D, the “catalyst story” in ammonia technology is complete. This is partly true when it comes to optimizing existing iron-based catalysts within advanced reactor designs. The fine-tuning of catalyst properties (e.g., modifiers, particle size, and shape) is largely managed by technology providers. However, the availability of high-purity hydrogen from electrolysis opens opportunities to develop new catalytic systems based on more active transition metals (e.g., Ru, Co, and rare earth elements), offering potential for novel process configurations [470,474]. These new systems can operate at much lower pressures (0.1–3 MPa), drastically reducing the energy needed for recycling unreacted synthesis gas. This leads to a significant reduction in overall costs (CAPEX and OPEX). Some “exotic” processes, such as thermal photocatalysis (e.g., Ru/C at 350 °C and 0.1 MPa, where deactivation is virtually eliminated by photon activation [481]), may gain interest, though mainly for small-scale production.

### 6.2. Fluid Catalytic Cracking (FCC)

Fluid Catalytic Cracking (FCC) aims to convert heavier hydrocarbons into lighter fractions (C_4_–C_10_) suitable for gasoline. The feedstock is usually Vacuum Gas Oil (VGO), consisting predominantly of hydrocarbons in the C_20_–C_35_ range, though it can include various heavy fractions, including those derived from coal and lignite processing [265]. The total global FCC capacity is approximately 1.09 billion tons per year, accounting for about 18% of global crude oil consumption (6 billion tons per year, as of 2024). These figures, combined with forecasts that crude oil exploitation will continue for the next 30–40 years, underline the strategic importance of FCC technology.

The basic cracking reaction can be described as:R^1^-CH_2_-CH_2_-CH_2_-R^2^ ⇆ R^1^-CH_3_ + R^2^-CH=CH_2_(27)

Cracking is catalyzed by acid catalysts with Lewis and Brønsted acid centers [265,482]. Alongside the main reaction (27), a wide variety of isomerization, cyclization, and condensation reactions occur [482,483]. In addition to desired products, heavier molecules—and even coke—are formed, leading to rapid catalyst deactivation.

The first FCC process was commercialized in 1915, using AlCl_3_ as a homogeneous catalyst. However, high costs and environmental challenges related to catalyst disposal limited its adoption. In the 1930s, Houdry demonstrated that Fuller’s earth, a clay-based aluminosilicate material, could catalyze the conversion of lignite-derived oil to gasoline. By 1933, Houdry’s small batch unit processed about 200 barrels/day (32 m^3^/day). This technology evolved further in 1941. The real leap came in 1959, with the synthesis of zeolites (by Mobil Co.). The introduction of rare earth-exchanged Y-zeolite catalysts revolutionized FCC by significantly improving gasoline yield and reducing the formation of coke and dry gas (C_1_–C_3_). However, their strong hydrogen-transfer ability reduced the yield of light olefins and the octane number of gasoline.

Due to rapid catalyst deactivation (within 5–60 s), FCC is fundamentally different from fixed-bed processes like ammonia synthesis. Continuous catalyst regeneration is necessary [265,483]. Moreover, catalyst particles are subject to abrasion in the fluidized environments of both the reactor and regenerator (Figure 28, left). Therefore, a high density of active sites throughout the catalyst volume is essential. Because of the large size of reactant molecules, reactions primarily occur on the external surface, favoring small catalyst particles with high external surface area.

To ensure proper fluidization and flowability, the particle size must be optimized—typically around 0.1 mm. Fluidization is achieved by the flow of lighter hydrocarbon products and steam. Proper particle sizing is also critical to allow movement in areas with reduced fluid velocity (Figure 28, left). The right side of Figure 28 shows typical components of an FCC catalyst particle.

FCC reactors operate at 470–580 °C and 0.7–1.5 bar. Lower pressure promotes lighter product formation but may reduce fluidization efficiency. The regenerator operates at higher temperatures—up to 800 °C, which is the thermal stability limit of zeolite catalysts. Its pressure is slightly higher than in the reactor to facilitate the flow of regenerated catalyst.

Modern FCC research focuses on enhancing catalyst stability (resistance to deactivation and abrasion), optimizing product selectivity, and reducing energy consumption [483,484,485]. Hierarchical zeolites, which offer a higher external surface area, are promising candidates for more efficient FCC catalysts. However, their lower thermal stability compared to conventional zeolites must be addressed [486].

### 6.3. Methanol

Methanol production in 2024 exceeded 100 million tons, placing it among the most important bulk chemicals. Although methanol is an established compound, its applications continue to expand. Major uses include the production of formaldehyde, acetic acid, plastics, methanol-to-olefins (MTO), direct use as a fuel or fuel additive, synthesis of methyl esters as fuel components, and potential use in fuel cells [55]. The primary (desired) reaction for methanol synthesis is:CO + 2H_2_ ⇆ CH_3_OH,     Δ_r_H_298K_ = −90.8 kJ mol_CH3OH_^−1^(28)

The influence of temperature and pressure on equilibrium composition is depicted in Figure 29. The reaction (28) is accompanied by side reactions that produce dimethyl ether, water, methane, and other byproducts, which reduce the methanol yield.

In the earliest methanol synthesis processes, ZnO catalysts were used under high-pressure conditions (Figure 29), but these catalysts deactivated quickly. After 1960, more effective copper-based catalysts (Cu/ZnO/Al_2_O_3_) were developed for lower-pressure processes (modern processes). The catalytic centers are believed to be copper nanocrystallites. Initially, the catalyst was used in powder form (~0.1 mm), but currently, cylindrical pellets about 5 mm in diameter are common. The process is carried out in fixed-bed or tubular reactors, with input conditions optimized to maximize the reaction rate.

Under optimal conditions (maximum temperature 550 K, maximum pressure 10 MPa, and sulfur compounds < 10 ppm—as they act as deactivators), the average catalyst lifetime is around five years.

A sustainable alternative starts with CO_2_ hydrogenation:CO_2_ + 3H_2_ ⇆ CH_3_OH + H_2_O,     Δ_r_H_1000K_ = −49.6 kJ mol_CH3OH_^−1^(29)

From a technological point of view, it is important to highlight the much higher thermodynamic stability of CO_2_ compared to CO. The Gibbs free energy of formation (ΔfG_298_K) is −394.4 kJ/mol for CO_2_ and −135.6 kJ/mol for CO [479]. This greater stability means that CO_2_ conversion is significantly lower, especially at pressures below 5 MPa—only about 1/10 the conversion rate of CO [98]. This high stability also results in more difficult activation (chemisorption) of CO_2_, leading to slower reaction kinetics compared to CO.

Similar catalysts used for CO hydrogenation can be employed for the exploitation of CO_2_, but their activity is lower, and deactivation occurs faster. Better results have been achieved with more expensive systems, such as In_2_O_3_/ZrO_2_, which showed stable activity over 1000 h [487]. The referenced paper also summarizes results with other catalysts, including noble metals. MoS_2_-based catalysts [488] show other challenges; however, acceptable selectivity (above 90%) was obtained only at low conversion. Given these challenges, many researchers [487,488,489,490,491,492] have revisited the well-known concept of first converting CO_2_ to CO, followed by CO hydrogenation to CH_3_OH. This can be conducted in a closed reactor system to minimize CO toxicity risk. The two-step process requires different conditions (catalysts, temperature, pressure, residence time) and necessitates separation of water formed during CO_2_ hydrogenation. Alternative approaches are also under investigation, including biological [493], photochemical [494], and electrochemical synthesis pathways [495]. A comprehensive overview of green methanol technologies can be found in [496]. Despite the commercial maturity of green hydrogen production and effective CO_2_ capture from flue gases and the atmosphere [497], the direct conversion of CO_2_ with hydrogen into methanol remains an ongoing challenge requiring further research to develop economically viable and scalable technologies.

### 6.4. Alkyl Tert-Butyl Ethers (ATBE)

ATBEs are important octane boosters for gasoline. Their Research Octane Numbers (RONs) are as follows: methyl tert-butyl ether (MTBE): 120, ethyl tert-butyl ether (ETBE): 117, and *tert*-amyl methyl ether (TAME): 114. In addition to improving octane ratings, the use of ATBEs helps reduce the concentration of harmful compounds in exhaust gases. Global production of ATBEs exceeds 35 million tons. As of 2025, ETBE is the most widely used, largely due to its production from bioethanol.

The general reaction is:R-OH + HC=C(CH_3_)_2_ ⇆ R-O-C(CH_3_)_3_(30)

This reaction is acid-catalyzed. Both inorganic and organic heterogeneous acid catalysts have been tested [498,499], with the most effective being resin-based catalysts, particularly macroreticular sulfonated poly(styrene-co-divinylbenzene). These catalysts are typically formed into beads with diameters of 300–1000 μm, packed in a catalyst bed. The synthesis is usually conducted in the liquid phase or combined with reactive distillation. In liquid-phase processes, the typical operating pressure is 1.5–2 MPa, and the temperature ranges from 60 to 90 °C. Catalyst lifetime is typically six months to one year. While higher temperatures increase the reaction rate, they also accelerate catalyst deactivation, mainly due to the cleavage of –SO_3_H groups from the polymer backbone.

### 6.5. Aniline (AN)

AN is one of the oldest products of organic chemical technology. Its annual production is currently around 6 million tons and is expected to reach approximately 15 million tons by 2032. A brief history and overview of various preparation routes can be found in [500]. Industrial production is primarily based on the hydrogenation of nitrobenzene (Figure 30):

Transition metals serve as catalysts for this process, with nickel–copper, palladium, platinum, and their combinations proving most effective. Hydrogenation can be carried out in the gas phase or in a gas–liquid (G–L) system, using either tube reactors or slurry reactors [500]. A Pd–Pt–Fe powder catalyst [501], used in a plug-flow reactor, offers a good balance between catalyst stability and productivity.

Utilizing the reaction heat is essential for the economic efficiency of the process. On the one hand, higher temperatures increase reaction rates; on the other hand, they also promote side reactions. The desired reaction pathway involves intermediates such as nitrosobenzene and N-phenylhydroxylamine, leading to aniline. Undesired by-products include azoxybenzene, azobenzene, and hydrazobenzene, which can also be hydrogenated to aniline, but only under more severe conditions. Further discussion of reaction mechanisms is available in [502,503,504].

Catalytic hydrogenation of nitro compounds typically suffers from catalyst deactivation, mainly due to oxidation of metal centers and formation of amine–metal complexes:M(0) + R-NO_2_ + H_2_O ⇆ M(II) + R-NO + 2HO^−^(31)M(II) + n R-NH_2_ ⇆ [M(R-NH_2_)n]^2+^(32)

These complexes can be reduced back to Pd(0) with hydrogen (Figure 31), though this often leads to the formation of larger, less active crystallites, as confirmed by X-ray powder diffraction (XRPD).

Optimal process design requires a stable catalyst (e.g., Pd–Pt–Fe) and minimization of nitrobenzene concentration in contact with the catalyst. This can be achieved by intensive axial mixing at the bottom of the reactor and recycling (Figure 32). In the upper part of the reactor, a piston–axial dispersion flow allows for near-complete conversion of NB. Before entering the hydrogenation reactor, nitrobenzene is used to extract aniline from the wastewater stream (denoted as W, AN). Metal losses are typically below 0.5 g per ton of AN. More details on the process are available in [505].

Alternatives to the outlined AN technology are investigated as follows:Photocatalytic Reduction of Nitrobenzene to Aniline by an Intriguing-Based Heteropolytungstate [506]. Photocatalytic hydrogenation of nitrobenzene to aniline over titanium(iv) oxide using various saccharides instead of hydrogen gas [507].Highly Selective Electroreduction of Nitrobenzene to Aniline by Co-Doped 1T-MoS2 [508].Direct Amination of benzene [509].Ammonolysis of phenol [510,511].

However, none of these alternatives are currently competitive with the well-established process based on benzene nitration followed by hydrogenation of NB to aniline using transition metal catalysts.

A peculiar attention has been given to bio-aniline, developed by Covestro [512,513] (Figure 33). The technology includes the following steps [512]:Fermentation of sugars in the presence of ammonia to form ammonium 2-aminobenzoate (NH_4_–OAB);Thermal and/or catalytic decarboxylation NH4-OAB to aniline;Extraction of the resulting aniline using a suitable solvent (e.g., dodecanol);Separation of aniline via rectification and solvent recycling;Separation and recycling of ammonia.

In 2024, Covestro built a pilot plant for this technology in Leverkusen, Germany [514].

### 6.6. Photocatalytic Water Splitting

Photocatalytic water splitting has become a prominent area of research focused on lowering the cost of hydrogen production. Unlike conventional methods that rely on organic feedstocks such as natural gas and crude oil, or electrolysis, photocatalytic water splitting requires simpler equipment. Figure 34 illustrates an array of photocatalytic reactors installed at the Kakioka Education and Research Facility of the University of Tokyo, consisting of 3 m^2^ units [515]. The setup included 1600 reactor panels, each equipped with a 0.0625 m^2^ photocatalytic sheet. Reactant water and oxyhydrogen gas were transported through plastic tubing made from fluoropolymer, polyurethane, or polyamide, with an inner diameter of up to 8 mm.

Water consumption, including losses due to evaporation, averaged only a few liters per day. Deionized water from the municipal supply was fed into the photoreactors once daily using small diaphragm pumps, which also intermittently removed the produced oxyhydrogen gas. To maintain stable pressure, a gas diaphragm pump was installed at the outlet of the entire 100 m^2^ photoreactor array. The reactors operated continuously for two years, employing an aluminum-doped SrTiO_3_ photocatalyst. Under an average irradiance of 0.88 kW·m^−2^ over 30 min, the system achieved a solar-to-hydrogen (STH) conversion efficiency of 0.76%.

During winter, degradation of the photocatalyst sheets was observed, causing catalyst particles to settle at the bottom of the reactors and leading to a drop in STH efficiency. The challenges of separating oxygen from hydrogen and safely storing hydrogen remain significant. Therefore, it is advisable to pair hydrogen production with processes that consume the hydrogen as it is produced. Additionally, the capture of CO_2_ and its conversion into valuable chemicals using hydrogen merits special consideration [515].

### 6.7. Catalysts in Treatment of Waste Plastics

The use of catalysts for waste treatment is another prominent research area in green chemistry [516,517] and the circular economy [518,519]. Catalytic depolymerization is likely the most promising technique for processing waste plastics. It involves moderate capital costs, which vary depending on the catalyst and reactor design, while operational expenses are influenced by catalyst regeneration and energy consumption. Although this method generally requires pre-sorted plastics, it produces high-value monomers at a cost of USD 400–USD 600 per ton [519].

Enzymatic recycling, applicable to specific plastics such as PET, entails low to moderate capital costs for bioreactor installation and moderate operational costs related to enzyme production and reuse. This process yields high-purity monomers at a cost of USD 300–USD 500 per ton. Plasma gasification, on the other hand, involves very high capital costs due to its sophisticated technology and gas cleaning systems, as well as high operational costs due to its energy-intensive nature. This method’s ability to process various waste types and generate syngas results in costs ranging from USD 700 to USD 1000 per ton.

For catalytic treatment of polymer waste, inexpensive and regenerable catalysts are essential. Unfortunately, catalysts based on platinum and ruthenium (e.g., mSiO_2_/Pt/SiO, Pt/SrTiO_3_, Ru/Al_2_O_3_, Ru/CeO_2_, Ru/SiO_2_, and Ru/TiO_2_) exhibit significantly higher efficiency than those based on iron or tin [520]. More affordable catalysts are used in photolysis, such as Ni-TiO_2_-Al_2_O_3_ for decomposing LDPE [519]. Kiln-type reactors are typically employed for the depolymerization of melted polyolefins. Catalyst regeneration primarily involves oxidation of the spent catalyst, followed by calcination and, in some cases, hydrotreatment [521].

## 7. Conclusions

In this comprehensive review, various facets of R&D_HeCaTe have been addressed. The following general guidelines can support the development of new (not licensed) technologies:Prioritize biotechnological methods (see examples in [522]).Utilize electrochemical processes when affordable electricity is available.Apply photocatalytic techniques for waste treatment and the production of chemical specialties.When opting for a heterogeneous catalyst, verify its availability from commercial suppliers.Catalyst development should follow thorough physicochemical analysis, including molecular modeling (using appropriate software), diffusion properties, thermal and chemical stability, toxicity, cost of preparation chemicals, ecological impact, regeneration potential, disposal methods, and reproducibility. Adhere strictly to principles of green and sustainable chemistry [63,66,523].For supported catalysts, leverage the advantages of polymer-based catalysts—such as higher diffusivity, functional groups, and diverse interactions—when the reaction temperature can be maintained below 200 °C [49,114].When inorganic, temperature-stable supports are necessary, focus on regenerable materials such as oxides, aluminosilicates (zeolites), and perovskites [100,524]. Additionally, explore hierarchically structured materials [160,486].Use commercial laboratory reactors or design custom ones for catalytic testing. Conduct a sufficient number of experiments to evaluate activity, selectivity, temperature effects on performance, and catalyst lifetime.Develop a robust physicochemical model of the reactor suitable for scale-up, employing appropriate software tools.Favor continuous reactors over batch reactors whenever feasible.Select the appropriate reactor type. Fixed-bed reactors are preferred initially, but mass and heat transfer limitations may necessitate tube reactors. For smaller catalyst particles (<1 mm), fluidized or slurry reactors might be required [86,269].For low-scale production (up to 1 kg/h), consider the use of flow microreactors.In exothermic reactions, design for the highest possible operating temperature to efficiently utilize generated heat while accounting for material stability and product selectivity.Address the trade-off between reaction rate and equilibrium composition by optimizing temperature regimes and feed locations. Consider autothermal reactors (e.g., those used in ammonia synthesis [6]).Compile data on the physicochemical hazards and toxicities of individual components and mixtures (e.g., explosion limits).If application tests are required (for new polymers/composites, biologically active compounds, polymer additives, fertilizers, etc.), propose test protocols and estimate the necessary sample quantities.Design principal technological schemes and apparatuses, focusing primarily on reactors and separators [228,429].Prepare detailed Block Flow Diagrams.Estimate raw material and energy consumption, along with waste generation.Assess economic feasibility.Identify and evaluate risks: economic, safety, ecological, and technological.Define open questions and uncertainties.Recommend pilot plant implementation when necessary to resolve open questions and minimize risks.Collaborate on feasibility studies and basic design phases.Participate in large-scale technology testing.Engage in the optimization of large-scale processes.Collect and maintain data for the future development of similar technologies.

These guidelines provide a solid foundation for technology developers and contribute to building a valuable database for ongoing and future research. Consultation with specialists is essential on certain topics, and additional research may be necessary during basic design. Moreover, marketing requirements may call for detailed data to ensure successful product commercialization.

A significant advantage for modern R&D_HeCaTe is the access to extensive databases, artificial intelligence tools [525], powerful software platforms [458], and automated experimental systems [526]. These resources accelerate research and development activities and enhance design reliability. Nevertheless, comprehensive knowledge of catalysis, kinetics, and mass and heat transfer remains fundamental to success. Beyond technical capabilities and individual expertise, effective teamwork is equally critical for progress. Each of the 27 steps produces one or more reports, making a robust workflow system essential to keep the project agenda well-organized and on track.

It should be noted that the scope of work outlined in points 1–27 often exceeds—by a factor of ten or more—the effort typically required for the investigation of new catalytic systems and the preparation of a high-quality scientific publication. This increased workload is not offset by a corresponding increase in the number of publications, which, along with citations, remain the primary metrics for evaluating a researcher’s performance. Moreover, if the developed technology is not implemented, the potential economic benefit is often minimal or entirely absent. These contradictions contribute to the very limited number of high-quality publications that could otherwise serve as foundational references for further technological development. In many cases, meaningful results are only published after years of operating the new technology, typically in books or comprehensive reports.

Therefore, it is appropriate to reiterate the premise introduced at the beginning of this article. There is a need to find a balanced compromise between conducting complex catalytic research, pursuing opportunities for publication and patenting, and engaging in the practical implementation of new catalytic technologies.

## Figures and Tables

**Figure 2 molecules-30-03279-f002:**
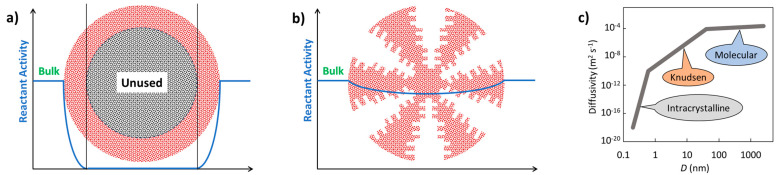
Effect of a catalyst type on its performance; hindrance by external mass transport is neglected: (**a**) activity profile in a microporous catalyst with indication of the unused interior space, (**b**) activity profile in a structured microporous catalyst, (**c**) a dependence of diffusion coefficient on the pore dimension (*D*) (prepared from data in [88]).

**Figure 3 molecules-30-03279-f003:**
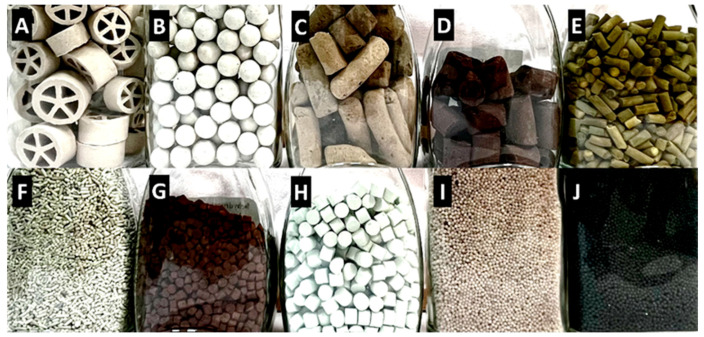
Pictures of commercial supports and catalysts: (**A**): Alumina catalytic supports, also suitable for distribution of gases and liquids, size 12–25 mm; (**B**): Ceramic spheric support, size 12.5 mm; (**C**): NiO-CuO/(Al_2_O_3_,SiO_2_) extrudates 13 × 15–30 mm, suitable for splitting of methane; (**D**): Iron catalyst with promotors, hollow extrudates 12 × 20–30 mm (hole: 4 mm), synthesis of ammonia; (**E**): V_2_O_5_/Al_2_O_3_ extrudates, 6 mm, removal of NOx from tail gas in the production of HNO_3_; (**F**): Mo-Ni-P/Al_2_O_3_ extrudates, 1 × 1–3 mm, hydrotreatment; (**G**): Fe_2_O_3_-Cr_2_O_3_-K_2_CO_3_, tablets 5 × 5 mm, dehydrogenation; (**H**): WO_3_-NiO/(Al_2_O_3_,SiO_2_) tablets 8 × 8 mm, desulfurization by hydrotreatment, hydrocracking; (**I**): Pt/(Al_2_O_3_, Al(OH)_x_Cl_(3−x)_), granules 3 mm, reforming, regenerated catalyst, (**J**): Pt/(Al_2_O_3_, Al(OH)_x_Cl_(3−x)_), granules 3 mm, reforming, used (deactivated) catalyst, covered by carbon species. (The photos are from the collection of catalysts of Prof. P. Hudec—STU in Bratislava).

**Figure 4 molecules-30-03279-f004:**
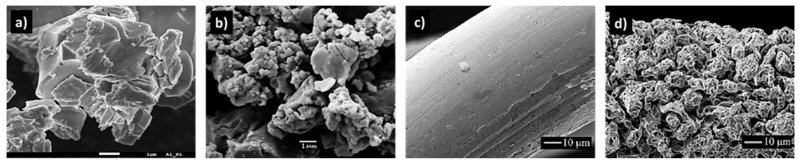
Development of skeletal catalyst structure: (**a**) SEM picture of the starting commercial Raney Al-Ni alloy; (**b**) cubic Ni particles after treatment of Al-Ni alloy with a water solution of NaOH/NaBH_4_ [97]; (**c**) SEM images of unused gauze wire (Pt-Pd-Rh-Ru alloy); (**d**) frontal side of used gauze (2000 h, 10% NH_3_ in air, 1133 K)—a continuous etched layer of “cauliflowers” with the size of ca. 10 μm [98]. Reproduced from ref. [97], 2022 with permission from the MDPI AG—Figure (**a**,**b**) and from ref. [98], 2021with permission from Elsevier—Figure (**c**,**d**).

**Figure 5 molecules-30-03279-f005:**
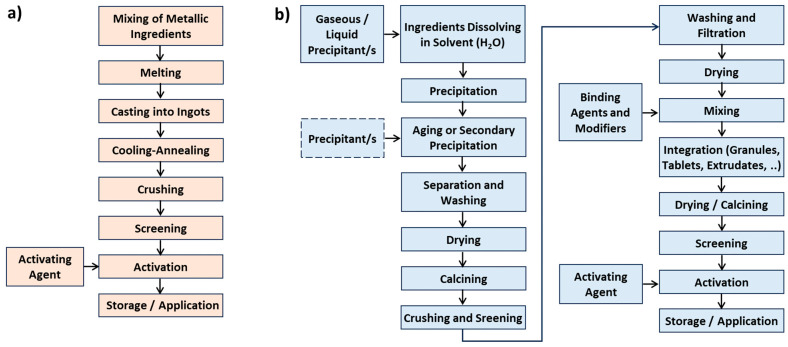
Main steps in the preparation of unsupported catalysts; (**a**) from metals and alloys; (**b**) by precipitation. The dashed block shows the possibility of adding another precipitant/s. It is not necessary in all procedures. (Images were prepared using knowledge from [6,95]).

**Figure 6 molecules-30-03279-f006:**
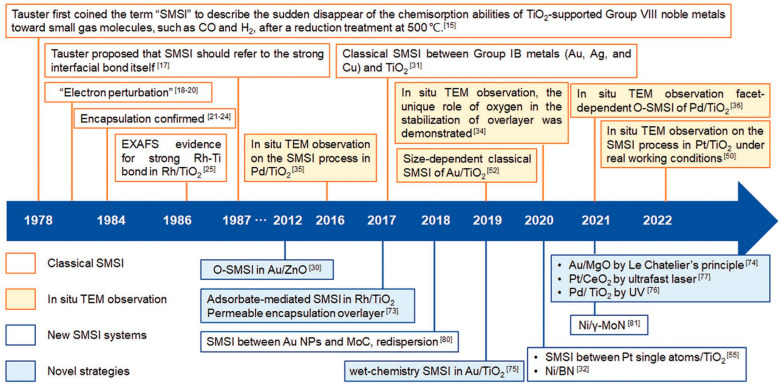
A timeline of representative achievements in the field of SMSI (reproduced from ref. [118] with permission from the Wiley-VCH GmbH). The numbers in square brackets indicate the references that are cited in [118].

**Figure 7 molecules-30-03279-f007:**
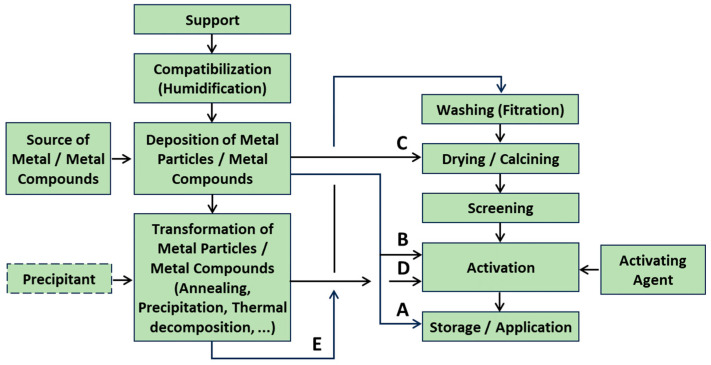
Preparation of supported catalysts. The dashed block shows the possibility of adding another precipitant/s. It is not necessary in all procedures. Routes A–E are discussed in the text concerning this Figure.

**Figure 8 molecules-30-03279-f008:**
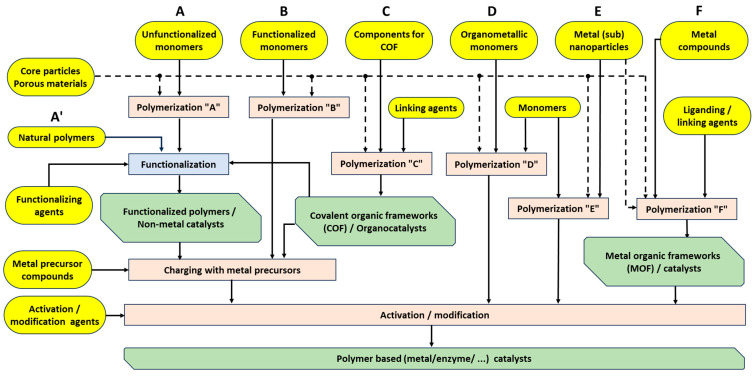
Preparation of Organometallic and Metalloorganic Catalysts. Dashed lines indicate alternatives when core particles, or porous materials serving as carriers are used, Reproduced from [49] with permission from MDPI.

**Figure 9 molecules-30-03279-f009:**
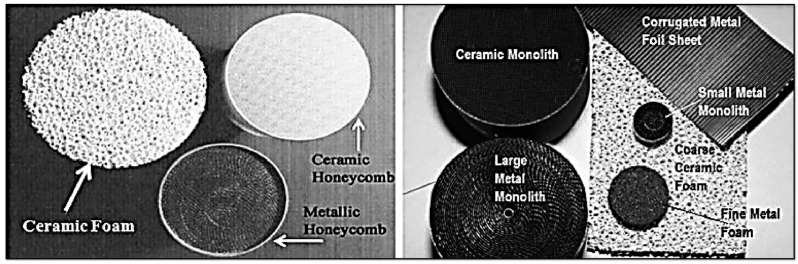
Various types of ceramic and metal monoliths [148]. Reproduced with permission from Elsevier.

**Figure 10 molecules-30-03279-f010:**
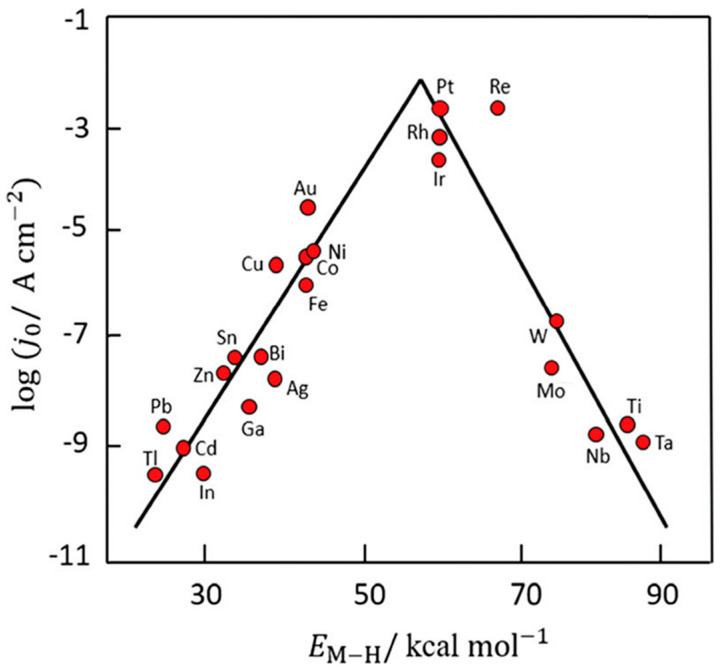
The exchange current density of the hydrogen oxidation reaction vs. M–H bonding strength [176,180]. Reproduced with permission from RSC.

**Figure 11 molecules-30-03279-f011:**
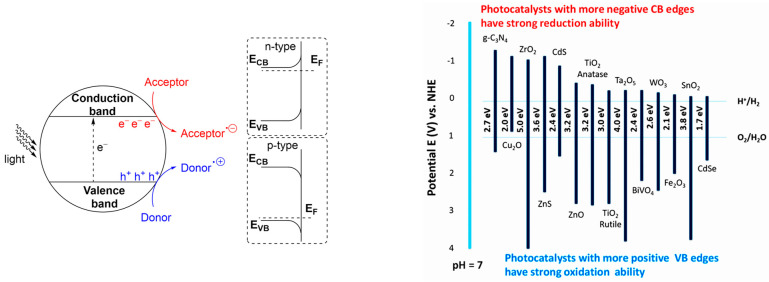
**Left**: Principle of heterogeneous photocatalysis. E_CB_ and E_VB_ are Energy of the valence Band (HOMO) and Conductive Band (can be LUMO), respectively; E_F_—Energy of the Fermi level. Adopted from [192], with permission from Comenius University in Bratislava. **Right**: Band positions and potential applications of some typical photocatalysts (at pH 7 in aqueous solutions) relevant to water splitting [194]. Published with permission from ACS Publications.

**Figure 12 molecules-30-03279-f012:**
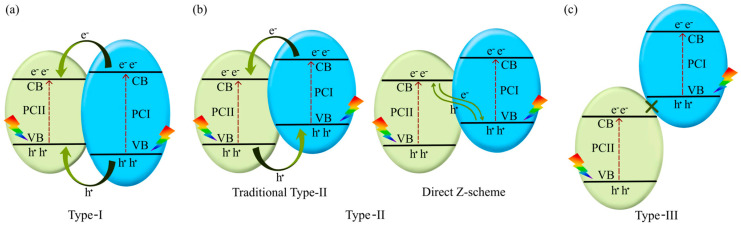
Carrier transfer of (**a**) type-I, (**b**) type-II, and (**c**) type-III heterojunctions [197]. Published with permission from Elsevier.

**Figure 13 molecules-30-03279-f013:**
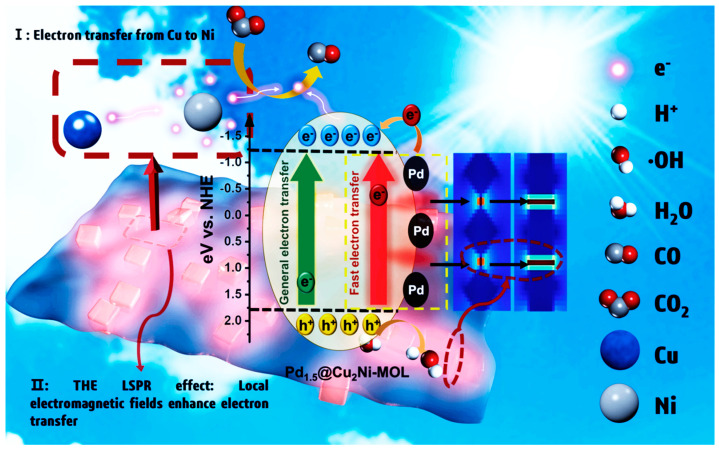
The mechanism of photoreduction conversion of CO_2_ to CO (upper part of Figure) at Pd_1.5_@Cu_2_Ni-MOL (layered MOF). The Cu site continuously supplies electrons to the Ni site through the Cu-O-Ni bond. The near-field enhancement effect generated by Pd NPs further accelerates electron transfer and promotes the photocatalytic reaction [210]. Reproduced with permission from Elsevier.

**Figure 14 molecules-30-03279-f014:**
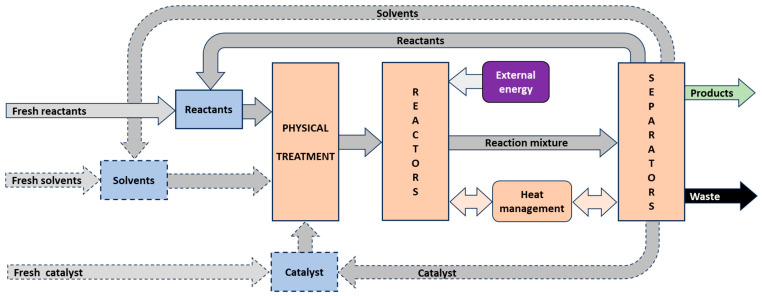
A general arrangement of catalytic technology. Streams in objects with dashed lines indicate alternatives of technology (the chart was prepared using knowledge from [228]).

**Figure 15 molecules-30-03279-f015:**
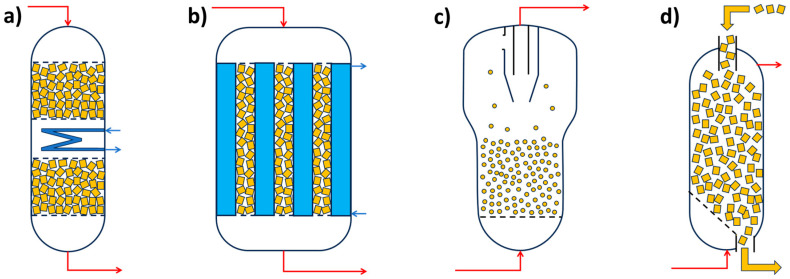
Reactors: (**a**) fixed-bed with a heat exchanger, (**b**) multitube reactor with cooling, (**c**) fluidized with a cyclone separator and (**d**) with a moving catalyst bed (Red arrows indicate flow of reaction components. Tthe images were prepared using knowledge from [56,86]).

**Figure 17 molecules-30-03279-f017:**
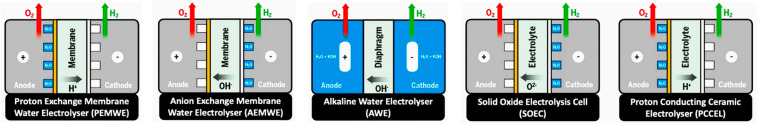
Schematic presentation of the five main types of water ELZ [219]. Published with permission of RSC.

**Figure 18 molecules-30-03279-f018:**
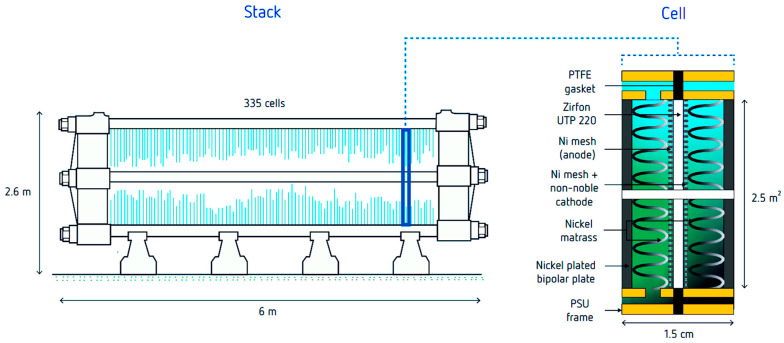
Schematic representation of the advanced AWE stack [277]. Operating temperature: 100 °C, Operating pressure: 5 bar, Nominal current density: 1.3 A cm^−2^, Cell potential—nominal load (at start): 1.8 V, Nominal stack power: 19.6 MW, Current (Faradaic) efficiency: 98%, Nominal stack hydrogen output: 4460 Nm^3^·h^−1^, Minimal load 15%, Composition—H_2_ in O_2_ (nominal load/minimum load) 0.3/1.6%, Degradation rate: 1% per year. Published with permission from ISPT (https://ispt.eu/).

**Figure 19 molecules-30-03279-f019:**
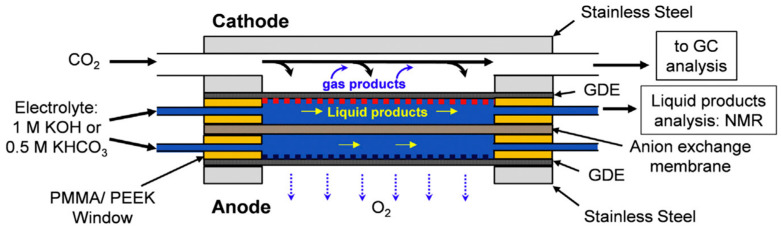
Schematic representation of the electrochemical flow cell used in [295]. Cathode and anode: copper nanoparticles deposited on Gas Diffusion Layer (Sigracet 35 BC, Ion Power) forming Gas Diffusion Electrode (GDE), anion exchange membrane (Fumatech^®^). Published with permission from Elsevier.

**Figure 20 molecules-30-03279-f020:**
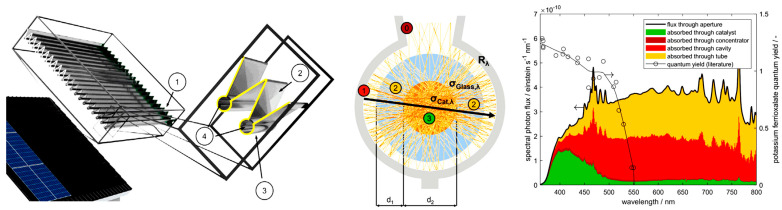
A photoreactor set positioned on the roof of a house. **Left**: Sunlight-driven set (1) of photoreactors (2) with mirrors (3) and photoreactors coated with a photocatalyst (4); **Middle**: One segment of the photoreactor with minimal inherent parasitic absorption share through concentrator (0), cavity walls (1), and glass (2) and maximum feasible absorption share through the catalyst (3), together with representative ray paths in the free-form optimized concentrator cavity channel with an incidence direction given by a = b = 0° (orange); **Right**: quantum yield of the potassium iron(III) oxalate system (empty circles with thin solid line fit). The “solar” spectrum underlying the presentation corresponds to the experimentally determined spectrum of the class ABA solar simulator employed in [312]. Published with permission from Elsevier.

**Figure 21 molecules-30-03279-f021:**
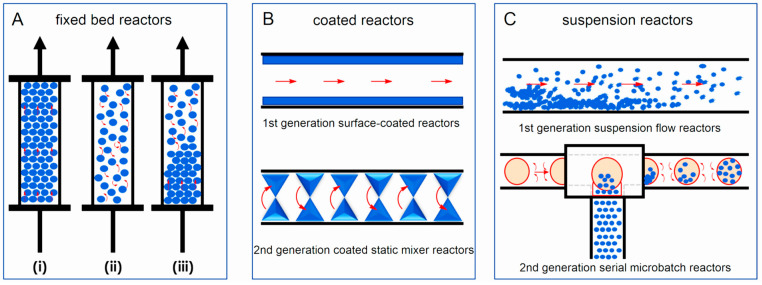
Types of reactors employed in heterogeneous photocatalysis (HP) in flow [219]. (**A**) Fixed-bed reactors and the subcategories of (i) packed bed, (ii) fluidized bed, and (iii) hybrid/mixed bed. (**B**) Coated reactors in which the HPCat is immobilized onto a surface. (**C**) Suspension reactors that have the HPCat freely flowing through reactor channels. Published with permission from Beilstein Journal of Organic Chemistry.

**Figure 22 molecules-30-03279-f022:**
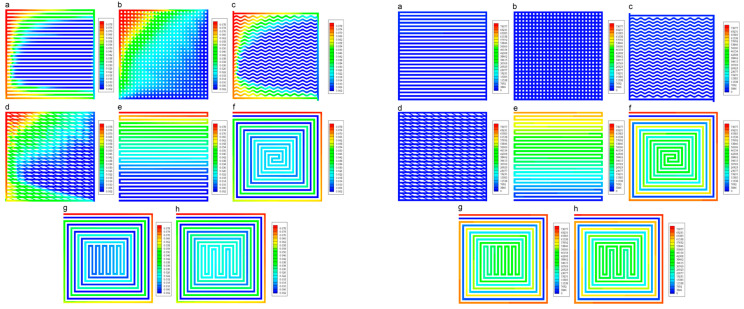
Modeling of catalytic oxidation of methane in catalytic microreactors [402]; Pt deposited on walls of channels 1 × 1 mm, total length of channels approx. 1.4 m, Inlet temperature: 300 K, Wall temperature: 1290 K, Outlet pressure: 101,325 Pa, Re ≈ 1500. Input mole fraction for CH_4_: 0.078. Quantities along the mid-plane of the flow channel (z = 5 ×10^−4^ m). **Left**: mole fraction of CH_4_. **Right**: Pressure distribution (Pa). MR types: (**a**) parallel; (**b**) pin-hole; (**c**) wavy; (**d**) oblique fin; (**e**) serpentine; (**f**) coiled; (**g**) coiled with serpentine; and (**h**) coiled with double serpentine. Reproduced from [402] with permission from Elsevier.

**Figure 23 molecules-30-03279-f023:**
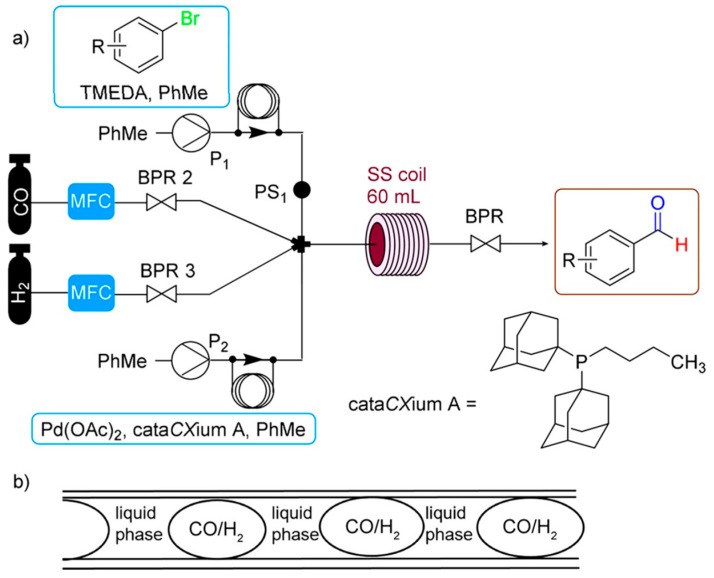
(**a**) Optimized continuous-flow configuration for reductive carbonylation (formylation) of aryl bromides using carbon monoxide and hydrogen [410]; 60 mL stainless-steel coil reactor (1/8 in OD, 1/16 in ID), 0.25 M aryl bromides in anhydrous PhMe, 5 mol% Pd(OAc)2, 15 mol% cataCXium A, 0.75 equiv. TMEDA (*N*,*N*,*N*′,*N*′-Tetramethylethylenediamine), the liquid pumps were set at equal flow rates, T = 100–140 °C, P = 5 or 10 bar. MFC—mass flow controller, BPR—back pressure regulator; (**b**) gas–liquid segmented (Taylor) flow regime. Published with permission from Wiley-VCH GmbH.

**Figure 24 molecules-30-03279-f024:**
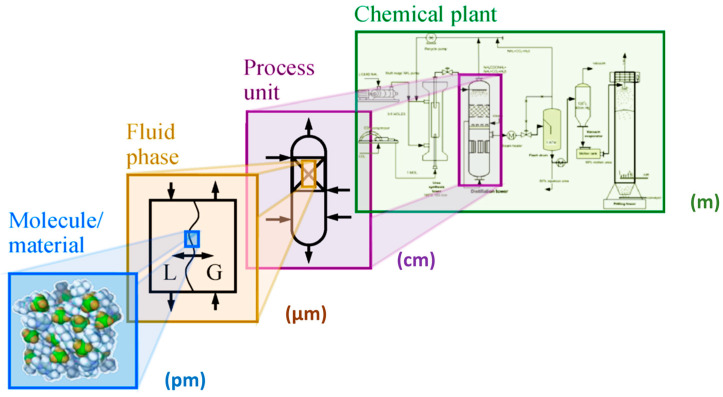
Multiscale vision of a chemical process. Adopted from [431]. Published with permission from Springer.

**Figure 25 molecules-30-03279-f025:**
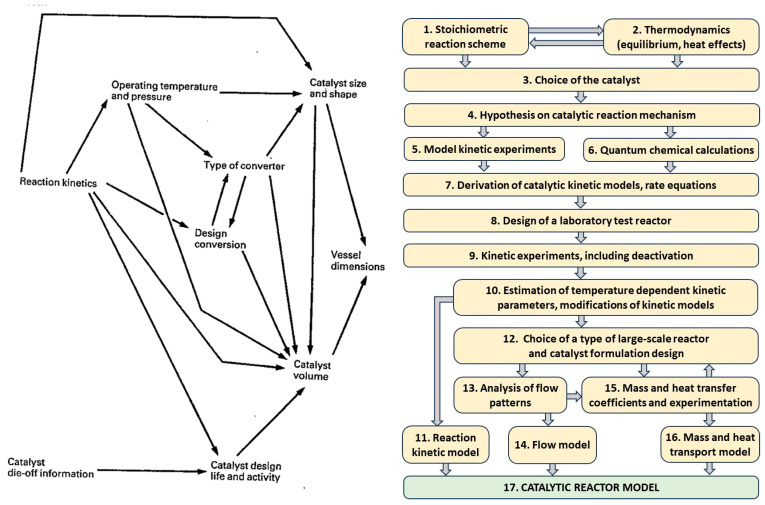
**Left**: Schematic representation of steps for the design of a catalyst and a catalytic reactor [6]. Published with permission from Manson Publishing. **Right**: Flow diagram of activities required for the choice of a proper catalyst and catalytic reactor model.

**Figure 26 molecules-30-03279-f026:**
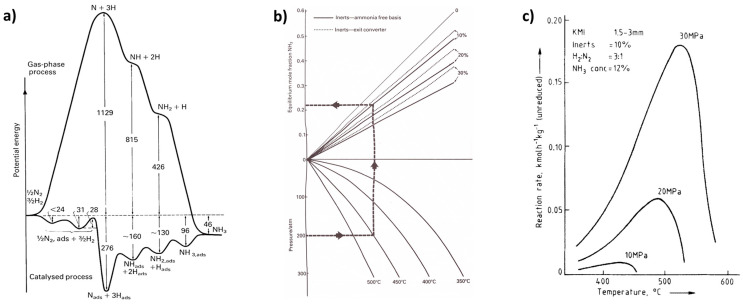
Ammonia synthesis. (**a**): Reaction coordinate with indication of chemisorbed intermediates and comparison with a homogeneous reaction; (**b**): Nomogram for estimation of equilibrium composition at various pressures, temperatures, and effect of inerts; (**c**): Dependence of the reaction rate on temperature and pressure. Reproduced from [6] with permission from Manson Publishing.

**Figure 27 molecules-30-03279-f027:**
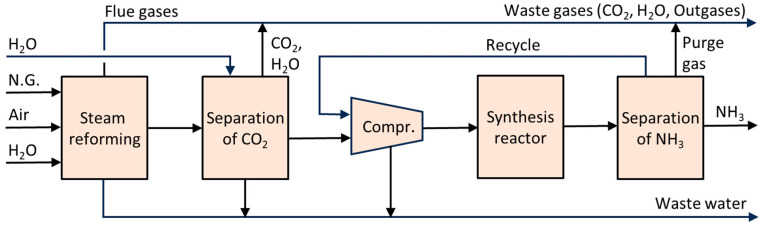
Principal block diagram of ammonia synthesis from natural gas (N.G.), air, and water. CO conversion (methanation) is not involved.

**Figure 28 molecules-30-03279-f028:**
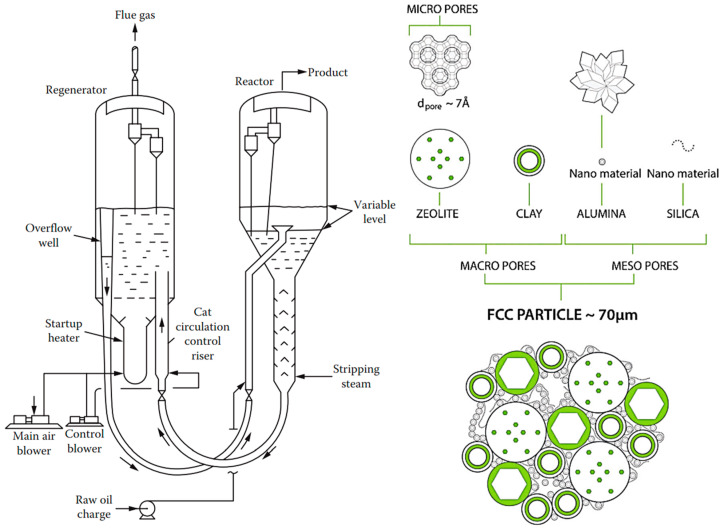
**Left**: One of the alternatives of fluid catalytic cracking reactors [265]. Published with permission from CRC, Taylor & Francis. **Right**: Typical chemical and structural composition of an FCC particle [482]. Published with permission from RSC.

**Figure 29 molecules-30-03279-f029:**
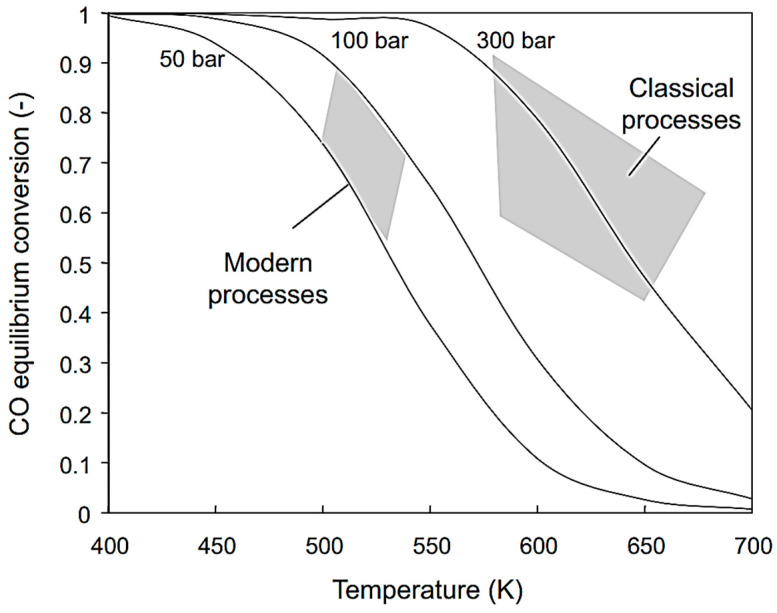
Equilibrium CO conversion to methanol (feed H_2_/CO = 2 mol/mol) [55]. Published with permission from John Wiley & Sons Ltd.

**Figure 30 molecules-30-03279-f030:**
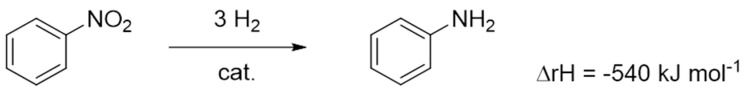
Catalytic hydrogenation of nitrobenzene to aniline.

**Figure 31 molecules-30-03279-f031:**
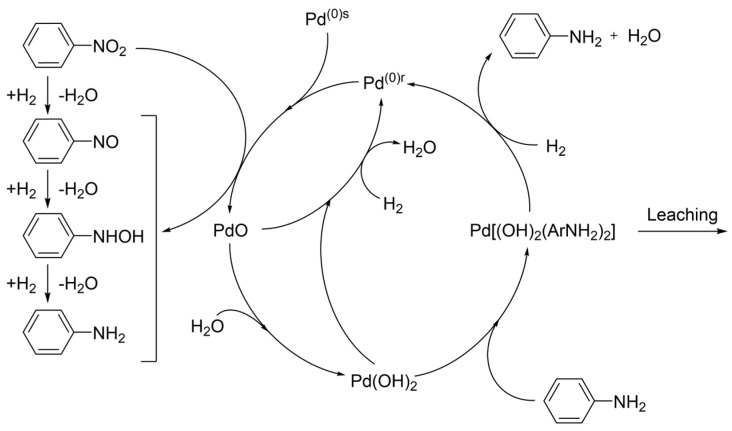
Hydrogenation of nitrobenzene (the Haber’s scheme), accompanied by reactions of metallic palladium (effects of hydrogen activity on the (re-)formation of PdHx are not considered).

**Figure 32 molecules-30-03279-f032:**
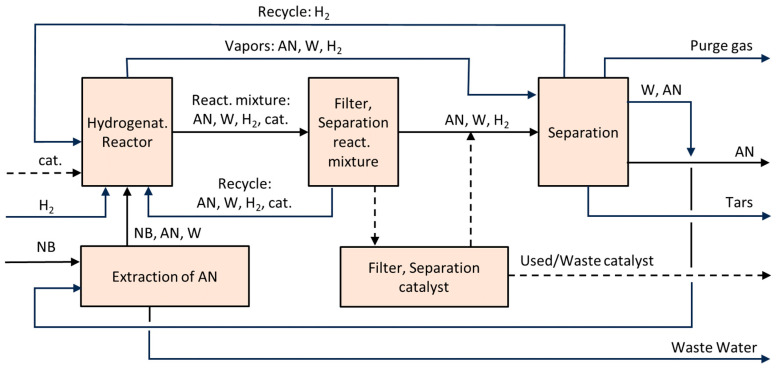
Block flow diagram for NB hydrogenation. Dashed lines indicate periodical loading of fresh catalyst and removal of the used catalyst.

**Figure 33 molecules-30-03279-f033:**
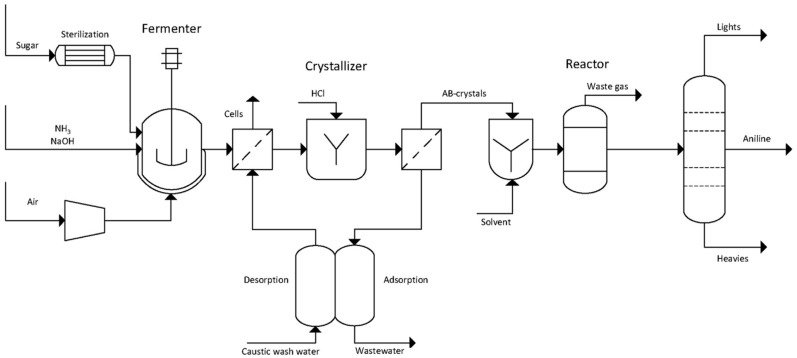
Process flowsheet of bio-based aniline production from glucose and ammonia via fermentation with Corynebacterium Glutamicum [513]. Published with permission from Elsevier Ltd.

**Figure 34 molecules-30-03279-f034:**
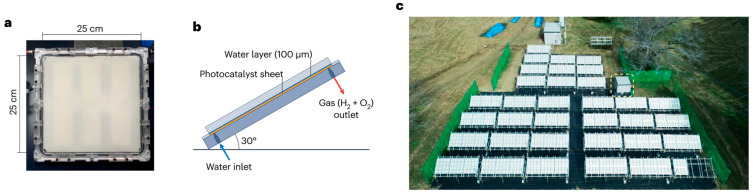
A 100-m^2^ water-splitting photocatalyst panel reactor. (**a**), A panel reactor unit (625 cm^2^). (**b**), Structure of the panel reactor unit viewed from the side. (**c**), Aerial view of a 100-m^2^ solar hydrogen production system consisting of 1600 panel reactor units and a hut housing the gas-separation facility (yellow box). Reproduced from [515] with permission from Nature reviews materials.

**Table 1 molecules-30-03279-t001:** Examples of industrial unsupported catalysts [6,100].

Catalyst	Process	CLT ^(1)^, Deactivation
Fe/Al_2_O_3_/CaO/K Hollow extrudates	NH_3_ synth. (N_2_ + 3H_2_ -> 2NH_3_), 400–550 °C, 150–500 bar	5–10 years, slow sintering
Pt-Pd-Ru-Rh alloy gauze	Oxidation of NH_3_ to NOx, production of HNO_3_, 800–940 °C, 1–10 bar	1–6 months, deterioration/ evaporation of atoms from wires
V_2_O_5_/K_2_SO_4_/SiO_2_ extrudates	Sulfuric acid (2SO_2_ + O_2_ -> 2SO_3_) 400–600 °C, 1 bar	5–10 years, slow physical deterioration (dust), pressure drop
Cu/ZnO/Al_2_O_3_ pellets	CH_3_OH synth. (CO + 2H_2_ -> CH_3_OH) 200–300 °C, 50–200 bar	2–8 years, slow sintering
Co/Mo sulf./Al_2_O_3_ extrudates	Petrochem. Desulf. (R_2_S + 2H_2_ -> 2RH + H_2_S) 300–400 °C, 20–40 bar	2–8 years, slow coking, pressure drop
Ag granules, 1–3 mm	Oxidat. (CH_3_OH + 1/2O_2_ -> HCHO + H_2_O) 500–600 °C, 1 bar	0.3–1 year, coking, loss of selectivity
Raney Ni skeletal particles	Hydrogenation, e.g., vegetable oils, 100–200 °C, 3–30 bar	1–10 days, deactivation by sticking of side products on the surface
Zeolites/SiO_2_-Al_2_O_3_ microspheroids	Fluid catalytic cracking (FCC), 350–550 °C, 1–5 bar	1–10 s, permanent loss of activity, reactivation by burning coke

^(1)^ CLT strongly depends on operational conditions; lower temperature prolongs the CLT; however, the yield per one pass is lower; therefore, reaction and separation units need to be optimized.

**Table 2 molecules-30-03279-t002:** Examples of inorganic catalyst supports.

Support	S ^(1)^ (m^2^ g^−1^)	V_p_ ^(2)^ (cm^3^ g^−1^)	Other Properties	T_op.max_ ^(3)^ (°C)
Graphite [103]	10–100	0.01−0.1	Mesopor. Struct., high el. ^(4)^ and heat conduct. ^(5)^	500
Activated carbon [103]	200–3000	0.6−2	Micropor. Struct., low el. and heat conduct.	400
Carbon nanotubes, [103]	50–500	2–2.5	Micropor. Struct., high el. and heat conduct.	300
Graphene sheets [103]	1500–3000	2–3.5	Micropor. Struct., low el. and heat conduct.	300
Nickel (Raney) [104]	5–30	0.01–0.05	Low micropor. struct. high el. and heat conduct.	300
γ-Al_2_O_3_ [79]	50–300	0.4–0.8	Meso-micropor. Struct., mild el. and heat conduct.	800
α-Al_2_O_3_ [79]	0.3–5	0.01–0.05	Low meso-micropor. Struct., mild el. Conduct., good heat conduct.	1100
SiO_2_—gels [105,106]	100–800	0.2–0.6	Micropor. Struct., mild el. and heat conduct.	400
CaCO_3_—precipitated [107,108]	5–40	0.01–0.05	Micropor. Struct., mild el. and heat conduct.	400
Clays [109]	250–800	0.1–0.3	Micropore, mild el. and heat conduct. Used mainly as an acid catalyst (H^+^ form), after calcination (>400 °C) mixed oxides are formed—basic catalysts	200
Hydrotalcites [110]	100–300	0.1–0.3	Micropore., mild el. and heat conduct. Used mainly as a base catalyst (OH^+^ form), after calcination (>400 °C) mixed oxides are formed—basic catalysts	200
Zeolites [88]	300–800	0.5–2	Regular micropore structure 0.5–2 nm, mild el. and heat conduct, shape selectivity, in H^+^ form very efficient acid catalysts, e.g., in FCC	900
Cordierite, 2MgO·2Al_2_O_3_·5SiO_2_ Skeleton [101,102],	1–8	0.01–0.05	Mild el. and heat conduct. (semiconductor), high mechanical strength and toughness	900
TiO_2_ [79]	20–400	0.1–0.6	Mild el. and heat conduct. (semiconductor), photocatalytic act.	400
Perovskites—“ABO_3_”, [111,112] CaTiO_3_ as a representative	5–40	0.05–0.6	Mild el. and heat conduct. (semiconductor), photocatalytic act.	300
Glasses [79,113]	0.01–0.1	1–40	Low el. and heat conduct. Surface treatment is needed	400

^(1)^ Specific surface, commonly determined by adsorption–desorption measurements of nitrogen. ^(2)^ Volume of pores, commonly determined by adsorption of nitrogen at saturated pressure. ^(3)^ Maximum application temperature also depends on the type of reaction environment. ^(4)^ Electrical conductivity (S m^−1^): low: <10^2^, mild: 10^2^–10^5^, high: >10^5^. ^(5)^ Thermal conductivity (W m^−1^K^−1^): low: <0.5, mild: 0.5–10, high: >10.

**Table 3 molecules-30-03279-t003:** Selected features of G-S heterogeneous catalytic reactors.

Property	Fixed Bed	Multitube	Fluidized	Moving Bed
*D*_cat_ (mm)	2–30	1–5	0.1–0.5	2–30
*V* (m^3^)	3–20	1–10	3–20	3–20
*P*_max_ (MPa)	15	5	3	5
*T*_max_ (°C)	1000	1000	800	600
Mass_transport	Average	Good	Very good	Average
Heat_transport	Bad	Very good	Very good	Bad
CAPEX	Low	Very high	Average	Average
OPEX	Low	Very high	Very High	High
Examples	Oxidation, hydrogenation, alkylation, hydrocracking	Oxidation	Polymerization, cracking	Hydrotreatment

**Table 4 molecules-30-03279-t004:** Selected features of G-L-S heterogeneous catalytic reactors.

Property	Trickle Bed	Mechan. Stirred	Bubble Column	Ebullated	Loop
*D*_cat_ (mm)	2–30	0.01–1	0.01–0.2	0.01–0.2	0.01–0.1
*V* (m^3^)	3–20	10^−4^–1000	1–20	1–20	0.05–5
*P*_max_ (MPa)	5	15	5	3	3
*T*_max_ (°C)	600	500	400	400	300
Mass_transport	Average	Very good	Average	Average	Excellent
Heat_transport	Bad	Very good	Average	Good	Excellent
CAPEX	Low	High	Low	High	Very high
OPEX	Low	High	High	Very high	Very high
Examples	Hydrotreatment	All processes	Hydrogenation	Hydrotreatment	Hydrogenation, Oxidation

## Data Availability

Additional data are available in the cited papers of MK, PK, MM and PL.

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
