# Peer review of "Research and Developments of Heterogeneous Catalytic Technologies†"

_molecules, 2025, doi:10.3390/molecules30153279_

Round 1
Reviewer 1 Report
Comments and Suggestions for Authors
This review is carefully explained, but on the other hand, the content is too boring and unfocused.
In that sense, unfortunately, it is difficult to distinguish the "main objection" addressed in this review.
On the other hand, as the authors state, "This review seeks to encourage more integrative studies among catalyst researchers, promoting the translation of fundamental insights into applied technologies," this paper was written with the intention of converging basic and general knowledge.
Although it is lumped together as "Heterogeneous Catalysis", a wide variety of research is being conducted in this field, and by sharing such basic knowledge among researchers in this field, it can be said that this review is valuable in the sense that it encourages approaches to various issues that have not yet been resolved.
Some researchers in this field are not pure catalytic chemistry researchers who have thoroughly studied the basics of catalytic chemistry, but have intervened in this interdisciplinary field from the fields of other chemistry, such as chemical engineering, physical chemistry, organic chemistry, and inorganic chemistry.
For such researchers, such a concise review seems to be meaningful.
Another point of this review is that it gives examples of how old understandings and interpretations are still being used in cutting-edge fields currently being researched, such as photocatalysis, electrocatalysis and flow technology. There are even examples of extremely new research, which makes it noteworthy in a way that other similar reviews do not.
One part that the reviewer would like to request for improvement is the weakness of the latest catalytic technology.
I assume that the authors narrowed it down from a vast amount of recent research, but if they claim to contribute to green and sustainable chemistry, I would like them to introduce industrial topics related to green chemistry, rather than the hydrogenation of nitrobenzene.
Ultimately, the reviewer recognizes that this paper is significant from the perspective of fulfilling the basics of heterogeneous catalytic technology. On the other hand, I would also like to point out that the focus of the review is blurred by being too basic. In addition, it is expected that examples such as contributions to green chemistry from the perspective of chemical industry will be added. If this basic review article is in line with the aims of Molecules, the reviewer agrees to its publication once improvements are made in response to the above comments.
Author Response
Comments and Suggestions for Authors
This review is carefully explained, but on the other hand, the content is too boring and unfocused.
Comments 1: In that sense, unfortunately, it is difficult to distinguish the "main objection" addressed in this review.
On the other hand, as the authors state, "This review seeks to encourage more integrative studies among catalyst researchers, promoting the translation of fundamental insights into applied technologies," this paper was written with the intention of converging basic and general knowledge.
Response 1: We have added some notes to the Introduction to help readers navigate the text more easily. Lines: 139 - 146
Comments 2: Although it is lumped together as "Heterogeneous Catalysis", a wide variety of research is being conducted in this field, and by sharing such basic knowledge among researchers in this field, it can be said that this review is valuable in the sense that it encourages approaches to various issues that have not yet been resolved.
Response 2: We agree with the reviewer that many open questions remain in modern heterogeneous catalysis. Nevertheless, the classification of key areas, supported by cited references, may assist readers in exploring potential answers to these questions.
Comments 3: Some researchers in this field are not pure catalytic chemistry researchers who have thoroughly studied the basics of catalytic chemistry, but have intervened in this interdisciplinary field from the fields of other chemistry, such as chemical engineering, physical chemistry, organic chemistry, and inorganic chemistry.
For such researchers, such a concise review seems to be meaningful.
Response 3: Again, we agree with the reviewer and would like to reiterate that this paper is intended as a modest provocation—to encourage a more complex approach to solving catalytic problems and to promote the practical application of improved results.
Comments 4: Another point of this review is that it gives examples of how old understandings and interpretations are still being used in cutting-edge fields currently being researched, such as photocatalysis, electrocatalysis and flow technology. There are even examples of extremely new research, which makes it noteworthy in a way that other similar reviews do not.
Response 4: The availability of technological examples of new or modern catalytic procedures is rather limited. Therefore, the examples focus primarily on 'old established technologies'; however, new variants—though not yet implemented in practice—are also discussed. Lines: 2397 - 2449
Comments 5: One part that the reviewer would like to request for improvement is the weakness of the latest catalytic technology.
Response 5: We have added a new example on water splitting, along with a few lines addressing the catalytic treatment of plastic waste. Lines: 2397 - 2449
Comments 6: I assume that the authors narrowed it down from a vast amount of recent research, but if they claim to contribute to green and sustainable chemistry, I would like them to introduce industrial topics related to green chemistry, rather than the hydrogenation of nitrobenzene.
Response 6: See the comment above. As for production of aniline, there is a “green route” described in Figure 32.
Comments 7: Ultimately, the reviewer recognizes that this paper is significant from the perspective of fulfilling the basics of heterogeneous catalytic technology. On the other hand, I would also like to point out that the focus of the review is blurred by being too basic. In addition, it is expected that examples such as contributions to green chemistry from the perspective of chemical industry will be added. If this basic review article is in line with the aims of Molecules, the reviewer agrees to its publication once improvements are made in response to the above comments.
Response 7: Green technologies are mentioned in the part regarding treatment of plastic waste.
Thank you very much for reviewing the manuscript
Reviewer 2 Report
Comments and Suggestions for Authors
The manuscript provides a thorough overview of heterogeneous catalytic technologies, spanning fundamental principle to industrial applications. It systematically addresses catalyst properties, kinetics, mass/heat transport, reactor design, and techno-economic considerations, forming a cohesive framework from theory to practice. The manuscript may be accepted for publication after considering the following issues.
- There are numerous books or reviews on catalysis. What, then, sets this apart from the others?
- Preparation methodology is crucial for catalytic performance of a catalyst. However, the manuscript missed this important part, why?
- Sections on electrocatalysis and photocatalysis (Sections 3.6–3.7) lack sufficient mechanistic detail compared to other sections. For example, the discussion of plasmonic catalysis could benefit from more specific reaction mechanisms or case studies.
Author Response
Comments and Suggestions for Authors
The manuscript provides a thorough overview of heterogeneous catalytic technologies, spanning fundamental principle to industrial applications. It systematically addresses catalyst properties, kinetics, mass/heat transport, reactor design, and techno-economic considerations, forming a cohesive framework from theory to practice. The manuscript may be accepted for publication after considering the following issues.
Comments 1: There are numerous books or reviews on catalysis. What, then, sets this apart from the others?
Response 1: Books are cited in accordance with Molecules rules.
Comments 2: Preparation methodology is crucial for catalytic performance of a catalyst. However, the manuscript missed this important part, why?
Response 2: The section on catalyst preparation technologies has been expanded with Figures 5, 7 and 8 and accompanying commentary; Lines: 576 – 593, 706 - 726
Comments 3: Sections on electrocatalysis and photocatalysis (Sections 3.6–3.7) lack sufficient mechanistic detail compared to other sections. For example, the discussion of plasmonic catalysis could benefit from more specific reaction mechanisms or case studies.
Response 3: The scope of this paper is not to provide detailed descriptions of individual catalytic phenomena, as such information can be found in the cited literature. Nevertheless, we have added a few lines regarding plasmonic catalysis. Lines: 1033 - 1047
Thank you very much for reviewing the manuscript
Reviewer 3 Report
Comments and Suggestions for Authors
This article reviews the comprehensive methods for the research and development of multiphase catalytic technology (R&D_HeCaTe). The focus is on the fundamental interactions between reactants, solvents, and heterogeneous catalysts, particularly the roles of catalytic centers and support materials (such as functional groups) in regulating activation energy and stabilizing catalytic functions. However, this review still has some issues that need to be improved. It is recommended that a major revision be conducted before being accepted by this journal. The specific problems are as follows:
1. The content layout should be adjusted, and appropriate linking words should be used to enhance the logical flow of the text.
2. In lines 32 and 33 of this article, the number of unnecessary references should be reduced to avoid the phenomenon of literature accumulation.
3. The layout of this article should be adjusted to avoid the occurrence of a large number of sub-points (for example, from line 1022 to 1047). Additionally, the images and tables of this article should be adjusted to make the images and text more coordinated and to make the layout more aesthetically pleasing. For example, the size of the image at line 382 needs to be adjusted.
4. The extensive analysis of basic definitions and phenomenon explanations in this article should be supported by some articles related to the mechanism to deeply analyze how the mechanism affects the phenomenon.
5. There are many inconsistent reference formats in this article. Please unify the reference format of the entire article according to the journal requirements.
6. This article contains some grammatical errors. The article should be polished.
7. The content of this article should be deepened. Instead of merely analyzing conditions such as temperature and particle size, some literature related to microstructure, intrinsic electronic structure, etc. can be cited and analyzed in depth. For example, the analysis of coherent interfaces and incoherent interfaces can be added.
8. From an economic perspective, please analyze the impact of actual production factors, operating conditions, etc. on the economy, and emphasize the best solution.
9. Please deepen the depth of the introduction and cite some latest literature to reflect the current research status. For example: (1) https://doi.org/10.1002/elt2.58 (2) https://doi.org/10.1002/elt2.70000
The English level should be improved.
Author Response
Comments and Suggestions for Authors
This article reviews the comprehensive methods for the research and development of multiphase catalytic technology (R&D_HeCaTe). The focus is on the fundamental interactions between reactants, solvents, and heterogeneous catalysts, particularly the roles of catalytic centers and support materials (such as functional groups) in regulating activation energy and stabilizing catalytic functions. However, this review still has some issues that need to be improved. It is recommended that a major revision be conducted before being accepted by this journal. The specific problems are as follows:
Comments 1: The content layout should be adjusted, and appropriate linking words should be used to enhance the logical flow of the text.
Response 1: In our experience, structured text is more comprehensible than plain text. This point is also emphasized in the Introduction; Lines: 139 - 146.
Comments 2: In lines 32 and 33 of this article, the number of unnecessary references should be reduced to avoid the phenomenon of literature accumulation.
Response 2: We have reduced the number of references by 76 and added 11, Lines: 93 – 99, 1567 – 1580.
Comments 3: The layout of this article should be adjusted to avoid the occurrence of a large number of sub-points (for example, from line 1022 to 1047). Additionally, the images and tables of this article should be adjusted to make the images and text more coordinated and to make the layout more aesthetically pleasing. For example, the size of the image at line 382 needs to be adjusted.
Response 3: Respectfully, we must disagree with the reviewer on this point. The paper presents a substantial amount of technical information, and it is a common practice in technical writing to present such data in a structured format. A comment to this was added: Lines: 139 – 146.
Regarding the image size (Figure 1), it conforms to standard formatting commonly accepted by open-access journals. Higher-resolution images will be included in the final version of the manuscript.
Comments 4: The extensive analysis of basic definitions and phenomenon explanations in this article should be supported by some articles related to the mechanism to deeply analyze how the mechanism affects the phenomenon.
Response 4: The intention of this paper is to inspire the exploration of connections between basic, applied, and technological research. We are aware of the vast number of topics within these areas. The selection of data was made to highlight the principles and applications or effects of specific phenomena in catalytic processes/steps. Detailed information can be found in the cited literature.
Comments 5: There are many inconsistent reference formats in this article. Please unify the reference format of the entire article according to the journal requirements.
Response 5: The references have been reviewed and corrected. We hope to have minimized errors, which will be addressed during the proofreading stage.
Comments 6: This article contains some grammatical errors. The article should be polished.
Response 6: The English grammar has been checked and corrected by a professional technical writer.
Comments 7: The content of this article should be deepened. Instead of merely analyzing conditions such as temperature and particle size, some literature related to microstructure, intrinsic electronic structure, etc. can be cited and analyzed in depth. For example, the analysis of coherent interfaces and incoherent interfaces can be added.
Response 7: Thank you very much for highlighting the concepts of coherent and incoherent interfaces. These terms are not commonly used in classical catalysis but have been introduced through more in-depth studies in photocatalysis. We have added relevant lines to the Introduction to reflect this; Lines: 58 – 65.
Regarding a more detailed theoretical discussion, we would like to repeat that the intention of this paper is to focus on the connections with practical applications. Detailed explanations of catalytic phenomena can be found in the cited references.
Comments 8: From an economic perspective, please analyze the impact of actual production factors, operating conditions, etc. on the economy, and emphasize the best solution.
Response 8: Conducting a detailed and precise economic analysis is a challenging task, as it must consider various technical and social factors, such as environmental regulations, wages, and more. Therefore, such an analysis needs to be tailored to specific geo-demographic conditions and the relevant regulatory framework. We have added lines addressing this point in the text, Lines: 1957 – 2061.
Comments 9: Please deepen the depth of the introduction and cite some latest literature to reflect the current research status. For example: (1) https://doi.org/10.1002/elt2.58 (2) https://doi.org/10.1002/elt2.70000
Response 9: About sixty older citations were removed in the revised manuscript.
The newly mentioned citations were added along with brief comments. Lines 992 – 998, 1400 -1412.
Comments 10: Quality of English Language: The English level should be improved.
Response 10: The English grammar has been checked by a professional technical writer.
Thank you very much for reviewing the manuscript
Reviewer 4 Report
Comments and Suggestions for Authors
Thank the authors for their very detailed review of heterogeneous catalysis, which is well-organized and systematic.
The article not only introduces the basic theory and characterization methods of heterogeneous catalysis, but also covers catalyst types, reactor processes, industrial examples, economic and environmental evaluations, etc., which is both academic and engineering application perspectives. It helps the reader to understand the many processes involved in catalysts from development to application. This is the highlight of this review in my opinion. I think it will be of interest to researchers and technicians in the field of heterogeneous phase catalysis.
Author Response
Comments and Suggestions for Authors
Thank the authors for their very detailed review of heterogeneous catalysis, which is well-organized and systematic.
Comments 1: The article not only introduces the basic theory and characterization methods of heterogeneous catalysis, but also covers catalyst types, reactor processes, industrial examples, economic and environmental evaluations, etc., which is both academic and engineering application perspectives. It helps the reader to understand the many processes involved in catalysts from development to application. This is the highlight of this review in my opinion. I think it will be of interest to researchers and technicians in the field of heterogeneous phase catalysis.
Response 1: Thank You for reviewing the manuscript.
Round 2
Reviewer 3 Report
Comments and Suggestions for Authors
The author has successfully resolved the issues present in the manuscript. Now it is suitable for publication in Molecules.